# Genome-wide CRISPR screens identify novel regulators of wild-type and mutant p53 stability

YiQing Lü [1,2,8], Tiffany Cho [1,2], Saptaparna Mukherjee[3], Carmen Florencia Suarez[4], Nicolas S Gonzalez-Foutel[4], Ahmad Malik [1,2], Sebastien Martinez[1], Dzana Dervovic[1], Robin Hyunseo Oh[1,2], Ellen Langille[1,2], Khalid N Al-Zahrani [1], Lisa Hoeg [1], Zhen Yuan Lin[1], Ricky Tsai [1], Geraldine Mbamalu[1], Varda Rotter[3], Patricia Ashton-Prolla[5], Jason Moffat [2,6,7], Lucia Beatriz Chemes[4], Anne-Claude Gingras [1,2], Moshe Oren [3], Daniel Durocher [1,2] & Daniel Schramek [1,2✉]

## Abstract

**Tumor suppressor p53 (TP53) is frequently mutated in cancer, often resulting not only in loss of its tumor-suppressive function but also acquisition of dominant-negative and even oncogenic gain-of-function traits. While wild-type p53 levels are tightly regulated, mutants are typically stabilized in tumors, which is crucial for their oncogenic properties. Here, we systematically profiled the factors that regulate protein stability of wild-type and mutant p53 using marker-based genome-wide CRISPR screens. Most regulators of wild-type p53 also regulate p53 mutants, except for p53 R337H regulators, which are largely private to this mutant. Mechanistically, FBXO42 emerged as a positive regulator for a subset of p53 mutants, working with CCDC6 to control USP28-mediated mutant p53 stabilization. Additionally, C16orf72/HAPSTR1 negatively regulates both wild-type p53 and all tested mutants. C16orf72/HAPSTR1 is commonly amplified in breast cancer, and its overexpression reduces p53 levels in mouse mammary epithelium leading to accelerated breast cancer. This study offers a network perspective on p53 stability regulation, potentially guiding strategies to reinforce wild-type p53 or target mutant p53 in cancer.**

**Keywords** Genome-wide CRISPR Screening; Mutant p53; p53 Stability; Breast Cancer; Fluorescence-based Stability Reporter
**Subject Categories** Cancer; Chromatin, Transcription & Genomics; Proteomics

## Introduction

Approximately half of all tumors harbor mutations in the p53 (*TP53*) gene, making *TP53* the most commonly mutated gene in cancer (Freed-Pastor and Prives, 2012; Oren and Rotter, 2010; Petitjean et al, 2007; Vogelstein et al, 2000). The majority of these mutations are missense mutations, most of which not only deprive p53 of its tumor suppressor activities, but might also function in a dominant-negative manner, suppressing canonical p53 functions upon oligomerization with wild-type p53 (Chene, 1998; Giacomelli et al, 2018; Liu et al, 2010; Malkin, 2011; Varley et al, 1997). Some missense mutations also confer a so-called gain-of-function (GOF) phenotype, converting mutant p53 into a cancer-promoting protein that renders cancer cells more malignant by increasing growth rate, motility, invasion, drug resistance, and tumorigenicity, while reducing the apoptotic rate (Blandino et al, 1999; Dittmer et al, 1993; Eliyahu et al, 1984; Muller and Vousden, 2013; Oren and Rotter, 2010; Peled et al, 1996; Vogelstein et al, 2000; Weisz et al, 2004; Wolf et al, 1984; Zalcenstein et al, 2003). Importantly, compared to tumors with wild-type p53 or those lacking p53 altogether, tumors expressing GOF mutant p53 are more invasive, metastatic, and proliferative, and display increased genome instability and chemoresistance in mouse and human (Blandino et al, 1999; Bougeard et al, 2008; Lang et al, 2004; Liu et al, 2000; Morton et al, 2010; Olive et al, 2004; Oren and Rotter, 2010; Zerdoumi et al, 2013). Two main classes of p53 hotspot mutations have been distinguished — those that affect residues directly involved in protein-DNA interaction, such as R248 or R273 ("contact mutants"), and those that affect residues involved in stabilizing the tertiary structure of the protein, such as R175, G245, R249 and R282 ('conformational' or 'structural' mutants).

Germline *TP53* mutations also exist, often target the same hotspot residues, and are the underlying cause of Li-Fraumeni

[1]Centre for Molecular and Systems Biology, Lunenfeld-Tanenbaum Research Institute, Mount Sinai Hospital, Toronto, Ontario M5G 1X5, Canada. [2]Department of Molecular Genetics, University of Toronto, Toronto, Ontario M5S 1A8, Canada. [3]Department of Molecular Cell Biology, Weizmann Institute of Science, Rehovot, Israel. [4]Instituto de Investigaciones Biotecnológicas (IIBiO-CONICET), Universidad Nacional de San Martín, Buenos Aires, Argentina. [5]Departamento de Genética, Universidade Federal do Rio Grande do Sul and Serviço de Genetica Médica HCPA, Porto Alegre, Brasil. [6]Institute of Biomedical Engineering, University of Toronto, Toronto, Ontario M5S3G9, Canada. [7]Genetics and Genome Biology Program, Hospital for Sick Children, Toronto, Ontario M5G 1X8, Canada. [8]Present address: Department of Biology, Suffolk University, Boston, MA 02108, USA. ✉E-mail: schramek@lunenfeld.ca

Syndrome, which predisposes to a wide spectrum of early-onset cancers. In Brazil, the *TP53* R337H founder mutation exists at high frequency and represents the most common germline *TP53* mutation reported to date (Bouaoun et al, 2016). Interestingly, unlike most hotspot mutations, this mutation is not located in the DNA-binding domain of p53 but in the oligomerization domain and disrupts p53 oligomerization.

While wild-type (WT) p53 protein expression is tightly regulated and kept at a low under homeostatic conditions, mutant p53 is often stabilized and highly overexpressed in tumors, which is thought to be required for mutant p53 to exert its oncogenic effects. In fact, strong immunohistochemical staining patterns of nuclear p53 still serve as a surrogate marker for *TP53* mutations in the clinic (Rotter, 1983; Yemelyanova et al, 2011). Interestingly, knock-in mouse models of p53 and Li-Fraumeni patients carrying germline p53 GOF mutations highly express mutant p53 specifically in tumor cells, but show low or undetectable levels of mutant p53 in the surrounding, phenotypically normal tissues (Lang et al, 2004; Olive et al, 2004; Oren and Rotter, 2010; Terzian et al, 2008). This observation indicates that mutant p53 is not intrinsically stable and that its levels are kept in check in healthy cells, but that this regulation is perturbed in cancer (Lang et al, 2004; Olive et al, 2004; Oren and Rotter, 2010). The stability of wild-type p53 is regulated mainly through MDM2/4-mediated ubiquitination and degradation (Eischen and Lozano, 2014; Haupt et al, 1997; Kubbutat et al, 1997), but little is known about the factors that regulate mutant p53 stability.

Here, we employed functional genomics and proteomics approaches to systematically profile the processes that regulate the stability of wild-type, as well as of the most common p53 mutants that collectively account for ~50% of all mutant p53 (Bouaoun et al, 2016). These screens identified 864 genes whose loss either increases or decreases the stability of p53 mutants. Mining this dataset, we report that the FBXO42-CCDC6-USP28 axis acts as a positive regulator of mutant p53 stability, and the C16orf72/HAPSTR1-HUWE1-USP7 axis acts as a negative regulator of p53 stability.

# Results

## A fluorescence-based p53 stability reporter system

To monitor p53 stability at the single-cell level, we generated a lentiviral protein stability reporter (Fig. 1A) consisting of a p53-mClover-P2A-mRFP cassette that permits translation of a mClover-p53 protein fusion and a red fluorescent RFP protein from the same mRNA transcript, similar to previous protein stability reporters (Yen et al, 2008; Yu et al, 2014). The p53-mClover fusion assesses p53 stability, while mRFP serves as an internal control to monitor the expression of the bicistronic transgene (Fig. 1A).

To benchmark this reporter, we used an hTERT-immortalized retinal pigment epithelium-1 (RPE1) cell line that expresses Cas9 and was previously used in CRISPR screens (Hart et al, 2015). We used an RPE1 subclone in which the gene encoding p53 was knocked out by gene editing (Noordermeer et al, 2018) and generated isogenic lines expressing stability reporters containing either wild-type p53 or eight of the most common hotspot p53

mutations (R175H, G245S, R248Q, R248W, R249S, R273H, R282W, and R337H). As a control, we also generated a cell line expressing the mClover-P2A-RFP cassette without p53. Upon activation by the MDM2 inhibitor Nutlin-3a, the levels of the WT p53 reporter were comparable to endogenous p53 levels in parental RPE1 cells. We also observed upregulation, at the protein level, of the p53 target gene *CDKN1A* (encoding p21) upon Nutlin-3a treatment in the reporter cell line, albeit to a reduced level compared to the parental cell line. The R273H p53 reporter showed increased p53 levels even in untreated cells, but did not upregulate p21 after Nutlin-3a addition, as expected for p53 hotspot mutations (Appendix Fig. 1A).

We observed that cells transduced with p53 G245S, R248Q, R248W, or R273H exhibited bimodal distribution of p53 levels, with one subset expressing hardly any p53 and the other subset expressing higher levels of p53. This pattern of p53 protein expression was observed with both structural (G245S) and contact (R248Q, R248W, R273H) p53 mutants (Fig. 1B). Such bimodal distribution was also observed in lymphoma cell lines expressing endogenous p53 R248Q, R248W or R273H mutants (Jethwa et al, 2018) and thus likely reflects the intrinsically unstable nature of mutant p53 (Terzian et al, 2008). In line with previous data showing that MDM2 promotes the degradation of wild-type and some mutant p53(Haupt et al, 1997; Lukashchuk and Vousden, 2007; Terzian et al, 2008), treatment with Nutlin-3a elevated the levels of wild-type p53 and the p53 mutants G245S, R248Q, R248W, and R273H, while R175H, R249S, R282W and R337H did not respond to Nutlin-3a (Fig. 1B). Upon removal of Nutlin-3a, levels of wild-type and mutant p53 returned to baseline levels, re-establishing the uni- or bimodal distribution of the starting population (Fig. 1B). Similarly, we observed that irradiation led to significantly increased levels of p53 mutants (G245S, R248Q, R248W, R273H) with the exception of p53 R175H, R249S, R282W, and R337H (Appendix Fig. 1B), further indicating that several p53 mutants and especially the contact p53 mutants are regulated by the same machinery that regulates wild-type p53. In line with these data, CRISPR/Cas9-mediated ablation of MDM2 elevated the p53^R273H^-mClover, which could be reversed by ectopically re-expressing MDM2, further validating our reporter system (Fig. 1C).

## Genome-wide CRISPR screens identify regulators of wild-type and mutant p53 stability

To identify regulators of p53 stability, we performed genome-wide pooled CRISPR screens in RPE1-hTERT Cas9 cells expressing either the control mClover-P2A-RFP cassette or the stability reporters for p53 wild-type, R175H, G245S, R248Q, R273H, or R337H. We used the TKOv3 lentiviral sgRNA library, which contains 70,948 guides (~4 guides/gene) targeting 18,053 protein-coding genes and 142 control sgRNAs targeting EGFP (N.B. mClover is a monomeric variant of GFP, so it is targeted by the GFP guides), LacZ, and luciferase (Hart et al, 2017). We transduced the reporter lines at 200X coverage, selected for infected cells and isolated cell populations expressing low or high p53-mClover by flow cytometry (Fig. 1D). For reporter lines with a unimodal p53 distribution, we sorted cells with the lowest and highest 15% of p53-mClover expression, and for lines with a bimodal p53 distribution, we isolated the lower and the upper populations.

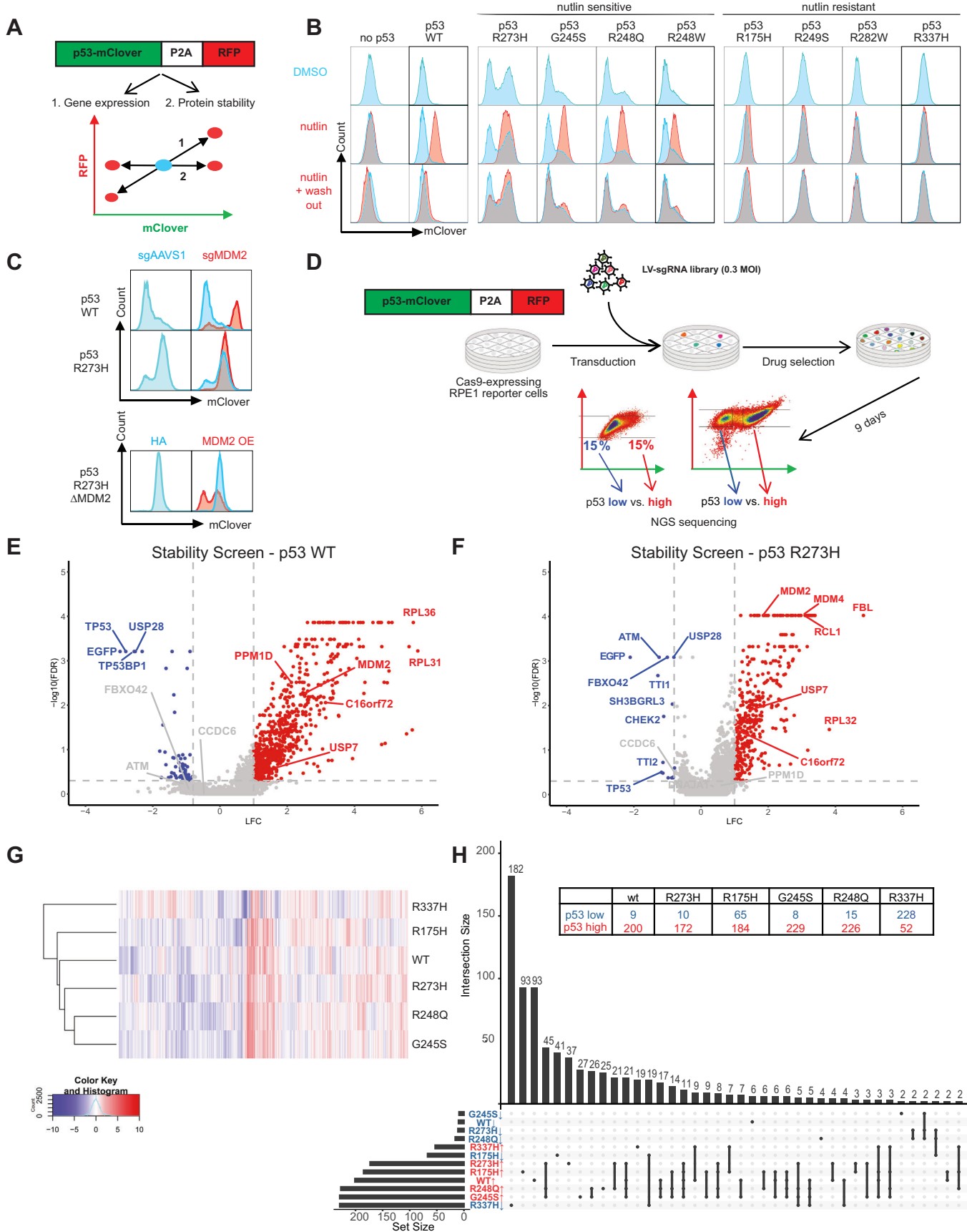

**Figure 1. CRISPR screen for regulators of wild-type and mutant p53 stability.**

(A) Reporter design. Schematic of the fluorescence reporter based on a bicistronic p53-mClover stability sensor followed by a co-translational self-cleaving P2A peptide and mRFP that serves as an internal control. Any perturbation that specifically affects p53 stability would result in an altered mClover/RFP ratio (#2 horizontal axis), while any perturbation that results in overall increased transcription or general differences of proteostasis would affect RFP as well as p53-mClover (#1 diagonal axis). (B) Flow cytometry blots depicting the level of wild-type and mutant p53 protein reporters upon Nutlin-3a (10 μM, 24 h) treatment to inhibit the interaction between MDM2 and p53 and upon Nutlin-3a withdrawal (24 h). Results are reproducible over biological triplicates. (C) Flow cytometry blots depicting the levels of wild-type and p53 R273H protein reporters upon CRISPR/Cas9-mediated MDM2 depletion (top), or upon subsequent MDM2 overexpression (bottom), measured 7 days post-transduction. Results are reproducible over biological triplicates. (D) Schematic of the screening setup and analysis. Each clonal reporter line was transduced with a lentiviral genome-wide CRISPR knockout library (TKOv3). Infected populations were drug-selected and sorted by Fluorescence-Activated Cell Sorting (FACS) into mClover-low and mClover-high pools. sgRNA barcodes were amplified and their abundance in each pool was determined by next-generation sequencing. (E, F) Screen results for wild-type and p53 R273H protein reporters. Volcano plots displaying the perturbation effects (log 2-fold change, LFC) of each gene based on two replicates per screen. To compare among different screens (G, H), the perturbation effects of each gene were further normalized as normZ scores, considering both the LFC and false discovery rate (FDR) values. Hits were defined as having |normZ| ≥ 3. On the volcano plots, hits satisfying both FDR < 0.5, and LFC ≥ 1 (red) or LFC ≤ −0.8 (blue), are labeled in red for genes whose losses lead to increased p53 levels and blue for genes whose losses decreased p53. Screens were performed in two technical replicates. (G) Unsupervised hierarchy clustering of the wild-type and five screened p53 mutants, using the normalized screening results (normZ) of all 18053 genes. (H) UpSet plot displaying the relationships of hits shared amongst each mutant screened. The loss-of-function of an "up" (red) or "down" (blue) hit would result in the p53 mutant to destabilize or stabilize, respectively. Each column on the plot denotes a set of WT and/or p53 mutants, and the histogram above indicates the number of genes in this intersecting set; the filled-in cells denote which p53 (WT or mutants) is a part of this intersection. Source data are available online for this figure.

The seven cell lines were screened in duplicates at a minimum, and we observed a good reproducibility between sgRNA abundance in the replicates (r = 0.44–0.71) (Hart et al, 2015) (Appendix Fig. 1C).

For gene-level depletion/enrichment, we calculated a normalized z-score (NormZ) of the low and high p53-mClover populations, combining multiple sgRNAs per gene. Negative NormZ scores represent genes whose inactivation leads to decreased p53 levels (i.e., positive regulators of p53 stability), whereas positive NormZ scores represent genes whose inactivation leads to increased p53 levels (i.e., negative regulators of p53 stability). As expected, sgRNAs targeting p53 and mClover were the most depleted sgRNAs, while sgRNAs targeting MDM2 or MDM4 were among the most enriched sgRNAs in the p53-high populations. Other known positive regulators of p53, such as ATM, USP28, TTI1/2, TP53BP1, or CHEK2, and negative regulators of p53 stability such as USP7, HUWE1, or PPM1D/G also scored in several p53 wild-type or mutant screens, further validating the screens (Fig. 1E,F; Appendix Fig. 2A–D and Dataset EV1). Unsupervised clustering of the screens showed that p53 wild-type, G245S, R248Q, R273H mutants clustered closely together, while p53 R175H and p53 R337H showed distinct profiles (Fig. 1G).

To identify hits, we selected genes with NormZ values +/− 3, a false discovery rate (FDR) lower than 0.5 and excluded genes that also affected the negative control mClover-P2A-mRFP reporter. These cutoffs identified 292 and 548 genes, whose loss led to decreased and increased p53 levels, respectively. While most genes regulated the stability of wild-type and several p53 mutants, we also identified p53 wild-type- and p53 mutant-specific regulators (Fig. 1H). For example, wild-type p53 levels were specifically sensitive to perturbation of the 20S and 19S proteasomal subunits, consistent with the high protein turn-over of p53. This is in contrast to the levels of some mutant p53 proteins, such as R273H and R248Q, whose levels were not significantly affected upon genetic ablation of proteasome subunits (Appendix Fig. 2E). In addition, of the 292 hits whose loss caused p53 destabilization, the vast majority (182 genes) regulated only p53 R337H. Conversely, of the 548 hits whose loss led to p53 stabilization, only a small subset (52 genes) regulated p53 R337H (Fig. 1H; Appendix Fig. 1B). These data suggest that p53 R337H is controlled by mechanisms that are different from those modulating wild-type p53 and all other p53 mutants tested.

## FBXO42-CCDC6 axis regulates wild-type and mutant p53 stability

One of the strongest hits whose loss led to decreased p53 levels (i.e., represents a positive p53 regulator) in the R273H and R248Q screens was *FBXO42* (NormZ value of −4.05 and −5.65 for R273H and R248Q, respectively), coding for F-Box Protein 42, which functions as a substrate-recognition subunit of an SCF (SKP1-CUL1-F-box protein)-type E3 ubiquitin ligase complex. Analysis of the Dependency Map (DepMap) project (Tsherniak et al, 2017) indicated that the genetic dependency profile of *FBXO42* correlated with that of p53 and its activators *CHEK2, TP53BP1, ATM*, and *USP28* and was inversely correlated with those of negative p53 regulators such as *MDM2, MDM4, PPM1G*, and *USP7* (Fig. 2A,B), strongly suggesting a functional connection to the p53 pathway. The strongest genetic co-dependency of *FBXO42*, and one of the strongest correlations across all genes and cell lines, was with *CCDC6*, which encodes a coiled-coil domain-containing protein (Fig. 2A,B). CCDC6 shows a similarly strong co-dependency with the p53 pathway in DepMap and loss of *CCDC6* caused phenotypes similar to those associated with the loss of *FBXO42* in our screens, i.e., resulting in decreased p53 R273H and R248Q levels (Fig. 2C; Dataset EV1).

We used two independent sgRNAs with good on-target efficacy to corroborate the effect of FBXO42 and CCDC6 loss on p53 stability and used sgRNAs targeting the *AAVS1* locus as control (Appendix Fig. 2F). We observed significantly reduced R273H, R248Q, and R248W p53mClover levels but failed to see significant effects on wild-type, R175H, G245S or R337H p53-mClover levels (Fig. 2D and Appendix Fig. 3A), indicating specificity for some p53 mutants. The difference in p53 levels was even more apparent upon irradiation (Appendix Fig. 3B–D) and in single-cell knock-out clones (Fig. 2E). We also tested whether proteasomal degradation is involved in FBXO42/CCDC6-mediated regulation of mutant p53. Inhibition of the proteasome by MG132 modestly increased mutant p53 levels in FBXO42 and CCDC6 knock-out cells as assessed by WB analysis and flow cytometry (Fig. 2E; Appendix Fig. 3E), which is consistent with our genetic results of ablating proteasome subunits (Appendix Fig. 2E). Interestingly, we found that inhibition of the lysosomal degradation pathway using chloroquine resulted in increased levels of mutant p53 R273H levels especially in CCDC6

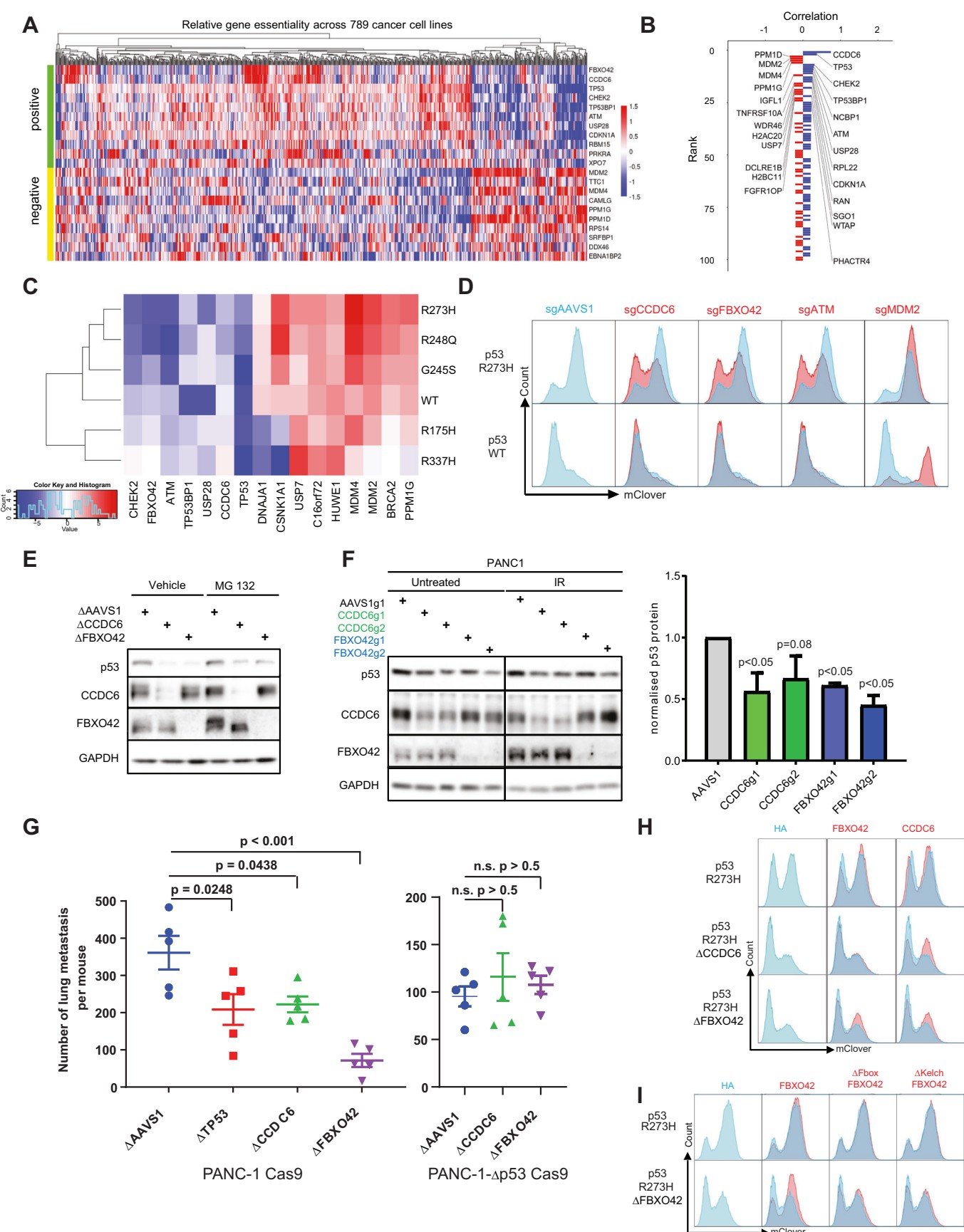

**Figure 2. FBXO42-CCDC6 axis regulates mutant p53 protein stability.**

(A) Heatmap of the essentiality scores of top correlated (positive) and anti-correlated (negative) genes with *FBXO42* across 789 cancer cell lines screened in DepMap (depmap.org, generated using FIREWORK (Amici et al, 2020)). (B) The 50 top genes correlated (blue) and anti-correlated (red) with *FBXO42*, based on coessentiality results from CRISPR screens in 789 cancer cell lines (depmap.org). (C) Heatmap displaying the screening results (as normZ scores) of selected hits across wild-type and five p53 mutants. A positive normZ (red) indicates that genetic ablation of a gene leads to increased p53 protein stability, and negative normZ (blue) indicates decreased p53 stability. (D) Flow cytometry blots depicting wild-type and R273H p53-mClover levels upon CRISPR/Cas9-mediated depletion of indicated genes. Independent sgRNAs different from the sgRNAs in the screening library were used (red) and the effects were compared against control guides targeting the *AAVS1* safe harbor (blue). Results are reproducible over biological triplicates. (E) Representative Western Blot showing p53$^{R273H}$-mClover protein levels in clonal RPE1 p53 R273H reporter cell line upon clonal depletion of *CCDC6* or *FBXO42*, and upon proteasomal inhibition (10 µM MG132 for 12 h). GAPDH serves as a loading control. Results are reproducible over biological triplicates. (F) Representative Western Blot showing endogenous p53 R273H protein levels in PANC-1 cells upon depletion of *CCDC6* and *FBXO42*, and upon genotoxic stress (IR 0.5 Gy, 24 h). GAPDH serves as a loading control. The bar graph depicts the quantification of p53 levels over three independent biological replicates (*$p < 0.05$, two-tailed t-test). Error bar = standard error of the mean (S.E.M.). Results are reproducible over biological triplicates. (G) Quantification of the metastatic lung colonization. The number of lung foci for each mouse injected with PANC1-Cas9 cells with the indicated genotype was plotted ($n = 5$ mice for each condition, with a mix of males and females housed in different cages). Mantel-Cox (log-ranked) test was used for statistical analysis. Error bar = S.E.M. (H) Flow cytometry blots depicting the p53$^{R273H}$-mClover levels upon depletion of *CCDC6* or *FBXO42*, and upon ectopic re-expression of *CCDC6* or *FBXO42*. Results are reproducible over biological triplicates. (I) Flow cytometry blots depicting p53$^{R273H}$-mClover levels upon depletion of *FBXO42* and ectopic re-expression of ΔFbox FBXO42 (lacking aa 44–93) or ΔKelch FBXO42 (lacking aa 132–432). Results are reproducible over biological triplicates. Source data are available online for this figure.

and FBXO42 KO cells (Appendix Fig. 3F), which is in line with previous data showing that mutant but not wild-type p53 is degraded via chaperone-mediated autophagy in a lysosome-dependent fashion (Vakifahmetoglu-Norberg et al, 2013). Of note, genetic ablation of *FBXO42* or *CCDC6* did not alter cell cycle progression, ruling out potential indirect effect on p53 levels through the cell cycle (Reyes et al, 2018) (Appendix Fig. 3G). Importantly, genetic ablation of *FBXO42* also reduced the levels of p53 in the pancreatic cancer cell lines PANC-1 and Mia PaCa-2, which are homozygous for p53 R273H and p53 R248W (Redston et al, 1994), respectively (Fig. 2F; Appendix Fig. 3H), without affecting *TP53* mRNA expression, confirming that FBXO42 acts on p53 via posttranscriptional regulation (Appendix Fig. 3I).

Next, we set out to assess the functional consequences of reduced mutant p53 R273H levels upon knockout of *FBXO42* or *CCDC6*. Since accumulated mutant p53 is required for many gain-of-function properties, reducing its level may lead to suppression of tumor growth and attenuation of invasion and metastasis formation (Alexandrova et al, 2015; Freed-Pastor and Prives, 2012; Oren and Rotter, 2010). We thus injected PANC-1 cells that were depleted of either *FBXO42* or *CCDC6* as well as control (sg*AAVS1*) PANC-1 cells into the tail vein of nonobese diabetic/severe combined immunodeficiency-gamma (NSG) mice and evaluated lung metastasis colonization. Genetic ablation of *FBXO42* and *CCDC6* resulted in a significant reduction of metastatic colonization relative to the control, similar to the depletion of *TP53* (R273H) (Fig. 2G and Suppl Fig. 4A). Of note, loss of *FBXO42* showed the most dramatic reduction, which was likely due to a stronger knock-out efficacy compared to depletion of *CCDC6* or *TP53* (Fig. 3I). Importantly, to test whether these effects were specifically mediated by mutant p53 R273H, we repeated this experiment using PANC-1 cells devoid of endogenous p53 R273H. Genetic ablation of FBXO42 or CCDC6 no longer affected the metastatic colonization of PANC-1-Δp53 cells (Fig. 2G and Suppl Fig. 4A). Collectively, these results suggested that the loss of FBXO42 and CCDC6 destabilize mutant p53 and attenuate mutant p53-driven metastatic colonization of mouse lungs.

Given the strong genetic correlation inferred from the DepMap dataset, we next set to test for a potential epistatic relationship between FBXO42 and CCDC6, using isogenic *FBXO42*-knockout (Δ*FBXO42*), *CCDC6*-knockout (Δ*CCDC6*) and *AAVS1*-targeted control RPE-1 p53$^{R273H}$-mClover reporter cell lines (Fig. 2E). As

expected, p53$^{R273H}$-mClover levels in Δ*FBXO42* and Δ*CCDC6* could be partially rescued by re-expressing FBXO42 or CCDC6, respectively. Interestingly, while expression of FBXO42 did not rescue p53$^{R273H}$-mClover levels in Δ*CCDC6* cells, we found that expression of CCDC6 partly rescued mutant p53 levels in Δ*FBXO42* cells (Fig. 2H), suggesting that FBXO42 may function upstream of CCDC6.

Using FBXO42 truncation mutants, we found that both the Kelch and F-Box domains are required to promote p53 R273H stabilization, indicating that FBXO42's ability to stabilize p53 R273H is dependent on both its substrate binding domain and its incorporation into an SCF complex, likely to promote SKP1/CUL1-mediated ubiquitination of an as-yet unidentified substrate (Fig. 2I; Appendix Fig. 4B–D). Together, these data indicate that FBXO42 and CCDC6 function as positive post-translational regulators of wild-type p53 and several p53 mutants in a ubiquitination-dependent manner.

## Mapping of the FBXO42-CCDC6 and mutant p53 interaction network

To investigate how FBXO42-CCDC6 regulates p53 stability, we identified vicinal proteins by proximity-dependent biotinylation coupled to mass spectrometry (BioID) (Kim et al, 2016; Lambert et al, 2015; Roux et al, 2013; Roux et al, 2012) using inducible expression of biotin ligase (BirA*)-tagged FBXO42, CCDC6 or p53 R273H in HEK293 Flp-In T-REx cells. Given the function of FBXO42 in the ubiquitin-proteasomal system, we performed these BioID experiments in the absence or presence of MG132. Consistent with its role as an SCF-E3 ligase, the top interactors for FBXO42 were CUL1 and SKP1. FBXO42 BioID also enriched for CCDC6, p53 itself, and proteins known to regulate p53, such as HUWE1 (Fig. 3A; Dataset EV2). p53 R273H proximal proteins included known p53 interactors such as MDM2, BRCA2, USP28, TP53BP1, PPM1G, BLM, ATRX, and PML (Fig. 3A). In addition, endogenous CCDC6 as well as V5-tagged CCDC6 co-immunoprecipitated (co-IP) with endogenous p53 R273H in PANC-1 cells (Fig. 3B; Appendix Fig. 4E). In further support of this interaction, proximity ligation assay (PLA) showed that CCDC6 and p53 R273H are proximal predominantly in the nucleus (Fig. 3C). Using in vitro binding assays of recombinant proteins, we detected a direct interaction between FBXO42 Kelch

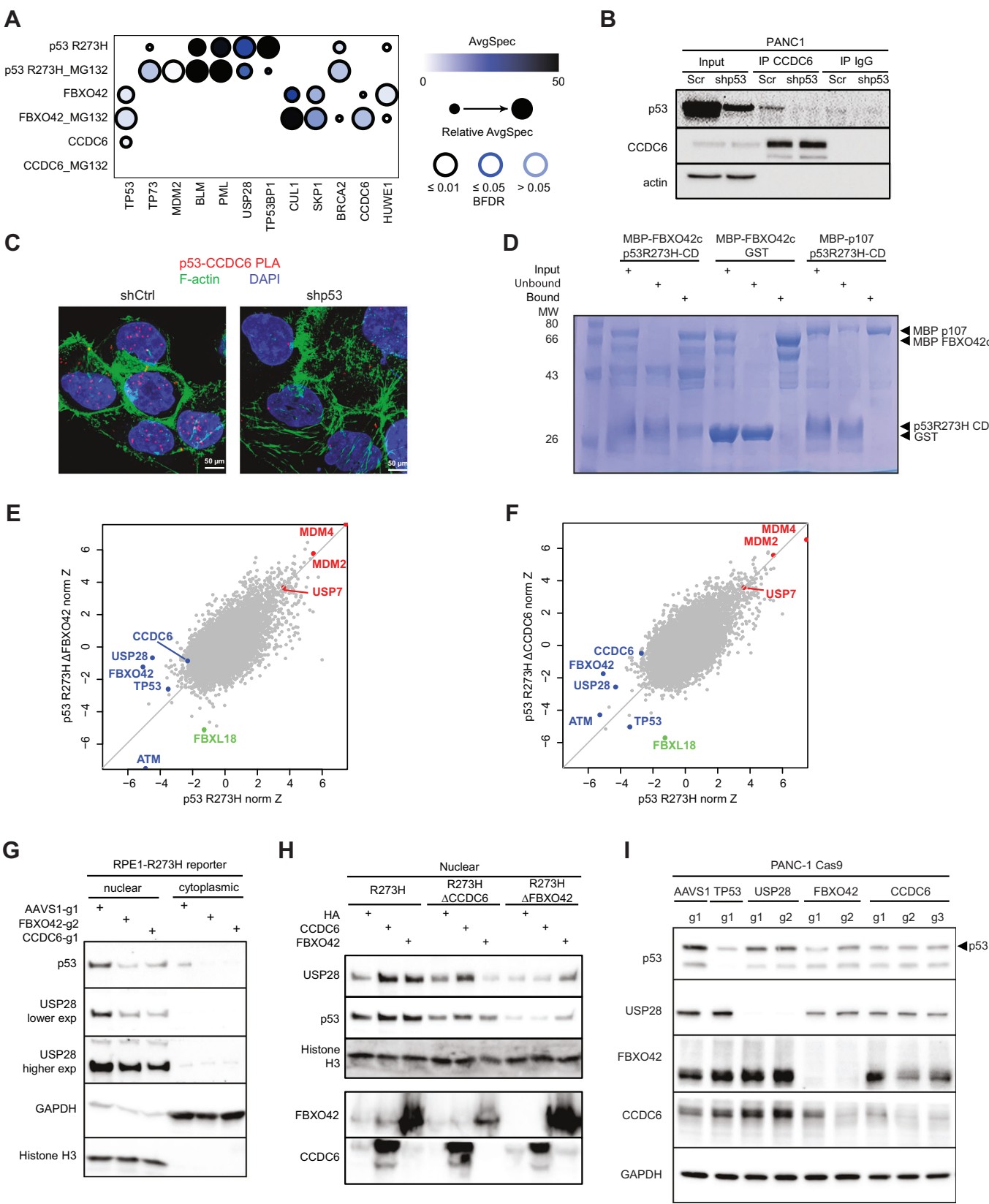

**Figure 3. Mapping the genetic interaction network of FBXO42-CCDC6 and mutant p53.**

(A) Selected proximity interactors of p53 R273H, CCDC6, and FBXO42 as mapped by BioID using HEK293-Flp-In T-REx cell lines stably expressing each bait, with or without the proteasomal inhibitor MG 132 [5 μM, 24 h]. The intensity of the shade filling depicts the spectral count of each prey, the relative abundance of this prey compared across all baits is indicated by the circle size, and the confidence (Bayesian false discovery rate, BFDR) is by the intensity of the edge. (B) Interaction of p53 R273H and CCDC6 in PANC-1 cells validated using immunoprecipitation (IP). Lysates of PANC-1 cells with or without depletion of the endogenous p53 R273H protein were immunoprecipitated using an CCDC6-specific antibody or an IgG-isotype control, followed by Western blot analysis of the endogenous p53. β-actin serves as a loading control for lysate input. Results are reproducible over biological triplicates. (C) Proximity ligation assay (PLA) between endogenous CCDC6 and endogenous p53 R273H in PANC-1 cells using tetramethylrhodamine-5-isothiocyanate (TRITC) as a probe (red) and counterstained with DAPI (blue) and phalloidin-FITC (green) to visualize nuclear DNA and F-actin, respectively. RNAi-mediated depletion of the endogenous p53 R273H protein was used as a control to show specificity. Results are reproducible over biological triplicates. (D) In vitro evidence for a direct and specific interaction between the p53-R273H core domain (p53CD-R273H) and FBXO42c. The MBP-tagged Kelch domain of FBXO42 (MBP-FBXO42c, aa 105–360) and the core DNA-binding domain of p53 R273H (p53-CD-R273H, aa 90–311) were recombinantly expressed and purified from the BI21DE3 *E. coli* strain. MBP-FBXO42c was pre-coupled to amylose resin. Following incubation of MBP-FBXO42c and p53-CD-R273H, amylose-resin coupled MBP-FBXO42c captured a fraction of p53-CD-R273H, which was found partly in the bound fraction. As a first specificity control, Amylose-coupled MBP-FBXO42c was unable to capture the unrelated protein GST in the bound fraction. A second specificity control was performed to show that p53-CD-R273H did not bind to the unrelated protein MBP-p107. In this control, no p53-CD-R273H was found in the bound fraction. The input, unbound, and bound fractions were resolved on SDS-PAGE and stained with Coomassie blue. Results are reproducible over biological triplicates. (E, F) Scatter plots of the perturbation effect of each gene as normalized Z-score (normZ), in the p53 R273H reporters before and after the loss of *CCDC6* or *FBXO42*. Selected genes whose depletion resulted in p53 stabilization and destabilization are marked in red and green, respectively. Screens were performed in technical duplicates. (G, H) Representative Western blot results showing levels of USP28 and p53 in nuclear and cytoplasmic fractions of the p53 R273H reporter cell line with clonal depletion of FBXO42 or CCDC6, and upon the ectopic re-expression of CCDC6 or FBXO42. Histone H3 and GAPDH served as nuclear and cytoplasmic markers, respectively. Levels of CCDC6 and FBXO42 were measured from the total cell lysates (H). Results are reproducible over biological triplicates. (I) Total cellular USP28 and p53 levels in PANC-1 cells upon depletions of *CCDC6, FBXO42*, or *USP28*. Cell lines harboring *AAVS1-g1, TP53-g1, FBXO42-g1*, and *CCDC6-g1* were further used in metastatic lung colonization experiment via tail vein injections (Fig. 2G; Appendix Fig. 4A). Results are reproducible over biological triplicates. Source data are available online for this figure.

domains 1–3 and the core DNA-binding domain of p53 R273H (Fig. 3D, Appendix Fig. 4C,F,G). This interaction was specific as the p53 R273H core domain did not bind p107 Rb, an unrelated protein of similar size (Fig. 3D; Appendix Fig. 4G). In addition, while wild-type p53 interacted with MDM2, the p53 R273H core domain did not, as expected given that MDM2 interacts with the N-terminal activation domain of p53 (Appendix Fig. 4H). Together, these data show that R273H p53 interacts with both its regulators CCDC6 and FBXO42, with a direct interaction demonstrated for FBXO42, and indicates a potential formation of a higher order complex that regulates p53 stability.

## FBXO42-CCDC6 regulates p53 via USP28

To identify genetic determinants of FBXO42/CCDC6-mediated p53 R273H stabilization, we performed genome-wide CRISPR screens using isogenic Δ*FBXO42*, Δ*CCDC6* or control RPE1 p53^R273H^-mClover-P2A-RFP reporter cell lines. This allowed us to systematically map genetic perturbations that regulate p53 R273H stability depending on the presence of FBXO42 and CCDC6 (Fig. 3E,F). Most genes such as *MDM2, MDM4, USP7, CSNK1A1, ATM*, or *TTI* retained their function in regulating p53 stability in Δ*FBXO42* or Δ*CCDC6* RPE1 cells. As expected for genes in the same pathway and based on our previous data, *CCDC6* did not score in the Δ*FBXO42* screen (nor did *FBXO42)* and *FBXO42* did not score in the Δ*CCDC6* screen (nor did *CCDC6)*. Interestingly, we identified one F-box protein-coding gene, *FBXL18*, which gained the ability to reduce p53^R273H^-mClover levels in Δ*FBXO42* or Δ*CCDC6* RPE1 cells but did not have an effect when lost in parental RPE1 wild-type cells, suggesting that FBXL18 may compensate for the loss of FBXO42/CCDC6 (Fig. 3E,F). In addition, loss of *USP28*, whose perturbation led to the strongest reduction in p53^R273H^-mClover levels in wild-type RPE1, had no effect on p53^R273H^-mClover levels in the Δ*FBXO42* screen, and only a weak effect on p53^R273H^-mClover levels in the Δ*CCDC6* screen, indicating that USP28 may function in the same pathway as FBXO42/CCDC6 (Fig. 3E,F). We corroborated these genetic interactions and showed that

concomitant loss of *USP28* in Δ*FBXO42* or Δ*CCDC6* RPE1 cells has no effect on p53^R273H^-mClover levels (Appendix Fig. 5A). Interestingly, loss of *FBXO42* or *CCDC6* resulted in decreased levels of nuclear USP28 as shown by nuclear/cytoplasmic fractionation as well as immunofluorescence (Fig. 3G; Appendix Fig. 5B). Importantly, ectopic expression of either FBXO42 or CCDC6 not only rescued the USP28 and R273H p53 levels in Δ*FBXO42* or Δ*CCDC6* RPE1 cells but also led to increased USP28 levels in the parental RPE1 cells, concomitant with a slight upregulation of p53 levels (Fig. 3H). Genetic ablation of *FBXO42* or *CCDC6* in PANC-1 cells also reduced USP28 protein abundance, without affecting USP28 transcript levels (Fig. 3I; Appendix Fig. 5C), indicating that the post-translational regulation of USP28 by FBXO42/CCDC6 is not unique to RPE1 cells. Next, we generated Δ*USP28* p53^R273H^-mClover RPE1 cells and assessed how overexpression of FBXO42 and CCDC6 impacted p53 R273H levels (Appendix Fig. 5D). While expressing USP28 rescued p53 R273H levels, overexpressing FBXO42 or CCDC6 in Δ*USP28* cells had no effect, indicating that FBXO42/CCDC6 act via USP28. Moreover, we found that USP28 overexpression rescued mutant p53 levels in Δ*FBXO42* or Δ*CCDC6* RPE1 cells, indicating that USP28 is downstream of FBXO42/CCDC6 (Appendix Fig. 5D). Together, these data indicate that FBXO42/CCDC6 control mutant p53 levels via USP28.

## Synthetic viability screen maps positive regulators of p53

To better understand the genes whose loss leads to p53 stabilization (i.e., that act as p53 negative regulators), we first performed pathway analysis using gProfiler (Fig. 4A; Dataset EV3). Gene sets associated with cellular response to stress, extracellular stimuli or hypoxia, cell cycle, mitosis, stabilization of p53 and p53-dependent and -independent DNA Damage Responses were significantly enriched in the 547 hits whose inactivation result in higher p53 levels. Enrichment was also found for genes involved in regulation of nonsense-mediated decay and programmed cell death. Interestingly, metabolism of RNA and rRNA modification in the nucleus and cytosol were the most significantly enriched categories. This

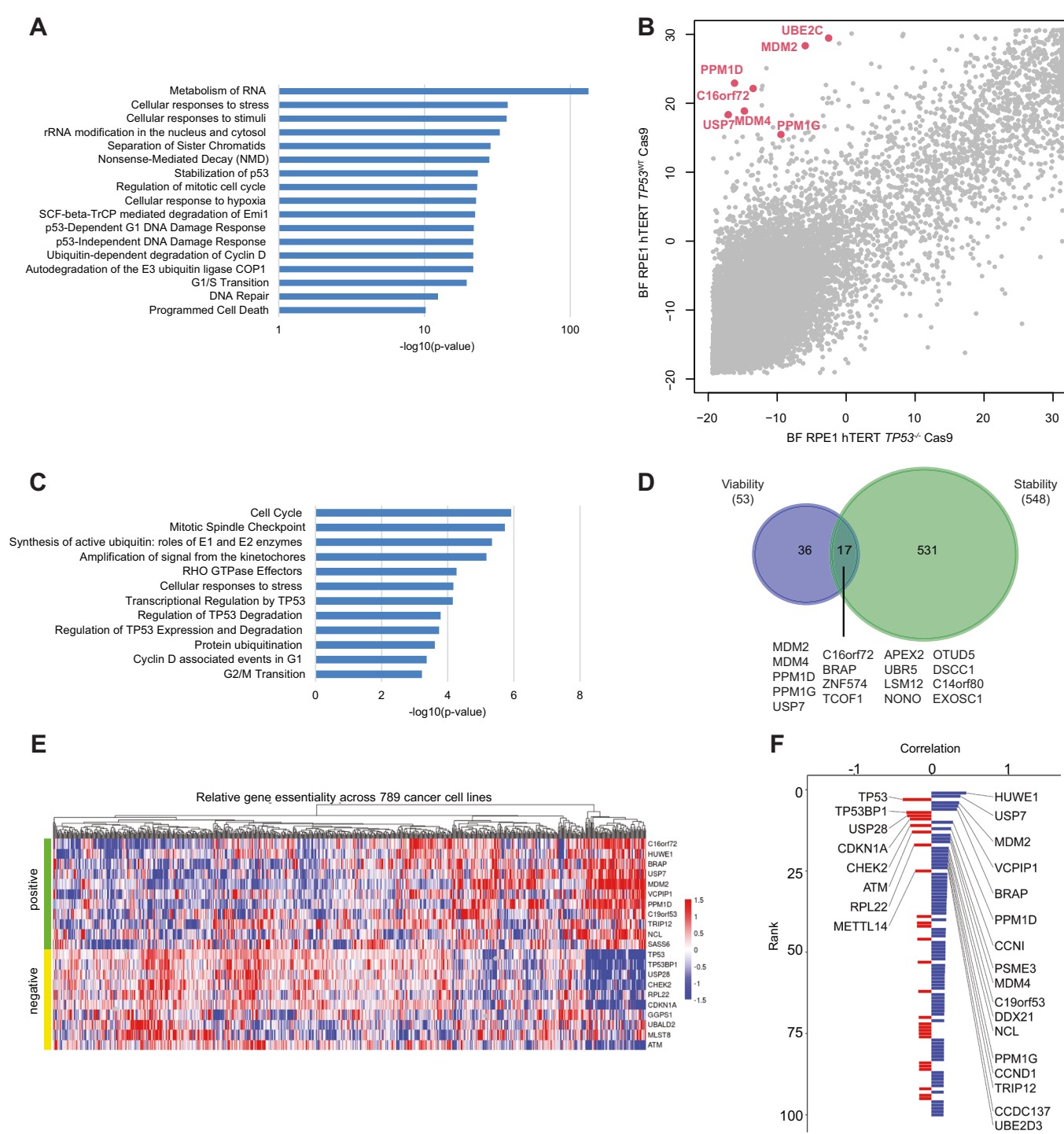

functional group was also enriched in various other screens done to identify genes that cause genomic instability or modulate responses to ionizing or UV radiation, or screens for ATM/ATR substrates (Hurov et al, 2010; Matsuoka et al, 1998; Paulsen et al, 2009; Stokes et al, 2007), indicating that RNA metabolism is tightly interconnected with DNA damage responses. Together, this data indicates that the stability of mutant p53 is, to a large extent, regulated by the physiological machinery that regulates stress-induced wild-type p53 stability. However, this data may also indicate that most of the hits whose loss leads to p53 stabilization might regulate p53 indirectly by causing cellular stress.

As an alternative approach to identify regulators of p53, we sought to exploit the concept of synthetic viability, which describes a genetic interaction where the viability of one genetic mutation is determined by the presence of a second genetic mutation (O'Neil et al, 2017). The mutations in *MDM2* and *TP53* display a synthetic viability interaction, as the embryonic and cellular lethality associated with the loss of MDM2 is completely rescued in a p53

**Figure 4. Synthetic viability screen maps regulators of p53.**

(A) Pathway analysis based on the top scoring genes from the protein stability reporter screens that resulted in increased wild-type or mutant p53 levels using Reactome pathway analysis. Selected Reactome pathways are shown. Fisher's exact test based on the hypergeometric distribution was used for pathway enrichment analysis. (B) Synthetic viability screen in RPE1 cells. Bayesian Factors (BF) as a measurement of essentiality (high values indicate a lethal gene) are shown for all protein-coding genes in p53 wild-type (y-axis) versus p53 null (x-axis) background. All BFs were computed using the BAGEL2 algorithm (Kim and Hart, 2021). Screens were performed in technical triplicates. (C) Pathway analysis based on the top-scoring genes in the synthetic viability screen using Reactome pathway analysis. Selected Reactome pathways are shown. Fisher's exact test based on the hypergeometric distribution was used for pathway enrichment analysis. (D) Venn diagram depicting the top scoring genes from the synthetic viability screen and the top scoring genes from the p53 stability screens whose mutation leads to increased p53 levels. The common genes from both screens are depicted. (E) Heatmap of the essentiality scores of top correlated (positive) and anti-correlated (negative) genes with *C16orf72/HAPSTR1* across 789 cancer cell lines screened in DepMap (depmap.org, generated using FIREWORK (Amici et al, 2020). (F) The 50 top genes correlated (blue) and anti-correlated (red) with *C16orf72/HAPSTR1*, based on coessentiality results from CRISPR screens in 789 cancer cell lines (depmap.org). Source data are available online for this figure.

null background (Montes de Oca Luna et al, 1995). To identify genes that are synthetic viable with p53, we performed genome-wide CRISPR screens in isogenic p53-proficient and -deficient RPE1 cell lines. Synthetic viable interactions were defined as genes that had high Bayes factor (BF) in p53-proficient cells, indicating essentiality, but negative BF values in the p53-deficient cells (Dataset EV4). As expected, the known negative p53 regulators MDM2, MDM4, USP7, PPM1D (encoding WIP1) all scored highly as synthetic viability hits in the screen. In addition, pathway analysis of the top 150 scoring genes using PANTHER (Mi et al, 2017) revealed enrichment of genes involved in the p53 pathway and the related Ubiquitin proteasome pathway (Fig. 4C; Dataset EV5).

To delineate high-confidence p53 regulators, we searched for hits that scored in both the p53 synthetic viability and the p53 reporter screens. 17 out of the 53 top-scoring genes in the synthetic viability screen also scored in the p53 reporter screens. In addition to the well-known p53 regulators (*MDM2/4, USP7, PPM1D/G*), *ZNF574, APEX2, BRAP, NONO, TCOF1, OTUD5, UBR5, LSM12, DSCC1, C14orf80, EXOSC1*, and *C16orf72/HAPSTR1* scored prominently in both screening formats (Fig. 4D; Dataset EV1 and 4). In addition to the RPE1 cell lines, we also performed p53 synthetic viability screens in two other human cell lines: A549, a lung cancer cell line, and RKO, a colon cancer cell line. While the synthetic viability hits of from the screen done in A549 cells exhibited similarity to those found in RPE cells, the synthetic viability hits from the screen done in RKO cells were distinct (Appendix Fig. 6A; Dataset EV4).

*C16orf72/HAPSTR1* (also known as *TAPR1*) was previously identified in a screen for genes that altered sensitivity to telomere attrition and was shown to buffer against p53 activation in response to telomere erosion (Benslimane et al, 2021). Moreover, we recently identified *C16orf72/HAPSTR1* in a screen that analyzed genetic vulnerabilities to ATR inhibition (Hustedt et al, 2019). In addition, analysis of coessentiality across 789 cancer cell lines from the DepMap project showed a striking association between *C16orf72/HAPSTR1* and several positive and negative p53 regulators (Fig. 4E,F) (Amici et al, 2022; Benslimane et al, 2021). Together, these data identified C16orf72/HAPSTR1 as a candidate negative regulator of p53 stability.

## C16orf72/HAPSTR1 is a regulator of wild-type and mutant p53 stability

To validate the genetic interaction between *TP53* and *C16orf72/HAPSTR1*, we performed clonogenic survival assays. Loss of

*C16orf72/HAPSTR1* resulted in decreased cell viability and relative cellular fitness selectively in the *TP53*$^{+/+}$ background but had no impact in the p53-deficient isogenic counterpart, indicating synthetic viability (Fig. 5A). Loss of *C16orf72/HAPSTR1* also caused an increase in p53 levels for wild-type as well as all tested p53 mutants (Fig. 5B,C; Appendix Fig. 6B).

To gain further insight into the functions of C16orf72/HAPSTR1, we performed cycloheximide-chase experiments together with Nutlin treatments, which showed that C16orf72 regulates p53 in an MDM2 independent manner (Appendix Fig. 6C). In line with this observation, overexpression of C16orf72 in cells depleted of MDM2 could not rescue R273H p53 levels, again indicating that MDM2 and C16orf72 function independently (Appendix Fig. 6D). To identify how C16orf72/HAPSTR1 might regulate p53 levels, we next performed affinity purification coupled to mass spectrometry (AP-MS) on FLAG-tagged C16orf72/HAPSTR1. This identified HUWE1 as a prominent interactor of C16orf72/HAPSTR1 (Fig. 5D; Dataset EV6), in line with previous findings that found that C16orf72 is required for the nuclear localization of HUWE1 (Benslimane et al, 2021; Monda et al, 2023). HUWE1, a known E3 ubiquitin ligase for p53, scored as a hit in our marker-based p53 stability screens and showed the strongest coessentiality in DepMap (Figs. 2C and 5E). siRNA-mediated knock-down of HUWE1 led to a slight increase in p53 levels compared to the loss of other known negative p53 regulators such as MDM2 or USP7 (Fig. 5E). Given that HUWE1 is a common essential gene (Hart et al, 2015) and HUWE1 knock-down results in rapid apoptosis of p53 wild-type RPE1 cells, these experiments are difficult and hard to interpret. We thus turned our focus to p53 null RPE1 cells expressing the mutant p53 R273H reporter and performed rescue experiments using wild-type C16orf72 as well as a truncation C16orf72 mutant lacking the nuclear localization signal (NLS). These experiments showed that the nuclear localization sequence of C16orf72 is important for the regulation of p53 (Fig. 5F). We further showed that the ability of C16orf72 to regulate p53 is dependent on HUWE1 (Fig. 5G).

The second most significant co-essential gene of *C16orf72/HAPSTR1* in DepMap was *USP7* (Fig. 4E,F), which also scored as a synthetic viable p53 interaction in RPE1 cells, as well as scoring as a strong p53 stability regulator in the reported-based CRISPR screens (Figs. 2C and 4B). To test for a functional relationship between C16orf72/HAPSTR1 and USP7, we performed co-immunoprecipitation experiments and found that FLAG-tagged full-length USP7 interacts with HA-tagged C16orf72/HAPSTR1 (Fig. 5H). Together, these data indicates that C16orf72/HAPSTR1 might regulate p53 via USP7.

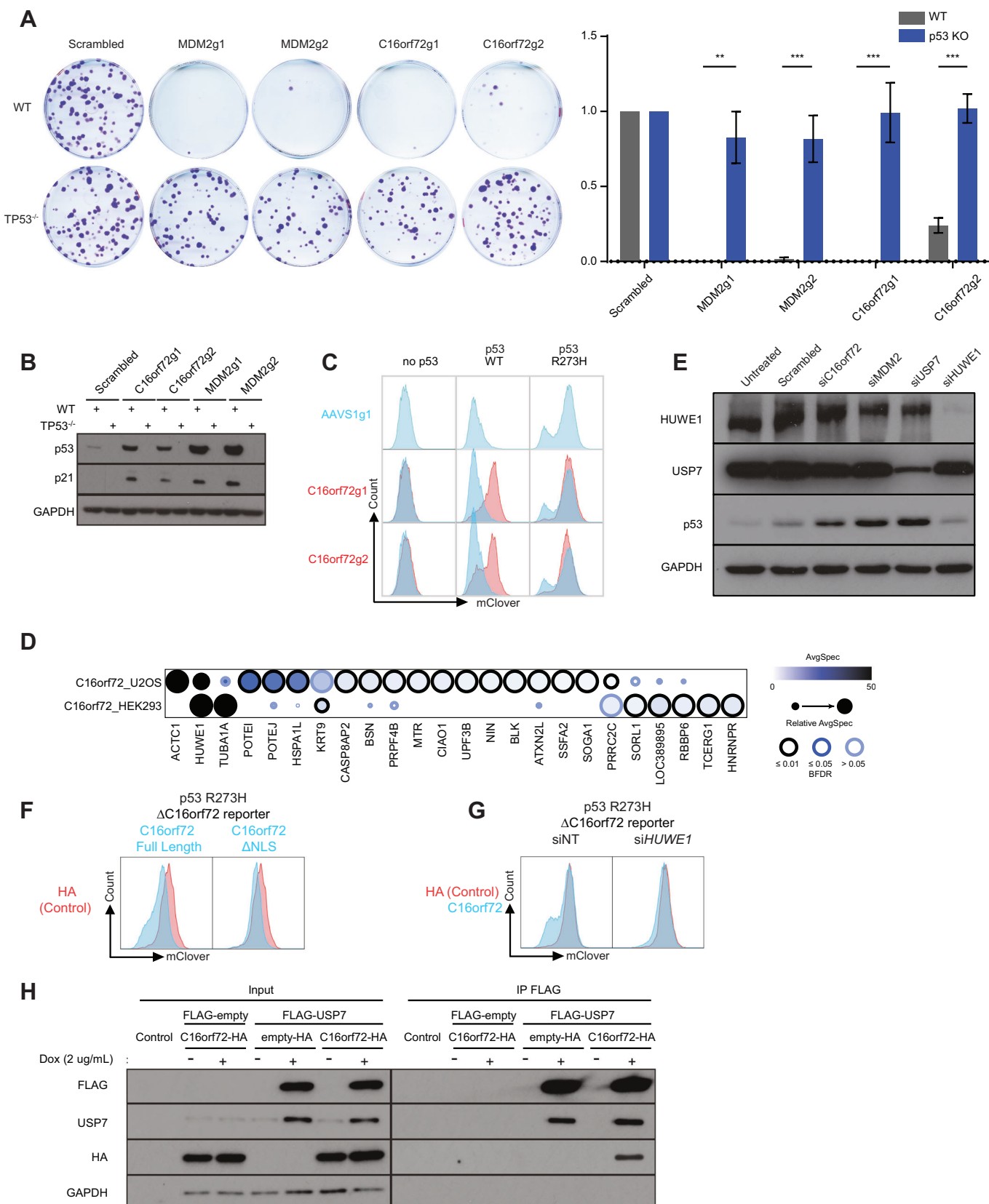

◀

**Figure 5. C16orf72/HAPSTR1 is a regulator of wild-type and mutant p53 stability.**

(A) Clonogenic survival assays validating the synthetic viability between C16orf72/HAPSTR1 and p53. Assayed 13 days after plating, the surviving fractions of RPE1-hTERT-$TP53^{+/+}$ or $TP53^{-/-}$ cells transduced with the indicated sgRNAs were normalized to those depleted with the non-targeting control guide. The two-tailed unpaired t-test was used for statistical analysis. Error bar = standard error of the mean (S.E.M.), $n = 3$, **$p < 0.01$, ***$p < 0.001$. Results are reproducible over biological triplicates. (B) Representative Western Blot results showing p53 and p21 protein levels in RPE1 cells transduced with the indicated sgRNAs. GAPDH serves as a loading control. Results are reproducible over biological triplicates. (C) Flow cytometry blots depicting the level of wild-type or p53$^{R273H}$-mClover protein levels upon depletion of C16orf72/HAPSTR1. Results are reproducible over biological triplicates. (D) Interactors of C16orf72/HAPSTR1 as mapped by AP-MS in HEK293-Flp-In T-REx and U2OS-Flp-In T-REx cells stably expressing FLAG-tagged C16orf72/HAPSTR1. Mass spectrometry was performed in biological triplicates. (E) Representative Western Blot results showing p53 protein levels in parental RPE1 cells in response to depletions of C16orf72/HAPSTR1 and other known E3 ligases of p53 (MDM2, USP7, and HUWE1) using siRNAs. Results are reproducible over biological triplicates. (F) Flow cytometry blots depicting the level of p53$^{R273H}$-mClover protein levels in a clonal p53$^{R273H}$ reporter RPE1 cell line depleted of C16orf72 and transduced with either full-length C16orf72 or a C16orf72 truncation mutant lacking the NLS or a control overexpression construct (HA). Results are reproducible over biological triplicates. (G) Flow cytometry blots depicting the level of p53$^{R273H}$-mClover protein levels in a clonal p53$^{R273H}$ reporter RPE1 cell line depleted of C16orf72 and transduced with either full-length C16orf72 or a control overexpression construct (HA) with concomitant loss of either HUWE1 by transient siRNA knockdown (siHUWE1) or the corresponding non-targeting control (siNT) control. Results are reproducible over biological triplicates. (H) Co-immunoprecipitation (co-IP) showing an interaction of C16orf72/HAPSTR1 and USP7. HEK 293 cells stably transduced with an inducible FLAG-USP7 or FLAG-empty vector control vector were transfected with either an HA-tagged C16orf72/HAPSTR1 or an HA-empty control vector. Lysates with or without doxycycline-induction were co-immunoprecipitated using a FLAG-specific antibody, followed by Western Blot analysis of HA. GAPDH serves as a loading control for the lysate input. Results are reproducible over biological triplicates. Source data are available online for this figure.

## C16orf72/HAPSTR1 functions as an oncogene and regulates p53 stability in the mammary gland

The USP7 gene lies directly adjacent to C16orf72/HAPSTR1 and both genes are commonly co-amplified in up to 7.6% of cancers as well as being co-gained in up to 53% of cases ($p < 0.001$; Appendix Fig. 7A,B). Invasive breast cancer showed the highest level of amplification and gains of the USP7/C16orf72/HAPSTR1 locus in up to 55% of tumors ($p < 0.001$; Fig. 6A), suggesting that these two co-amplified negative p53 regulators might cooperate in modulating p53 levels in cancer.

To test the role of C16orf72/HAPSTR1 in vivo, we first conducted multicolor competition assays in the mammary glands of LSL-Cas9-GFP mice. Intraductal injection of lentiviral particles expressing Cre, RFP, and a sgRNA cassette targeting Mdm2 or C16orf72/HAPSTR1 led to a drastic reduction of mammary epithelial cells of LSL-Cas9-GFP mice when compared to a control lentivirus expressing Cre, BFP and an sgRNA targeting the inert Tigre locus. The reduced cell viability was dependent on p53, as conditional p53 knock-out mice (Trp53$^{fl/fl}$; LSL-Cas9-GFP) did not show this phenotype (Fig. 6B).

Conversely, overexpression of C16orf72/HAPSTR1 led to a significant reduction in p53 R273H levels in RPE1 cells (Fig. 6C) and levels of wild-type p53 expression in human Pik3ca$^{H1047R}$-mutant mammary epithelial MCF10A cells (Fig. 6D). This led us to ask whether overexpression of C16orf72/HAPSTR1 can decrease the latency of Pik3ca$^{H1047R}$-driven mammary tumors. Of note, we observed that loss of p53 cooperates with Pik3ca$^{H1047R}$ to accelerate mammary tumor initiation (Appendix Fig. 7C) (Adams et al, 2011; Langille E, 2022). Similar to loss of p53 or overexpression of Mdm2, intraductal injection of lentiviral particles expressing Cre and C16orf72/HAPSTR1 or USP7 significantly reduced the latency of Pik3ca$^{H1047R}$-driven mammary tumor development, which was strictly dependent on the presence of p53 within those mice (Fig. 6E). Importantly, C16orf72/HAPSTR1 overexpressing tumors and hyperplastic mammary epithelium showed a drastic reduction in p53 levels compared to Ruby control lesions (Fig. 6F). Together, these data show that C16orf72/HAPSTR1 overexpression leads to decreased p53 protein levels in vitro and in vivo and leads to accelerated tumor formation.

## Discussion

Over 22 million cancer patients today carry TP53 mutations, the majority of which are missense mutations, often resulting in elevated expression of mutant p53 proteins (Petitjean et al, 2007). Lowering mutant p53 expression can reduce tumor growth and metastasis and trigger tumor regression (Alexandrova et al, 2015; Bossi et al, 2006; Hui et al, 2006), suggesting that tumors become addicted to mutant p53. Targeting factors that regulate mutant p53 stability or reactivate wild-type p53 function might therefore constitute a viable therapeutic strategy.

We conducted genome-wide CRISPR screens in isogenic RPE1 cells expressing protein stability reporters for wild-type p53 and five of the most common p53 mutants. Unsupervised hierarchical clustering showed that most hotspot mutant p53 proteins (R175H, G245S, R248Q, R273H) are regulated by a common molecular network, which is, to a certain degree, distinct from wild-type p53. Interestingly, the Brazilian founder mutation R337H, which is located in the oligomerization domain and is the only mutation in our study that resides outside of the DNA-binding domain, behaved drastically differently to the other hotspot mutations and wild-type p53. As such, it will be interesting to further elucidate the molecular underpinnings of why some mutations behave like wild-type p53, while others do not; furthermore, mutant-specific regulators such as the transfer RNA (tRNA) synthetases genes, whose loss specifically leads to the destabilization of the R337H p53 mutant, are worth following up. Together, these data provide a comprehensive map of genes that regulate wild type and mutant p53 protein stability and might have implications for the development of agents that target mutant p53 in cancer therapy.

We next characterized two mutant p53 regulators, FBXO42 and CCDC6, and provided strong genetic evidence that FXBO42, in conjunction with CCDC6, are novel regulators of certain p53 mutants. FBXO42 was previously reported as a negative regulator of wild-type p53 and shown to bind a phosphodegron on p53, leading to its proteasomal degradation in U2OS cells (Sun et al, 2009). The COP9 signalosome-associated kinase was found to mediate the phosphorylation of p53's core DNA-binding domain, which was required to allow FBXO42 binding and ubiquitination

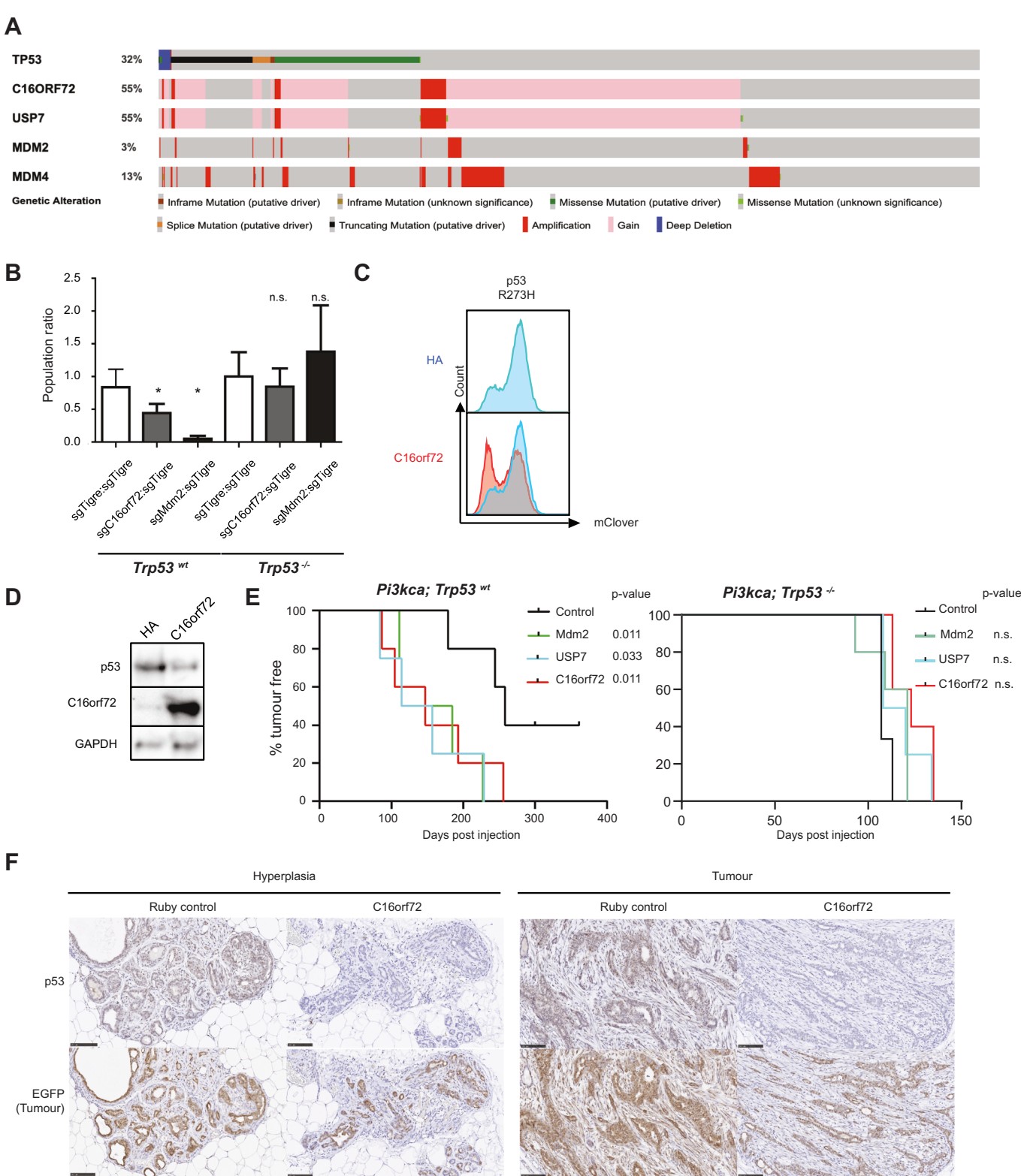

(Sun et al, 2011). We found no significant effect of FBXO42 on wild-type p53 or R175H, G245S or R337H p53, but genetic ablation of FBXO42 caused destabilization of p53 R273H, R248Q, and R248W. This is consistent with the co-dependencies of FBXO42 and CCDC6 in DepMap, which clearly align them with positive p53

regulators such as ATM, CHEK2, TP53BP1 and USP28, while they are anti-correlated with negative p53 regulators such as MDM2/4, TTC1 or PPM1G/D. Interestingly, other previously reported regulators of mutant p53 stability, such as TRRAP, BAG2, or BAG5, did not score as hits in our screen, suggesting that context

◄

**Figure 6.   C16orf72/HAPSTR1 functions as an oncogene and regulates p53 stability in the mammary gland.**

(A) cBioPortal OncoPrint displaying a trend toward mutual exclusivity between genetic ablation of *TP53* and *C16orf72/HAPSTR1* amplification, and co-amplification between *USP7* and *C16orf72/HAPSTR1*, among breast cancer patients. (B) In vivo cell competition assay in the mouse mammary glands. LSL-Cas9-EGFP (*Trp53*$^{+/+}$) or the LSL-Cas9-EGFP; *Trp53*$^{flox/flox}$ (*Trp53*$^{-/-}$) mice were intraductally injected with a mixture of control lentiviral particles expressing Cre and BFP as well as an sgRNA targeting the *Tigre* safe harbor, and experimental lentiviral particles expressing Cre and RFP as well as an sgRNA targeting *Tigre*, *C16orf72/HAPSTR1*, or *Mdm2*. The number of surviving cells that had been depleted of each gene was counted 12 days post injection and normalized to the number of cells depleted of *Tigre* in the same gland. This ratio was further normalized to the ratio of sgTigre:sgTigre in the LSL-Cas9-EGFP; *Trp53*$^{flox/flox}$ mouse. Two two-tailed unpaired t-test was used for statistical analysis. Error bar = standard error of the mean (S.E.M.), $n = 3$ glands, *$p < 0.05$. (C) Flow cytometry plot depicting the p53$^{R273H}$-mClover levels in RPE1 reporter cells upon overexpression of *C16orf72/HAPSTR1*. Results are reproducible over biological triplicates. (D) Western blot analysis of wild-type p53 levels in human MCF10A mammary epithelial cells overexpressing C16orf72/HAPSTR1, assayed after treatment with Doxorubicin [2 μg/mL] for 6 h. Results are reproducible over biological triplicates. (E) Kaplan-Meier plots depicting the tumor-free survivals of tumor-prone LSL-*Pi3k*$^{H1047R}$ mice (left) and LSL-*Pi3k*$^{H1047R}$; *Trp53*$^{flox/flox}$ mice (right) that were intraductally injected with lentiviral particles expressing Cre as well as C16orf72/HAPSTR1, USP7, Mdm2, or control (mRuby) ($n = 5$ for each condition; n.s. $p > 0.25$, log-rank test was used for statistical analysis). (F) Immunohistochemistry staining of p53 and GFP in mouse mammary hyperplasia and tumor from mice LSL-*Pi3k*$^{H1047R}$; LSL-EGFP intraductally injected with lentiviral particles expressing Cre as well as C16orf72/HAPSTR1 or control. Stage-matched lesions from LV-C16orf72/HAPSTR1-Cre or LV-Ruby-Cre transduced LSL-*Pi3k*$^{H1047R}$ glands were stained for p53 and GFP in consecutive sections and counterstained by Hematoxylin. GFP serves as a lineage tracer to identify transduced cells. Scale bar = 100 μm. Source data are available online for this figure.

may be important for p53 regulation (Jethwa et al, 2018; Yue et al, 2016; Yue et al, 2015; Zhao et al, 2015).

Similarly, we found that CCDC6 also regulates the same p53 mutants as FBXO42 and CCDC6 and FBXO42 were recently shown to interact genetically (Garofano et al, 2021; Shimada et al, 2021). We corroborated this interaction data and showed that they also interact physically. In addition, we found an interaction between CCDC6 and p53 (by PLA and IP) and a direct interaction between FBXO42's Kelch domains and p53 R273H. Epistasis experiments showed that FBXO42 may function upstream of CCDC6 in regulating p53. In addition, we also found that another well-known p53 regulator, BRCA2, interacts with mutant p53 and also surfaced as an FBXO42 vicinal protein and a potential FBXO42 target (Fig. 3A). However, it is still unclear whether these interactions are required to regulate p53 protein levels.

Using further genetic screening in FBXO42- and CCDC6-knockout p53 R273H reporter query lines and additional epistasis experiments, we uncovered that FBXO42/CCDC6 and USP28 interact genetically. USP28 was originally implicated as a protective deubiquitinating enzyme counteracting the proteasomal degradation of p53, TP53BP1, CHEK2, and additional proteins (Cuella-Martin et al, 2016; Lambrus et al, 2016; Wang et al, 2018; Zhang et al, 2006). USP28 regulates wild-type p53 via TP53BP1-dependent and -independent mechanisms. Concordantly, our data shows that USP28 and TP53BP1 are strong positive regulators of wild-type p53. However, while USP28 was also a strong hit in the mutant p53 R273H screen, TP53BP1 was not, indicating that the effects observed upon loss of USP28 on p53 R273H are independent of TP53BP1. In addition, we found that genetic ablation of FBXO42 or CCDC6 leads to a significant reduction of nuclear USP28 levels, adding further biochemical support to the genetic interaction data. Vakifahmetoglu-Norberg et al presented evidence that mutant p53 is resistant to proteasomal degradation due to an inability of mutant p53 to be ubiquitinated, which favors lysosomal degradation of mutant p53 (Vakifahmetoglu-Norberg et al, 2013), and it is conceivable that FBXO42/CCDC6/USP28 might be involved in this phenomenon. While the exact molecular mechanisms are currently unclear, we provide strong evidence that FBXO42/CCDC6 is required for USP28-mediated regulation of p53 R273H stability, suggesting that interfering with this regulatory circuit could present an avenue to prevent or reduce mutant p53 accumulation in tumors.

USP28 also regulates other important proteins, such as MYC, by counteracting FBXW7-mediated proteasomal degradation, as well as cJun, Notch1, LSD1, HIF-1a, and MDC1 (Wang et al, 2018). It will be interesting to elucidate whether FBXO42 and CCDC6 also impinge on those cellular pathways or whether there is some selectivity.

Our data also shows that there are many genes whose loss results in increased p53 levels, which presumably is rooted in the fact that any gene loss which causes cellular stress will probably indirectly lead to p53 stabilization. Therefore, we cross-referenced our p53 stability screen with a synthetic viability screen, which revealed a key role of C16orf72/HAPSTR1 as a negative regulator of wild-type and mutant p53. The role of C16orf72/HAPSTR1 in a p53 regulatory mechanism is further supported by the mutual exclusivity of *TP53* genetic alteration and *C16orf72/HAPSTR1* amplification/gains, which is observed in up to 55% of breast cancer genomes. However, the direct implication of the role of *C16orf72/HAPSTR1* amplification in tumor development is complicated by the fact that the *USP7* gene is located adjacent to the *C16orf72/HAPSTR1* locus, resulting in concurrent amplification of *USP7* and *C16orf72/HAPSTR1*. In addition, recurrent *de novo* copy number amplifications encompassing *USP7* and *C16orf72/HAPSTR1* are also seen in autism spectrum disorder (Sanders et al, 2011), and the role of p53 and the DNA damage response pathway in neurodegenerative diseases and autism is increasingly recognized (Chang et al, 2012; Wong et al, 2016), further indicating a potentially interesting connection.

By generating autochthonous breast cancer mouse models, we could show that overexpression of *C16orf72/HAPSTR1* or *USP7* independently accelerates *Pik3ca*$^{H1047R}$-driven mammary tumor formation. Mechanistically, we show that C16orf72/HAPSTR1 interacts with USP7 and HUWE1, which are bone fide regulators of p53, potentially hinting at how C16orf72/HAPSTR1 may regulate p53 stability. Our findings are also in line with a recent report showing that C16orf72/HAPSTR1 is crucial for mediating telomere attrition-induced p53-dependent apoptosis and regulating the effects of ATR inhibition (Benslimane et al, 2021; Hustedt et al, 2019).

In summary, our study provides a rich resource to mine for candidate regulators of wild-type and mutant p53 stability. More-over, it can serve as a template to reveal regulators of any protein of interest on a genome-wide level.

# Methods

## Materials availability

DNA constructs and other research reagents generated by the authors will be distributed upon request to other academic research investigators under a Material Transfer Agreement. In addition, DNA constructs will also be deposited to Addgene.

## Cell lines

RPE1-hTERT (ATCC CRL-4000), PANC-1 (ATCC CRL-1469), and MDA-MB-468 (ATCC HTB-132) cell lines were maintained in Dulbecco's modified Eagle's medium (DMEM; Wisent Inc.), supplemented with 10% (vol/vol) fetal bovine serum (Wisent Inc.), 100 U/ml of penicillin, and 100 mg/ml of streptomycin (Wisent Inc.) at 37 °C under 5% $CO_2$.

To generate RPE1-based clonal stability reporters, RPE1-hTERT-Cas9-TP53-KO cells (Zimmermann et al, 2018) were transduced with recombinant pLKO.1-based lentiviruses carrying the p53mClover-P2A-mRFP1 cassette at 0.1 MOI, and single clones selected. The pLKO.1-based p53mClover-P2A-mRFP1 cassette carrying control (no p53), wild-type, or hotspot mutant p53s was generated by restriction enzyme cloning, where all hotspot mutants were first generated using site-directed mutagenesis in the pDONR223 entry clone vector and then sub-cloned into the cassette.

To further generate clonal reporters depleted of hit genes, the RPE1-based reporter was transfected with CRISPR ribonucleoprotein (RNP) complexes containing Cas9 and guides against the gene-of-interests. Clonal lines were selected and verified for homozygous deletions by both PCR-based Inference of CRISPR Edits (ICE) and Western blotting.

To generate other cell lines stably expressing Cas9, each line was transduced with lentiviruses expressing lentiCas9-Blast (Addgene #52962) and selected by Blasticidin to generate either single or clonal lines as specified.

## CRISPR and FACS-based screens

Stability reporters [150 million (M) cells total] cultured on 15 cm dishes (3 M/plate) were first transduced with the lentivirus-based Toronto Knockout v3 library (TKOv3) (Hart et al, 2017) at a low MOI (~0.30). They were then puromycin-selected [17 µg/mL] for two days starting at 24th hour-post-transduction (hpt). At 48 hpt, cells were trypsinized and replated back to the same plates while maintaining puromycin-selection. At 72 hpt (time point T0), the remaining cells were pooled together, counted to confirm MOI, and divided into two technical replicates (referred to as p53.1 and p53.2 in figures) for subculturing; 30 M cells were further collected to confirm the library gRNA abundance at T0. The sub-cultured replicates were further maintained in puromycin-free media and passaged every three days (T3 and T6).

10 dpt (T7), all cells were harvested by trypsinization and subjected to live sorting by Fluorescence-Activated Cell Sorting (FACS). Harvested cells were resuspended in FACS sorting buffer (Hanks Balanced Salt Solution, 25 mM HEPES pH 7.0, 2 mM EDTA, 1% Fetal Bovine Serum) at a concentration of 5 M cells per mL and were filtered by 40-µm nylon mesh to eliminate large

aggregates. All cells were then live-sorted (MoFlo Astrios EQ cell sorter, Beckman Coulter) based on both the mClover and mRFP1 signals, where only populations with medium-mRFP1 signals were collected to eliminate populations with large gene-expression changes. For mutants displaying bimodal-mClover populations, all cells from the mClover-low and mClover-high populations were each collected; for uni-modal mutants, cells with mClover intensity within the highest and lowest 15% populations were each collected (Fig. 1D). Both technical duplicates were independently sorted and collectively maintained the 200× coverage.

Upon sorting, gDNA from cell pellets was isolated using Wizard Genomic DNA Purification Kit (Promega, Cat# A1120); genome-integrated sgRNA sequences were then amplified by PCR using Q5 Mastermix Next Ultra II (New England Biolabs, Cat# M5044L), with primers v2.1-F1-5' GAGGGCCTATTTCCCATGATTC 3' and v2.1-R1-5' GTTGCGAAAAAGAACGTTCACGG 3', followed by a second round of PCR reaction containing i5 and i7 multiplexing barcodes. Final PCR products were gel-purified and sequenced on Illumina NextSeq500 systems to determine sgRNA representation in each sample. The abundance of each guide (guides count) was then analyzed by MAGeCK count function with default settings (Li et al, 2014).

To generate the log-fold-change (LFC) rank of each gene, the guides counts from both the p53-low and p53-high populations were processed and compared with the MAGeCK test function with default settings (Li et al, 2014), and the LFC (p53-high vs p53-low) and padj values were computed.

To generate the normZ scores for normalizing the LFCs across all screens, the guides counts from both the p53-low and p53-high populations were then analyzed by the DrugZ program with default settings (Colic et al, 2019). The normZ and padj values were computed. In this study, a hit was defined as having a |normZ| > 3 and padj < 0.5.

All flowcytometry was performed on Fortessa X20 (BD), and data was analyzed using the FlowJo software (BD).

## Plasmids, transfection, and transduction

To generate the lentiviral plasmids for transducing the reporter lines, the reporter plasmid was first built upon the pLKO.1 - TRC cloning vector (Addgene #10878), by substituting the Puromycin resistance gene with the p53mClover2-P2A-mRFP1 or mClover2-P2A-mRFP1 (empty control) cassette. All hotspot mutants were first generated using site-directed mutagenesis in the pDONR223 entry clone vector and then sub-cloned into the cassette by substituting the relevant regions of WTp53. All constructs were confirmed by DNA sequencing. To generate V5-tagged lentiviral overexpression constructs, each open reading frame (ORF) was first generated as an entry clone into the pDONR223 vector using the BP recombination reaction, and then was cloned into the pLEX_306 expression vector (Addgene #41391) using the LR recombination reaction, following manufacturer's instruction (Invitrogen). Lastly, HA-C16orf72/HAPSTR1 was generated in the pCDNA3.1-HA backbone, and the FLAG-USP7 was a generous gift from Lori Frappier (University of Toronto).

To generate the entry clone for each of the p53 mutants, the coding sequence of WT p53 (NM_000546.6:143-1324) was first cloned into the pDONR223 vector, retaining Start codon (ATG)

and removing the Stop codon (TGA), using the BP recombination reaction (Invitrogen). Each mutant was then generated using PCR-based site-directed mutagenesis to generate p53$^{mutant}$-pDONR223, with specific codon and primers detailed in Dataset EV7. All constructs were sequence verified by Sanger sequencing covering the entire ORF.

For in vivo overexpression assays, the pLEX_306-ORF-Cre backbone was first generated by modifying the pLEX_306 by substituting the Puromycin resistance cassette with NLS-iCre. ORFs were then introduced by pDONR223-based entry clone similar to the V5-tagged constructs above. For in vivo competition assays, pLKO-H2BXFP-P2A-NLSiCre was first generated by replacing the NLSiCre with the H2BXFP-P2A-NLSiCre cassette in the pLKO-Cre stuffer v4 (Addgene #158032) backbone; the XFP used was mRFP1 or TagBFP. The guides targeting each desired gene were then cloned into the vector, and the sequences were confirmed.

For transfections, cells were grown in 100-mm (for immuno-precipitation) dishes to about 70% confluence and transfected with Lipofectamine 3000 (Invitrogen) according to the manufacturer's protocols.

For all lentiviral transduction for in vitro (cultured cells) experiments, lentiviral particles were first produced in the 293 TN cells (Systembio LV900A-1), by co-transfecting the lentiviral plasmid with helper plasmids pMD2.G and psPAX2. Supernatant was then harvested at 48 h post-transfection, and used to transduce cells at an MOI of ~0.1 to 0.3.

For all in vivo lentiviral transductions, lentiviral particles were first produced similar to those for in vitro experiment, and followed by concentration of the viruses. In brief, the produced supernatant was first filtered through a Stericup-HV PVDF 0.45 μm filter, and then concentrated by ~1000 fold by ultracentrifugation (Beckman Coulter). The titer was quantified by FACS of LSL-tdTomato mouse embryonic fibroblasts (MEFs).

sgRNA sequences for validation experiments from screen hits are detailed in Dataset EV7.

## Analysis of genome editing efficiency

PANC-1 Cas9 cells (for human guides) or LSL-Cas9-EGFP MEFs (for mouse guides) were cultured and transduced with sgRNAs-expressing lentivirus that carries either Puro (for PANC-1) or NLSiCre (for MEFs). carrying each guide. For PANC-1 cells, transduced cells were first selected under Puromycin (5.0 μg/mL) for 48 h, and then cultured in complete media with passaging every three days until harvesting on day 10 post-transduction. For MEFs, cells were live sorted for GFP expression and expanded further until harvesting on day 20 post-transduction. At the time of harvesting, the cells were collected by trypsinisation and genomic DNA was extracted using DNeasy Blood & Tissue Kit (Qiagen). The genomic region centered at the sgRNA cutting site, along with >250 bps flanking it on each side, was PCR amplified, for both cells transduced with guides targeting the desired genes or the control [sgAAVS1 (human) or sgTigre (mouse)]; they were then subject to Sanger sequencing. The editing efficiency was then determined by analyzing the sequencing chromatograms with the web-based Interference of CRISPR Edits (ICE) tool, https://www.synthego.com/products/bioinformatics/crispr-analysis. Primers for ICE analysis are detailed in Dataset EV7.

## Irradiation and chemical perturbations

RPE1 reporters were treated with ionizing radiation (IR) with indicated dose using a Faxitron X-ray cabinet (Faxitron, Tucson, AZ, USA). Chemical perturbation was performed by adding either Nutlin-3a (10 μM, Sigma SML0580) or an equal volume of DMSO to the complete media. For MG132 treatment of the reporters, either MG 132 (10 μM, Sigma) or DMSO was added to the complete media for indicated times.

## BioID and affinity purification mass spectrometry (AP-MS)

**BioID group (p53 R273H, CCDC6 and FBXO42).** BioID was carried out, essentially as previously described (Coyaud et al, 2015; Hesketh et al, 2017; Roux et al, 2012). In brief, 293 Flp-In T-REx (Invitrogen) cells inducibly expressing C-terminally-tagged full-length human p53R273H, CCDC6, and FBXO42 or controls (GFP in lieu of ORF, and vector control alone) were first generated, and inducible expression tested by immunoblotting. Sub-confluent (60%) cells (10 × 15 cm plates) were incubated for 24 h in complete media supplemented with 1 μg/ml tetracycline (Sigma) prior to incubation with 50 μM biotin (BioShop, Burlington, ON, Canada), and either 5 μm MG132 (5 plates; calpain inhibitor IV, Z-Leu-Leu-Leu-CHO; American Peptide Company, Sunnyvale, CA) or DMSO (5 plates) for 24 h. Cells were then collected by first washing twice with PBS and then pelleted (500 × $g$, 5 min), and dried pellets were finally snap-frozen.

**Affinity purification (BioID group).** Cell pellets corresponding to each 15 cm plate were incubated at 4 °C in 1:10 (w/v) modified-RIPA buffer (50 mM Tris-HCl pH 7.5, 150 mM NaCl, 1% Triton X-100, 1.5 mM MgCl$_2$, 1 mM EGTA, 0.1%SDS, Sigma protease inhibitors P8340 1:500, and 0.5% Sodium deoxycholate), and 1 μL of benzonase (250U) was added to each sample, for 30 min on a rotator. The lysates were then sonicated three times on ice at 35% amplitude. These lysates were then centrifuged and supernatant proceeded for streptavidin-Sepharose beads (GE Cat# 17-5113-01) affinity purification, which was performed at 4 °C on a rotator for 3 h, followed by washing the beads once with 2% SDS buffer, twice with modified-RIPA buffer and lastly once in TAP lysis buffer (50 mM HEPES-KOH pH 8.0, 100 mM KCl, 2 mM EDTA, 0.1% NP-40). Finally, the beads were washed in the ABC buffer (50 mM ammonium bicarbonate, pH 8.3), and proteins were digested on beads with TPCK-trypsin (Promega, Madison, WI, 16 h at 37 °C). The supernatant containing the tryptic peptides was collected and lyophilized. Peptides were resuspended in 0.1% formic acid and analyzed by mass spectrometry.

**AP-MS group (C16orf72/HAPSTR1).** Similarly, 293 Flp-In T-REx (Invitrogen) cells inducibly expressing N-terminally-tagged full-length human C16orf72/HAPSTR1 or GFP-control were generated and inducible expression tested by immunoblotting. Sub-confluent (60%) cells (2 × 15 cm plates) were incubated for 24 h in complete media supplemented with 1 μg/ml tetracycline (Sigma) prior to harvesting, similar to the BioID group.

**Affinity purification (AP-MS group).** Cell pellets corresponding to 2 × 15 cm plate were incubated at 4 °C in 1:4 (w/v) FLAG-IP lysis buffer (50 mM HEPES pH 8.0, 100 mM KCl, 2 mM EDTA, 0.1% NP40, 10% glycerol, 1 mM PMSF, 1 mM DTT, and Sigma protease inhibitors P8340 1:500). Resuspended lysates were

first frozen and thawed twice (dry ice and 37 °C water bath) for 5–10 min each. They were then sonicated for 20 s at 35% amplitude. 200 units of benzonase were then added to the sample and incubated at 4 °C on a rotator for 20 min, and centrifuged to collect the supernatant ($16,000 \times g$, 20 min, 4 °C). The supernatants were then incubated with magnetic anti-FLAG M2 beads (Sigma, # M8823) at 4 °C on a rotator for 2 h, at a ratio of 25 µL 50% slurry beads for each IP of $2 \times 15$ cm plates. Beads were then pelleted by centrifugation ($500 \times g$, 5 min), followed by three washes with 1 mL FLAG-IP lysis buffer and two washes with washing buffer (20 mL Tris-HCl pH 8.0 and 2 mM $CaCl_2$) using a magnetic stand. The samples were then processed similarly to the BioID group for on-beads digest and mass spectrometry analysis.

**Mass spectrometry and data analysis**. Mass spectrometry analysis was carried out as previously described (Hesketh et al, 2017) using the TripleTOF 6600 system (SCIEX). In brief, samples were loaded to fused silica capillary columns pre-loaded with C18 reversed phase material. Ionised peptides were emitted by nanoelectrospray ion source followed by a nano-HPLC system, and analyzed using Data Independent Acquisition (DIA) methods.

**Interactor classification**. Mass spectrometry data were filtered for a minimum unique spectral count of 2. For the AP-MS group, results from biological triplicates purifications of the C16orf72/HAPSTR1 and control were further filtered via significance analysis of interactome (SAINT) (Teo et al, 2016), which uses a probability model to assign a confidence score to each interaction by comparing the spectral counts in the sample and control across replicates, filtered at ProteinProphet $p$ value > 0.95 and SAINT BFDR score ≤0.01.

## Synthetic lethal screens

These screens were carried out following previously optimized protocols (Hart et al, 2015). In brief, isogenic pairs of RPE1-hTERT-Cas9 cells with $TP53^{+/+}$ and $TP53^{-/-}$ backgrounds were each infected with the TKOv3 pooled gRNA library at an MOI of ~0.3. Similar to the stability screen, the infected pools were then selected with puromycin (17 µg/ml, 48 h) and maintained in culture for 18 days after Day 0, with passaging every three days. The genomic DNA from the first and the last time point was extracted, and the incorporated gRNA sequences were amplified via 2-step polymerase chain reaction (PCR). The amplification products from the first and the last time point were subjected to Illumina sequencing in order to analyze the fold change of gRNA sequence in the cell population.

Data was analyzed with BAGEL2 (Kim and Hart, 2021), where positive Bayes factors (BF) values identify genes that are likely essential for proliferation. Hits are defined as concurrently fulfilling BF > 15 for $TP53^{+/+}$ and BF < 5 for $TP53^{-/-}$.

## Fluorescence microscopy

Cells were cultured on glass coverslips in 6-well plates for 48 h (reaching 70% confluency) before fixation with 3.2% paraformaldehyde for 20 min, followed by permeabilization (0.25% v/v Triton X-100 in 1× Phosphate Buffer Saline (PBS) and blocking (1% w/v BSA, 1% w/v gelatin, 0.25% v/v goat serum, 0.25% v/v donkey serum, 0.25% v/v Triton X-100 in PBS, in 1× PBS). Samples were then stained for respective antibodies diluted in the blocking buffer for 1 h at room temperature, followed by washing and staining with secondary antibodies. Coverslips were lastly counterstained with DAPI (4′,6-diamidino-2-phenylindole) and mounted on slides using DABCO (Sigma). Cells were then viewed using a Nikon Ti2-E/A1R-Multiphoton microscope equipped with DS-Qi2 camera (Nikon).

For quantification, laser power and gain for each channel and antibody combination were set using secondary-only control and confirmation with primary positive control and applied to all images. Images were analyzed using Cellprofiler (Lamprecht et al, 2007) with default settings to quantify the cytoplasmic to nuclear intensity ratios. The following antibodies were used for IF: p53 (DO-7 FITC conjugate, BD biosciences), USP28 (Bethyl A300-898A), and F-Actin (phalloidin-FITC, Sigma P5282).

## Histology and immunohistochemistry

Tumors or gland tissues were harvested from each endpoint mouse, and placed in 4% PFA for 48 h. They were then placed into 70% ethanol and stored at 4 °C until ready for standard embedding and sectioning procedure. For staining, the dissected serial section slides were heated at 60 °C for 15 min, then dewaxed and rehydrated. Slides for immunohistochemistry were treated with 3% hydrogen peroxide in PBS for 15 min to deactivate endogenous peroxidases. Slides were then washed in PBS followed by microwave antigen retrieval using Na citrate pH 6.0. Following the primary antibody (diluted in Na citrate solution) incubation at room temperature for 45 min, anti-rabbit secondary antibodies (Vector Labs BA-1000, 1:500 diluted in 0.2% v/v Triton X-100 in PBS) were applied for 35 min at room temperature. This was further followed by an ABC kit (Vector Labs PK-4100) treatment for 25 min, and finally, a DAB Reagent (Vector Labs SK-4100) treatment for 4 min at room temperature. Slides were lastly counterstained for 8 min in Harris Hematoxylin, dehydrated, and mounted in a xylene-based mounting medium. Stained sections were digitized at 40x using a Hamamatsu Nanozoomer Scanner (2.0-HT).

The following primary antibodies were used in this study: anti-p53 rabbit polyclonal antibody (Abcam ab241566, POE316A), anti-GFP rabbit polyclonal antibody (Abcam ab290). Antibodies are detailed in Dataset EV7.

## RNA isolation, cDNA synthesis, and Quantitative Real-Time PCR analysis

Total RNAs from PANC-1 cells were first collected using TRIzol (Ambion), treated with ezDNase (Invitrogen), and reverse transcribed into cDNA using SuperScript IV VILO (Invitrogen). Real-time quantitative PCR (qRT-PCR) reactions were performed on an CFX384 (Biorad) in 384-well plates containing 12.5 ng cDNA, 150 nM of each primer, and 5 µl PowerUp SYBR Green Master Mix (Applied Biosystems) in a 10 µL total volume. All primers were designed to span exon junctions using Primer3Plus and were melting-curve validated. Relative mRNA levels from experimental triplicates were calculated using the comparative Ct method normalized to *PPIB* mRNA.

With the normalized mRNA levels of targeted genes in the sgAAVS1 sample arbitrarily set as 1.00, the normalized mRNAs in other samples were expressed as a ratio relative to that of sgAAVS1. Statistical significance was determined at $p = 0.05$. Primers for qPCR are detailed in Dataset EV7.

## Immunoprecipitation and Western blotting

Cell extraction and immunoprecipitation were performed as previously described (Lu et al, 2014). In brief, whole-cell extracts were prepared by lysing cells in buffer X (50 mM Tris pH 8.5, 250 mM NaCl, 1 mM EDTA, 1% NP-40, protease inhibitor minitablet (Roche)) and quantified using Bradford assay (Bio-Rad). Equal amounts of protein (lysate or immunoprecipitation samples) were separated by SDS-PAGE and transferred to 0.45 μm Polyvinylidene fluoride (PVDF) membranes (Immobilon-P, EMD Millipore). For affinity purification of endogenous or epitope-tagged binding proteins, the lysates were incubated with anti-CCDC6 mouse monoclonal antibody (Santa Cruz Q-23) or mouse IgG (Abcam, ab124055), anti-V5 mouse monoclonal antibody (Roche R960-25), or anti-FLAG mouse monoclonal antibody (Sigma M2), followed by protein G beads (Sigma).

For Western blotting, the following antibodies were used for Western blot: p53 HRP-conjugated antibody (R & D Systems, HAF1355), p53 rabbit polyclonal antibody (Santa Cruz FL-393), CCDC6 (Santa Cruz Q-23), FBXO42 (Santa Cruz FL-6), GAPDH HRP-conjugated antibody (BioLegend W17079A), TP53BP1 (BD Biosciences 612523 Clone 19), USP28 (Bethyl A300-898A), USP7 (Bethyl A300-034A), HUWE1 (Abcam ab70161), V5 (Roche R960-25), FLAG (Sigma M2 F7425), HA (Covance HA.11 MMS-101R), and C16orf72/HAPSTR1 (rabbit polyclonal, in house). Membranes were incubated with primary antibodies followed by appropriate horseradish peroxidase-coupled secondary antibodies (anti-rabbit or anti-mouse from Jackson ImmunoResearch or TrueBlot anti-mouse from eBioscience). Western Lightning Plus enhanced chemiluminescence substrate (PerkinElmer) was used to visualize proteins on ChemiDoc MP Imager with Image Lab 4.1 software (Bio-Rad) or autoradiography film. Antibodies are detailed in Dataset EV7.

## Nuclear cytoplasmic fractionation

The fractionation was performed using NE-PER Nuclear and Cytoplasmic Extraction Reagents (ThermoFisher) as per the manufacturer's instructions. 10 μg of each fraction was separated by SDS-PAGE and transferred to 0.45 μm Polyvinylidene fluoride (PVDF) membranes (Immobilon-P, EMD Millipore).

## Clonogenic survival assays

RPE1 cells depleted of each gene using each indicated guide were seeded in 6-well plates (250 cells per well) and cultured using complete media. After 14 days, colonies were stained with crystal violet solution (0.4% w/v crystal violet, 20% methanol), scanned and manually counted. Relative survival was calculated by arbitrarily setting the number of colonies in the Scrambled control as 100%. Experiments were performed in biological triplicates, and the error bar represents the standard error of the mean.

## In situ proximity ligation assay

PLA was performed as previously described (Mukherjee et al, 2022). In brief, cells were first fixed with 4% PFA for 15 min and permeabilized with 0.1% v/v Triton for 5 min. PLA was performed using the DuoLink In Situ PLA Detection Kit (DUO92101, Sigma). Imaging was performed using an LSM 800 (Zeiss) confocal microscope with 40/60× objective oil immersion.

## Mice

Animal husbandry, ethical handling of mice and all animal work were carried out according to guidelines approved by the Canadian Council on Animal Care and under protocols approved by the Centre for Phenogenomics Animal Care Committee (18-0272H).

### In vivo competition and overexpression

The parental animals used in this study were Rosa26-LSL-Pik3caH1047R/+ [Gt(ROSA)26Sortm1(Pik3ca*H1047R)Egan in a pure FVBN background, a generous gift of Sean Egan, The Hospital for Sick Children/SickKids], Rosa26-LSL-Cas9-GFP (Jackson laboratories #026175, in C57/Bl6 background), and p53 flox/flox (B6;129S4-Trp53tm5Tyj/J, Jackson laboratories #008361 in Bl6 background]. Homozygous [p53 flox/flox; Rosa26-LSL-Cas9 GFP], [LSL-Pik3caH1047R], and [p53 flox/flox; LSL-Pik3caH1047R] were each generated by crossing the respective parental lines.

Mice were intraductorally injected with lentiviral particles containing either (1) sgRNAs (competition assay) in the #3 or 4 mammary glands with triplicates spread across a minimum of two mice (i.e., no more than two glands per mouse containing the same virus), or (2) ORFs (tumor-free survival assay) in both #3 and #4 glands (a total of four per mouse) with five mice per ORF.

**Tumor-free survival**. Mice were monitored weekly (initial three weeks) and then twice weekly for tumor formation by palpation, and the first appearance of tumors in any gland of the mouse was noted. Mice were harvested when tumors associated with any one gland had reached the tumor burden threshold as defined by the animal ethics guidelines.

**Competition**. Mice were intraductorally injected with a mixture of control lentiviral particles expressing Cre and BFP as well as an sgRNA targeting the *Tigre* safe harbor, and experimental lentiviral particles expressing Cre and RFP as well as an sgRNA targeting *Tigre*, *C16orf72*, or *Mdm2*. The number of surviving cells that had been depleted of each gene was counted 12 days post-injection, and normalized to the number of cells depleted of *Tigre* in the same gland. This ratio was further normalized to the ratio of sgTigre:sgTigre in the LSL-Cas9-EGFP; *Trp53*^flox/flox mouse.

**Mammary gland isolation and flow cytometry for lineage tracing**. For competition assays, individual mammary glands were harvested and digested according to Stemcell Technologies gentle collagenase/hyaluronidase protocol. In brief, glands were first digested overnight (~16 h) with gentle agitation at 37 °C in 250 μL Gentle Collagenase (Stemcell Technologies #07919) diluted in 2.25 mL of complete Basal Epicult media formulated according to manufacture instructions (Epicult Basal Medium Stemcell Technologies #05610, 10% Proliferation Supplement, 5% v/v FBS, 1% v/v Penicillin-Streptomycin, 10 ng/mL EGF, 10 ng/mL bFGF, 0.0004% v/v heparin). The digested glands were then treated with ammonium chloride and triturated for 2 min in pre-warmed trypsin, followed by dispase. Cells were stained with CD45, CD31, Ter119, CD49f, and EPCAM for luminal and basal cell identification, and Sytox Red for dead cell exclusion. The following antibodies were used for flowcytometry experiment: APC conjugated antiCD45 (Clone 30 F11, BioLgend), APC conjugated antiCD31 (Clone MEC133, BioLegend), APC anti-mouse Ter119 (Clone TER-119, BioLegend), PECy7 anti-human/mouse CD49f (Clone GoH3, BioLegend), APCVio770 mouse anti-CD326 EpCAM (Clone caa7-9G8, Miltenyi).

## Xenografting

PANC-1-Cas9 and PANC-1-Δ p53-Cas9 cells that were transduced with sgRNA-GFP (12 days post-transduction) were tail vein injected to NOD/SCID mice in replicates of five mice per experimental condition (*AAVS1, CCDC6, FBXO42*, and *TP53*). In brief, the cell suspension was prepared to a concentration of $2.5 \times 10^5$ cells in 50 µL of PBS solution per mouse, and kept on ice with occasional agitation. They were then injected into the tail vein of eight-week-old recipient NOD-SCID mice (NOD.Cg-*Prkdc^scid Il2rg^{tm1Wjl}*/SzJ) under anesthesia with isoflurane. Mice were monitored weekly (first three weeks), twice weekly (weeks four to six), and daily (week seven) for welfare. The metastatic colonization was observed at seven weeks post-tail vein injection. Mice were sacrificed, and their lungs were harvested and imaged under the fluorescence microscope for metastatic colonization in the lungs, as indicated by the green fluorescence signal and quantified by the spot count. The lungs were preserved by fixing with PFA.

## Statistics and reproducibility

Unless specified, independent biological replicates were performed, and group comparisons were made as indicated in the legends. $P < 0.05$ was considered significant. Statistical analysis was performed using GraphPad Prism 7. Unless specified, quantitative data are expressed as the mean ± SE. Differences between groups were calculated by two-tailed Student's t-test, Wilcoxon Rank-Sum test (normal distribution correction), or Log-rank test using Prism 7.

## Recombinant expression and purification of prey and bait proteins for pull down assays

Protein purification was performed as previously described (Xiao et al, 2020). All protein constructs [preys: p53-CD-R273H (aa 90-311), p53-FL-WT, and MBP-tagged baits: MBP-FBXO42c (aa 105-360), MBP-p107, and MBP-MDM2] were cloned in the pETM41 vector. p53 constructs were transformed in BL21DE3 *E. coli* cells and grown in 2TY medium supplemented with kanamycin at 200 rpm and 37 °C. Cultures were supplemented with 100 µM $ZnSO_4$ and were induced with 0.8 mM IPTG upon reaching an $OD_{600} = 0.6$. Following induction, cultures were grown ON at 200 rpm and 17 °C. Bacterial pellets were harvested and pellets lysed and passed over an Amylose column (NEB) with Buffer A: Tris-HCl 50 mM pH 7.5, PMSF 2 mM, NaCl 0.4 M, beta-mercaptoethanol 2 mM and eluted with Buffer B containing 20 mM Maltose. Elution peaks were collected and dialyzed overnight, at 4 °C in pre-IMAC Buffer (25 mM Tris-HCl pH 7.5, 0.2 M NaCl). The sample was removed from dialysis and stored cut with TEV protease to remove MBP 3 h at room temperature (Conditions: 1:5 m:m/1 mg TEV: 5 mg protein) followed by IMAC (Buffer A: Tris-25 mM HCl pH 7.6, 0.5 M NaCl, 1 mM PMSF; Buffer B: Buffer A + 0.5 M Imidazole pH 7.6). Flow fractions containing p53 constructs were collected and dialyzed ON in S75 Buffer (20 mM phosphate pH 7.0, 2 mM beta-mercaptoethanol, 200 mM NaCl, 2 mM PMSF). The sample was removed from dialysis and concentrated to 2.88 mg/ml (115.2 µM). Purity was assessed by SDS-PAGE and conformational homogeneity was assessed by Size Exclusion Chromatography. MBP-FBXO42c, MBP-MDM2, and MBP-p107 were similarly expressed as described above (without the addition of $ZnSO_4$) and 15 ml pellets were lysed, sonicated and coupled to Amylose resin.

## Size exclusion chromatography

An analytical Superdex 75 column (Cytiva) volume 21.23 ml was equilibrated in 20 mM sodium phosphate buffer (Biopack), 2 mM DTT (Sigma), 200 mM NaCl at pH 7.00. Runs were performed at a flow rate of 0.5 ml/min. 150 µL of marker or p53 mixture was prepared, brought to volume with buffer, and 100 µL loop injections were performed. The p53-CD-R273H stock was used at a concentration of 2.88 mg/ml (115.2 µM). Markers used were: Acetone 2% (Vo+Vi), Blue dextran 1 mg/ml (Vo), Immunoglobulin 2.5 mg/ml, Transferrin 1 mg/ml, BSA 5 mg/ml, and Trypsin inhibitor 3 mg/ml.

# Data availability

All mass spectrometry spectra counts are found in the data-sets EV2 and EV6. Raw data files have been deposited as a complete submission to the MassIVE repository (https://massive.ucsd.edu/ProteoSAFe/static/massive.jsp) and assigned the accession number MSV000093628 (p53 stability BioID), MSV000093630 (C16orf72 HEK293 AP-MS), and MSV000093632 (C16orf72 U2-OS AP-MS). The ProteomeXchange accession is PXD047710, PXD047717, and PXD047718, respectively.

# Peer review information

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

## Acknowledgements

We would like to thank all members of our laboratories, as well as Mikko Taipale (University of Toronto), Brian Raught (University of Toronto/University Health Network), Yael Aylon (Weizmann Institute of Science), and The Centre for Phenogenomics (TCP), for helpful discussions. We want to thank Henrique Melo, Zheng Luo, Mingkun Wu and Matthew Guo for assistance with data analysis, Annie Bang, Michael Parsons and Dione White in the Lunenfeld-Tanenbaum Research Institute Flow Cytometry Facilities for assistance with FACS, Kin Chan at the Lunenfeld Network Biology Centre for assistance with next-generation sequencing, Andrew Elia at the University Health Network for assistance with histology, and Sampath Loganathan for general experiment helps. The FLAG-USP7 construct is a generous gift from Lori Frappier (University of Toronto). Research in this work is supported by funding from the Joint Canada-Israel Research Program (DS, MO, VR, ACG, PA-P, and LBC, CIHR IRDC 384428). DS was supported by the Canada Research Chairs Program. DD was supported by a grant from the Canadian Cancer Society (705644). YQL is the recipient of doctoral fellowships from the Canadian Institute of Health Research (CIHR 157921 and MSFSS 431649), the Government of Ontario (OGS) and the University of Toronto. DS is supported by the Canada Research Chair Program. PA-P was supported by a scholarship from CNPq (Conselho Nacional de Desenvolvimento Científico e Tecnológico, Brazil, # 307826/2017-1).

## Author contributions

**Yiqing Lü**: Conceptualization; Data curation; Formal analysis; Investigation; Visualization; Methodology; Writing—review and editing. **Tiffany Cho**: Conceptualization; Data curation; Formal analysis; Investigation. **Saptaparna Mukherjee**: Investigation. **Carmen Florencia Suarez**: Investigation. **Nicolas S Gonzalez-Foutel**: Investigation. **Ahmad Malik**: Formal analysis. **Sebastien Martinez**: Investigation. **Dzana Dervovic**: Investigation. **Robin Hyunseo Oh**: Investigation. **Ellen Langille**: Investigation; Methodology. **Khalid N Al-Zahrani**: Investigation. **Lisa Hoeg**: Formal analysis. **Zhen Yuan Lin**: Investigation. **Ricky Tsai**: Investigation. **Geraldine Mbamalu**: Resources; Investigation. **Varda Rotter**: Funding acquisition. **Patricia Ashton-Prolla**: Funding acquisition. **Jason Moffat**: Resources. **Lucia Beatriz Chemes**: Conceptualization; Formal analysis; Supervision; Funding acquisition; Visualization; Methodology. **Anne-Claude Gingras**: Conceptualization; Formal analysis; Funding acquisition; Methodology; Writing—review and editing. **Moshe Oren**: Conceptualization; Formal analysis; Supervision; Funding acquisition; Writing—review and editing. **Daniel Durocher**: Conceptualization; Supervision; Funding acquisition; Project administration; Writing—review and editing. **Daniel Schramek**: Conceptualization; Supervision; Funding acquisition; Writing—original draft; Project administration; Writing—review and editing.

## Disclosure and competing interests statement

The authors declare no competing interests. ACG is an editorial advisory board member. This has no bearing on the editorial consideration of this article for publication.

