## [Peer Review File · Molecular Systems Biology]

Genome-wide CRISPR screens identify novel regulators of wild-type and mutant p53 stability

Yiqing Lu, Tiffany Cho, Saptaparna Mukherjee, Carmen Florencia Suarez, Nicolas Gonzalez-Foutel, Ahmad Malik, Sebastien Martinez, Dzana Dervovic, Robin Oh, Ellen Langille, Khalid Al-Zahrani, Lisa Hoeg, Zhen-Yuan Lin, Ricky Tsai, Geraldine Mbamalu, Varda Rotter, Patricia Ashton-Prolla, Jason Moffat, Lucia Chemes, Anne-Claude Gingras, Moshe Oren, Daniel Durocher, and Daniel Schramek

Corresponding author(s): Daniel Schramek (schramek@lunenfeld.ca)

Review Timeline:

Transfer from Review Commons:	12th Sep 22
Editorial Decision:	16th Sep 22
Revision Received:	7th Dec 23
Editorial Decision:	4th Jan 24
Revision Received:	6th Mar 24
Accepted:	12th Mar 24

Editor: Maria Polychronidou

Transaction Report: This manuscript was transferred to Molecular Systems Biology following peer review at Review Commons.

Review #1

1. Evidence, reproducibility and clarity:

Evidence, reproducibility and clarity (Required)

Summary:

In this manuscript by Lu et al., the authors describe some CRISPR screens and protein-protein interaction screens to identify novel regulators of wild-type p53 and mutant p53 function and stability. Besides generating a wealth of data, they discover FBXO42-CCDC6 as positive regulators of the some p53 hot-spot mutants, including R273H mutant p53, but not of all p53 mutants tested and also not of wild-type, indicating selectivity. Furthermore, the found C16orf72(TAPR1) as a negative regulator of p53 stability.

Mechanistically, the authors claim a direct interaction between FBXO42 and CCDC6 and p53, but the importance of these interactions has not been shown. On the other hand the authors suggest that the FBXO42/CCDC6 regulate p53 via destabilization of USP28, but also the mechanism has not been worked out. For c16orf72, they show that it interacts with USP7, but no relevance of this interaction is shown either.

Major points

One very important point for me is that the authors do not show the levels of expression of p53 in the p53-mClover stable cell lines. It is known that overexpressed p53 is usually more stable than endogenous levels of wt-p53. Therefore, I think it is necessary that the authors show the levels of the p53-mClover fusion proteins in the stably transduced cell lines compared to endogenous p53 levels in the parental RPE1 cells and also compared to the endogenous levels of R273H mutant in the PANC-1 cells.

Also the functionality of the wild-type p53-mClover fusion is questionable, at least not shown. One would expect that the overexpression of a functional wt-p53 in p53-KO cells will affect the survival of the RPE1 cells. In Figure 5A the authors show that depletion of MDM2 or C16ORF72 is toxic for the RPE1 cells in a p53-dependent manner, indicating that elevated levels of p53 cannot be handled by these cells. So, experiment(s) showing that the wt-p53/mClover fusion is functional is needed.

A second important point is that the 'verification' of the hits from the screens is only done in one cancer cell line, PANC-1, with mutant p53. I would have like to see at

least one other cell line with another p53 mutant endogenously expressed that is also regulated by FBXO42/CCDC6.

For many of the p53-mutants, a bimodal expression is observed. In the FBXO42- and CCDC6-depleted cells, the equilibrium shifts towards more negative cells but the levels in the two populations itself don't change (while for example for USP28 depletion also the right peak shifts further up, Fig S4E). Is there any correlation with the cell cycle and p53 expression? And can the authors exclude that FBXO42 and CCDC6 are involved in cell cycle progression and hereby influence p53 indirectly (by combining PI staining with Clover-p53 for example).

- The authors claim that the FBXO42-CCDC6 axis regulates stability specifically some p53-mutants, including R273H-mutant, in a manner involving USP28. But USP28 regulates all forms of p53, not just some mutants version. How can the authors reconcile this apparent contradiction?

On a similar note, the authors show that FBXO42 and CCDC6 interact with p53, but not USP28. Do FBXO42 and CCDC6 interact with each other and with USP28? And is the interaction with p53 specific for the R273H version? This part of the mechanism is very poorly defined and the Co-IPs are not very convincing or relevant for the proposed model.

****Minor points****

The mechanisms of p53 regulation may vary greatly in different cell lines. Can the authors discuss why they choose to do the screen with different mutants, rather than with different cell lines expressing these same mutant endogenously? .

Figure 1: Typo in the legends : Nultin ipv Nutlin

Figure 1b,1c : Show basal and Nutlin-3 induced MDM2 levels and in the overexpression cell lines; if WT-p53 is functional, MDM2 levels should be higher in WT-transduced cells compared to control or mt-p53 expressing cells. Authors should explain which they name USP7 a negative regulator of p53, since it is supposed to de-ubiquitinate p53?!

Figure 2E: the effect of MG132 on p53 seems to be very minimal on this Western blot; it would need quantification to be convincing...Quality of the blot is also not great.

The fact that in control cells the levels of p53 R273H are not affected by MG132 treatment fits with Suppl Figure 2E, indicating that the proteasome has no effect on

p53 R273H.

Suppl figure 3b, 3c, 3d:

Somehow, I have the feeling that the results from the western blots and the FACS do not match fully, although not all the time-points are shown in the various experiments. For example, the FACS analysis (3b) suggests that in control-transduced cells after 16 hr p53 is still increased. However, that is not clear at all in the Western blot (3c) Is Suppl Figure 3d the quantification of 3c experiment? If so, in the blot also the 24 hrs should be shown.

The blot shown in Suppl Figure 3c suggests that CCDC6 expression increased upon irradiation. Do the authors agree with that? Would that explain why depletion of CCDC6 has more effect upon irradiation?

Suppl Figure S3E: if I am right, this is essentially the same type of experiment as shown in figure 2e, but analysis of p53-expression by Western blot. In that blot no real effect of MG132 on p53 levels could be seen. But here, in the FACS analysis, MG132 clearly increases the p53-Clover fusion levels; for me again that Western blot and FACS data do not necessarily match.

Figure 3B: In the CCDC6 IP a very small amount of p53 can be found. I don't know how much input lysate compared to amount of lysate for IP is used, but the percentage of p53 found interacting with CCDC6 seems so marginal that is is difficult to explain the effect of KO of CCDC6 in PANC1 cells.

And, the authors called it a 'reciprocal IP' (Suppl Figure 4a) after transfection of V5-tagged CCDC6 into PANC1 cells, but it actually is the same type of IP. Did the authors try to IP p53 and blot for CCDC6? That would be a reciprocal IP.

Figure 3H: how can authors explain that basal levels of USP28 in control and CCDC6-KO cells transfected with control plasmid are more or less the same and not reduced in the CCDC6-KO cells?

Figure 3I: Essentially the whole blot here is of low quality; especially the FBXO42 blot; is deletion of USP28 increasing FBXO42 protein levels, or is it just the quality of the blot?

All in all it seems that FBXO42 is very low expressed in the used cell lines.

Figure 4B: I find it a bit surprising that USP7 is also found in the synthetic viability screen, since it has been shown that USP7 has many more essential targets and KO of p53 only partially rescues the development of USP7-KO mouse embryo's.

Figure 5: the authors nowhere show the efficacy of the guides targeting c16orf72. A

Western blot showing the expression and the reduction upon expressing the guide-RNAs is essential.

Figure 5E: First, here probably parental RPE1 cells have been used, but that is not stated. Second, the authors state 'only a slight increase in p53 levels upon siHUWE1'; I would say none compared to scrambled.

I know HUWE1 is a very huge protein, but the blot of HUWE1 is not convincing. I seem to be able to conclude that siMDM2 and siUSP7 reduces HUWE1 levels?

Figure 5F, in relation to figure 5D. Here the author overexpress both c16orf72 and USP7, and find an interaction. The implication of that is not clear. If they want to make point of this interaction, they should have looked at endogenous proteins.

It is worrying that USP7 apparently was not one of the hits in de Mass-spec experiment of which results are shown in Figure 5D. Also in that experiment c16orf72 was overexpressed, and USP7 is very highly expressed in essentially all cell lines, so do the authors have an explanation?

Suppl. figure 5D is missing

****Referees cross-commenting****

I agree essentially with all comments of Reviewer #2. Especially the major points 3 and 4. The use of more cell lines expressing endogenous mutant p53 is very important.

In addition, I can agree with almost all comments of Reviewer #3. The effects especially of FBXO42 ablation are rather minimal, so relevance is questionable.

2. Significance:

Significance (Required)

Nature and Significance

Compare to existing literature

The topic of the paper is of high interest given the relevance of p53 and its gain-of-function mutants in oncology, and the screens are well executed and clearly presented. In terms of novelty, FBXO42 has been linked to p53-degradation before, and c16orf72 was recently shown to be able to destabilize p53. However, the link between CCDC6 and p53 is novel and of interest, since they are both substrates of USP7 and are both regulators of the cell cycle.

We think the manuscript has potential to add something to the field, but would benefit

greatly from a better understanding of the molecular underpinnings of their newly described mechanisms, as well as the conditions in which the mechanism is active.

Therefore, it might be advisable to shorten the manuscript, and go more in-depth in finding the mechanisms of regulation.

3. How much time do you estimate the authors will need to complete the suggested revisions:

Estimated time to Complete Revisions (Required)

(Decision Recommendation)

Between 1 and 3 months

4. Review Commons values the work of reviewers and encourages them to get credit for their work. Select 'Yes' below to register your reviewing activity at Publons; note that the content of your review will not be visible on Publons.

Reviewer Publons

No

Review #2

1. Evidence, reproducibility and clarity:

Evidence, reproducibility and clarity (Required)

The paper describes several genome-wide CRISPR screens designed to identify regulators of p53 stability. The authors use a system in which p53 levels are marked by mClover expression, using RFP expression to normalise for gene expression changes.

1. The bimodal distribution of p53 expression levels in some reporter cell lines (G245S, R248Q, R248W and R273H) hampers the implementation of a robust readout and makes correct interpretation of the results challenging. While it is possible that the bimodal distribution indicates dynamic changes in p53 levels within one population, it also seems possible that a subclone of these cells have acquired additional alterations affecting p53 stability, and that the authors are screening a mixed population of two intrinsically different cell populations. This would make it difficult to interpret the results of the screen in these cell lines and may be a challenge when trying to identify something that has not already been highlighted on depmap.
2. The coverage of the sgRNA library (200x) is rather low for a negative selection screen, where a coverage of 500x would be more desirable. The FDR threshold is also rather lenient, a more stringent FDR threshold would seem more appropriate and shorten the list of potential hits.
3. Although the study is focused on the regulation of p53 stability, there are no experiments to show that any of the manipulations alter the ubiquitination or degradation (half-life) of p53. The rescue of expression by proteasome inhibition is very modest (Figure 2E), suggesting the loss of expression may not be a reflection of degradation. A role for endogenous FBXO42 and C16orf72 in regulating the ubiquitination and half-life of endogenous p53 should be confirmed
4. Many p53 mutants are used for the initial screens, but very little validation is carried out to show that the apparent differences in factors regulating their stability persists in cells naturally expressing these mutants. For example, FBXO42 is identified as a protein required to maintain the stability of R273H, 248W and R248Q, but not R175H, G245S and R337H. While the authors show an association of CCDC6 and p53 in PANC1 cells (expressing 273H), it would be important to show a panel of R273H, 248W and R248Q expressing tumor cells and the response of p53 to FBXO42 and CCDC6 depletion, compared to similar experiments in a panel of R175H, G245S and R337H expressing tumor cells. Again, it would be important to show that any changes in protein levels are due to changes in protein stability.
5. The potential hits should also be tested in wild type p53 expressing cells to confirm the specificity to mutant p53s.
6. The role of C16orf72 in restraining p53 activity has been reported previously, as has the interaction with HUWE1 (including a new publication PMID: 35776542). The authors suggest an interaction between C16orf72 and USP7, although this should be shown with endogenous proteins. The relative importance of USP7 and HUWE1 binding is not explored. The effect of C16orf72 overexpression in promoting mammary tumors is impressive, although maybe the more interesting question is whether inhibition of C16orf72 expression can limit tumor development in this system.

****Minor comments****

- Figure 1b: The nutlin concentration stated in the methods section is wrong. Should be 10 μ M instead of 10 nM (correct in figure legend).
- Figure 1c: Include results for a mutant that is not regulated by MDM2, such as R175H. Otherwise, as a standalone experiment, this figure doesn't add much.
- Figure 1e/f Legend: Should be FDR <0.5 not >0.5.
- Figure 1h: While an UpSet plot is an elegant way to present unique and overlapping hits between different screens, Venn diagrams might be more 'accessible' to many readers and easier to understand.
- Might be worth stating that mClover is an eGFP variant and can therefore be targeted by eGFP sgRNAs so that it is easier to understand the following:
 - Page 5, paragraph 1: "We used the TKOv3 sgRNA library, which contains [...] 142 control sgRNAs targeting EGFP, LacZ and luciferase"
 - Page 5, paragraph 2: "As expected, sgRNAs targeting p53 and mClover were the most depleted sgRNAs, [...]"
- Figure 6b: y-axis label is missing

2. Significance:

Significance (Required)

This is an interesting concept and the results could provide a useful resource for groups interested in the regulation of p53. The authors chose to focus on candidate genes that could have been identified by looking for the top 30 p53 co-dependent genes on depmap (C16orf72 is #24 in this list and FBXO42 is #28, most of the other genes ranking above are already known as p53 regulators). While this validates the screen, it would have been interesting if the authors had identified and validated new regulators of p53 that were not apparent from previously published work.

3. How much time do you estimate the authors will need to complete the suggested revisions:

Estimated time to Complete Revisions (Required)

(Decision Recommendation)

More than 6 months

4. Review Commons values the work of reviewers and encourages them to get credit for their work. Select 'Yes' below to register your reviewing activity at Publons; note that the content of your review will not be visible on Publons.

Reviewer Publons

No

Review #3

1. Evidence, reproducibility and clarity:

Evidence, reproducibility and clarity (Required)

The manuscript by Lu and coworkers performed genome wide CRISPR screens to search for genes that when knocked out, lead to p53 accumulation or degradation. Wt p53 and a panel of p53 hotspot mutants were chosen as reporter for the screen. The approach reassuringly identified many previously described regulators of p53 degradation, and also found a large set of new hits that many appear to be indirectly affecting p53 level.

A key step of this approach is the follow up functional and mechanistic study of the hits. To this end, the authors chose FBXO42 as a top hit that blocks mutant p53 degradation, and C16orf72 as a top hit that promotes wt/mutant p53 degradation.

Overall the functional data for FBXO42 is disappointing. FBXO42 knockout has quite modest effect on mutant p53 level (~50% reduction). The knockout also showed some effect on p53 mRNA level (~25% reduction), making the determination of mechanism difficult. It does not appear to be a promising targeting for reducing mutant p53 level and gain of function activity in tumor cells.

The C16orf72 finding unfortunately lost some novelty because it was independently identified as a p53 regulator in a recent study using CRISPR screening (PMID: 33660365). However, the repeated identification is reassuring and the current work

provides more convincing functional data, showing C16orf72 knockout increase wt p53 level, inhibits cell proliferation specifically in p53^{+/+} cells, and overexpression of C16orf72 reduce wt p53 level and accelerates progression of a breast tumor mouse model. Their results suggest C16orf72 is a biologically relevant regulator of p53 in cancer development. In order to provide a reasonable amount of new information and set it further apart from the published study, some biochemical analysis looking into the mechanism of C16orf72 will be helpful.

****Specific comments:****

There appears to be a mix up in the figure legend for Fig.1A describing line 1 and 2.

Fig.2. Data for some p53 mutants mentioned in the text cannot be found in the main figure 2D and supplemental figure S3A.

Fig.2 E-F. The effects of FBXO42 and CCDC6 KO on endogenous mutant p53 level is small (~50% decrease). Given that mutant p53 accumulates at high levels, whether a 50% decrease has meaningful effect on its gain of function activities is questionable. The knockouts also caused a ~25% decrease in p53 mRNA (FigS3F) which makes the mechanism quite difficult to investigate further.

Fig.3B. The IP experiment using p53 shRNA and control shRNA should be done by IP of p53 followed by CCDC6 western blot. If CCDC6 IP is used as in the figure, then a CCDC6 shRNA knockdown sample should be compared to control shRNA. The current data does not rule out the possibility that CCDC6 antibody can nonspecifically pull down some p53.

Fig.3D. The in vitro pull down experiment needs specificity controls such as non affected R175H p53 core domain. The data presented would suggest that MBP-FBXO42c captured more than 1:1 molar ratio of R273H core domain, which is unusual for specific binding unless there is aggregation of p53.

To increase the impact of the current study, the authors could provide more mechanism insight on how C16orf72 regulates p53 level, which was also missing in the other published study. For example, addressing whether C16orf72 effect is dependent on MDM2. Does it cooperate with MDM2 to ubiquitinate p53. Does it promote p53 ubiquitination in the absence of MDM2, since it interacts with HUWE1. Does it act by recruiting usp7 to stabilize MDM2.

The manuscript is in a form extremely unfriendly to review, text, figures and legends are all split up at multiple locations, the pdf figures are very sluggish to scroll.

2. Significance:

Significance (Required)

The work is significant in identifying a functionally relevant regulator of p53 stability.

3. How much time do you estimate the authors will need to complete the suggested revisions:

Estimated time to Complete Revisions (Required)

(Decision Recommendation)

Between 3 and 6 months

4. Review Commons values the work of reviewers and encourages them to get credit for their work. Select 'Yes' below to register your reviewing activity at Publons; note that the content of your review will not be visible on Publons.

Reviewer Publons

Yes

Revision Plan

Manuscript number: RC-2022-01563R

Corresponding author(s): Daniel, SCHRAMEK

[The “revision plan” should delineate the revisions that authors intend to carry out in response to the points raised by the referees. It also provides the authors with the opportunity to explain their view of the paper and of the referee reports.]

The document is important for the editors of affiliate journals when they make a first decision on the transferred manuscript. It will also be useful to readers of the reprint and help them to obtain a balanced view of the paper.

*If you wish to submit a full revision, please use our "Full Revision" template. **It is important to use the appropriate template to clearly inform the editors of your intentions.**]*

1. General Statements [optional]

This section is optional. Insert here any general statements you wish to make about the goal of the study or about the reviews.

Our team sincerely thank the reviewers for their constructive reviews of our manuscript, entitled “Genome-wide CRISPR screens identify novel regulators of wild-type and mutant p53 stability”. We were excited that overall the reviewers were unanimously positive about the manuscript: Rev 1: ‘generating a wealth of data’, ‘paper is of high interest’, ‘screens are well executed and clearly presented’; Rev 2: ‘interesting concept and the results could provide a useful resource’; and Rev 3: ‘current work provides more convincing functional data’, *The work is significant in identifying a functionally relevant regulator of p53 stability*. We think that our study and screen designs, and its impact and significance in finding novel regulators of p53 will be interesting in the broad fields of functional genetics and oncology. Our team further agree with the reviewers that this manuscript would benefit from a more detailed molecular underpinning relevant to the novel mechanisms identified. Along with improved readability, we will address all concerns in our revised manuscript. The reviewer comments in their entirety can be found below, followed by our response in **green** font.

Considering that the manuscript was very well received, we believe it makes a strong candidate for publication in Molecular Systems Biology, and it will be of great interest to a wide readership in the fields of p53, functional genetics, and cancer biology.

We hope that you will concur with us that the revision plan detailed below may adequately incorporate the peer reviewers’ suggestions to further strengthen our paper.

2. Description of the planned revisions

Insert here a point-by-point reply that explains what revisions, additional experimentations and analyses are planned to address the points raised by the referees.

While the main scope and strength of this work lies within comprehensive profiling the physiological regulators of 8 of the most common p53 hotspot mutants and as such should be viewed and reviewed as a resource paper, we strive to further enrich the mechanistic insights as suggested by the reviewers. Central to all three reviews, we will focus on the following two areas in our revision, (1) more detailed mechanisms relevant to our newly identified regulators and pathways, and (2) further validations of our hits in different cell lines carrying endogenous mutant p53s.

Please find below our point-to-point responses to the reviewer comments. Experiments proposed for the Revision are indicated by the phrase, "in the revised submission, we will".

Reviewer #1.

Reviewer #1 summary:

In this manuscript by Lu et al., the authors describe some CRISPR screens and protein-protein interaction screens to identify novel regulators of wild-type p53 and mutant p53 function and stability. Besides generating a wealth of data, they discover FBXO42-CCDC6 as positive regulators of the some p53 hotspot mutants, including R273H mutant p53, but not of all p53 mutants tested and also not of wild-type, indicating selectivity. Furthermore, the found C16orf72(TAPR1) as a negative regulator of p53 stability. Mechanistically, the authors claim a direct interaction between FBXO42 and CCDC6 and p53, but the importance of these interactions has not been shown. On the other hand the authors suggest that the FBXO42/CCDC6 regulate p53 via destabilization of USP28, but also the mechanism has not been worked out. For C16orf72, they show that it interacts with USP7, but no relevance of this interaction is shown either.

Response: we sincerely thank the reviewer for the constructive and thorough review. We have incorporated most of the suggestions into our planned revision, with our major focus on the molecular mechanistic follow-up.

Reviewer #1, major points.

1. One very important point for me is that the authors do not show the levels of expression of p53 in the p53-mClover stable cell lines. It is known that overexpressed p53 is usually more stable than endogenous levels of wt-p53. Therefore, I think it is necessary that the authors show the levels of the p53-mClover fusion proteins in the stably transduced cell lines compared to endogenous p53 levels in the parental RPE1 cells and also compared to the endogenous levels of R273H mutant in the PANC-1 cells.

Response: We fully agree that the levels of overexpressed p53s are often more than the endogenous ones, due in part to increased expression and stability. In designing the reporter, we first tried to avoid the stabilisation of p53-GFP due to GFP aggregation by using the monomeric mClover-variant. Further, we titrated the WT and R273H clones (similar to our recent work in PMID: 35439056), to select clones with p53 levels closer to endogenous protein, and exhibiting high dynamic response to Nutlin-3a treatment.

Revision Plan

In the revised submission, we will include Western blotting comparing the levels of p53-mClover (WT and R273H) expression to the endogenous p53s in RPE1 (WT) and PANC1 (R273H) cell lines, in the presence or absence of Nutlin-3a.

2. Also the functionality of the wild-type p53-mClover fusion is questionable, at least not shown. One would expect that the overexpression of a functional wt-p53 in p53-KO cells will affect the survival of the RPE1 cells. In Figure 5A the authors show that depletion of MDM2 or C16ORF72 is toxic for the RPE1 cells in a p53-dependent manner, indicating that elevated levels of p53 cannot be handled by these cells. So, experiment(s) showing that the wt-p53/mClover fusion is functional is needed.

Response: We agree that it will be an important point to benchmark the reporter design. The ectopically expressed WTP53 is often observed to have reduced functionality compared to the endogenous WTP53. The WTP53-reporter line behaves similarly to the RPE1 line (p53-proficient), where both chemical (e.g. Nutlin) or genetic perturbation (e.g. depletion of MDM2/C16orf72) would be toxic in a p53-dependent manner. In line with this data, we have observed that the WTP53-reporter line is able to induce a p53 response as demonstrated by induction of p53-target genes such as p21, which is not observed in p53 null RPE cells, albeit the p21 induction is not as dramatic as in RPE1 cells with endogenous WTP53. Together, these data indicate that our WTP53-reporter is functional albeit with a somewhat reduced activity.

In the revised submission, we will better demonstrate the functionality of the WTP53-mClover fusion by probing WTP53 target (e.g. p21), in the presence and absence of Nutlin. This is also performed as a part of the experiment addressing Point #1 above.

3. A second important point is that the 'verification' of the hits from the screens is only done in one cancer cell line, PANC-1, with mutant p53. I would have like to see at least one other cell line with another p53 mutant endogenously expressed that is also regulated by FBXO42/CCDC6.

Response: we will include validation of the hits (FBXO42, CCDC6) in other 1-2 tumour lines with confirmed R273H endogenous mutation (e.g. MB-MDA-468, etc).

4. For many of the p53-mutants, a bimodal expression is observed. In the FBXO42- and CCDC6-depleted cells, the equilibrium shifts towards more negative cells but the levels in the two populations itself don't change (while for example for USP28 depletion also the right peak shifts further up, Fig S4E). Is there any correlation with the cell cycle and p53 expression? And can the authors exclude that FBXO42 and CCDC6 are involved in cell cycle progression and hereby influence p53 indirectly (by combining PI staining with Clover-p53 for example).

Response: we have indeed observed that the "bimodal" levels in the reporters of several mutants, which are also observed in other studies probing the endogenous p53 level (PMID: 29653964); while the population equilibrium shifts, the location of each peak (as a proxy of the level of p53s) are more stable.

Regarding the relation between p53-level and cell cycle stage, indeed, both the authors in the paper above and we have probed this possibility, but were unable to establish a direct connection.

In the revised submission, we will add flow cytometry analysis of the p53-mClover level, and the cell cycle position using Hoechst 33342 (live-cell permeable DNA staining).

Revision Plan

5. The authors claim that the FBXO42-CCDC6 axis regulates stability specifically some p53-mutants, including R273H-mutant, in a manner involving USP28. But USP28 regulates all forms of p53, not just some mutants version. How can the authors reconcile this apparent contradiction?

Response: we thank the reviewer for this critical observation. From our screen (Supplemental Table 1A), we have indeed noticed a pronounced effects ($|Z \text{ score}| \geq 3$) of FBXO42 on R273H and R248Q stability, and a marginal effect on wild-type p53. Similarly, USP28 had pronounced effects on R273H and R248Q and WTp53.

In the discussion of the paper, we noted that USP28 was shown to regulate p53 levels through distinct mechanisms:

'USP28 was originally implicated as a protective deubiquitinating enzyme counteracting the proteasomal degradation of p53, TP53BP1, CHCK2, and additional proteins⁶⁸⁻⁷¹. USP28 regulates wild-type p53 via TP53BP1-dependent and -independent mechanisms. Concordantly, our data shows that USP28 and TP53BP1 are strong positive regulators of wild-type p53. However, while USP28 was also a strong hit in the mutant R273H p53 screen, TP53BP1 was not, indicating that the effects we see upon loss of USP28 on R273H p53 are independent of TP53BP1.'

Together, this indicates that the R273H-mutant is regulated by a FBXO42-CCDC6-USP28 axis while wild-type p53 is regulated mainly via a USP28-TP53BP1 axis. We will attempt to address and discuss it in the revision.

6. On a similar note, the authors show that FBXO42 and CCDC6 interact with p53, but not USP28. Do FBXO42 and CCDC6 interact with each other and with USP28? And is the interaction with p53 specific for the R273H version? This part of the mechanism is very poorly defined and the Co-IPs are not very convincing or relevant for the proposed model.

Response: This comment will be more extensively addressed in the revision. We have indeed observed the interaction between FBXO42 and CCDC6 (via BioID and APMS); however, we failed to recover USP28 as an interactor of either FBXO42 or CCDC6. The interaction between CCDC6/FBXO42 is not specific to R273H; although we were able to IP endogenous R273H with CCDC6 in PANC-1 line, the WTp53 (as in HEK 293 TRex BioID line) was also picked up in the BioID preys of CCDC6/FBXO42. In addition, we have new data to show that FBXO42 directly interacts with WTp53.

In the revised submission, we will improve the molecular underpinning of the FBXO42-CCDC6-USP28-p53 axis we propose. We will specifically address the following.

(1.1.) Biochemically, further support that CCDC6 and FBXO42 regulate p53 via regulating USP28 stability: We will address this by established biochemical assays, e.g. cycloheximide-chase/MG132 experiment. While USP28 is an established WTp53 regulator, little is known about the mechanism, and the “upstream” regulation of USP28; we will attempt to fill this gap:

(1.2.) And to an unbiased systematic approach, how R273H interactome changes upon the loss of CCDC6 or FBXO42.

We will perform R273H-BioID upon loss of CCDC6 and FBXO42 and USP28.

(1.3.) Furthermore, we will specifically exam the interaction of USP28-p53R273H with or without the genetic perturbation of FBXO42/CCDC6.

Revision Plan

Through these efforts, we hope to gain further mechanistic insights into this regulatory axis, but hope that the editors and reviewers will agree that a fully annotated mechanistic understanding is probably beyond the scope of this paper.

Reviewer #1, minor points.

7. The mechanisms of p53 regulation may vary greatly in different cell lines. Can the authors discuss why they choose to do the screen with different mutants, rather than with different cell lines expressing these same mutant endogenously?

Response: While it is certainly very interesting to assess how WT and mutant p53 is regulated in different cell lines, such an approach is confounded by the ‘genetic make-up’ of the respective tested cell lines. For example, TP53BP1 might be a regulator in one cell line but not in another for the simple reason that the later cell line harbors a TP53BP1 deletion or mutation or expression levels. In addition, while working with endogenous p53 mutations certainly has many advantages, comparing different mutants in different cell lines is again very much confounded by the ‘genetic make-up’ of the respective tested cell lines.

Our focus was slightly different, and we wanted to set out and specifically ask what the difference between p53 hotspot mutations are. Are they all the same or are there differences and importantly, are there differences between mutants and WT p53 and this can only be achieved when working in the same cellular background. In designing the screen, we have thus tried to optimise the inclusion of different hotspot mutants in an isogenic screening system. As such, we first depleted the endogenous WTp53 to minimise its interference and built the current isogenic system in the non-transformed RPE1 (“normal”) line.

However, as discussed above, we agree that the screen results will be validated in more cell lines carrying respective endogenous mutants.

8. Figure 1: Typo in the legends : Nultin ipv Nutlin

Response: We apologise for the typos. This is addressed in the current submission, along with improved figure legends to improve readability.

9. Figure 1b,1c : Show basal and Nutlin-3 induced MDM2 levels and in the overexpression cell lines; if WT-p53 is functional, MDM2 levels should be higher in WT-transduced cells compared to control or mt-p53 expressing cells.

Response: In the revised submission, we will include Western blotting probing MDM2 levels (antibody permitting); this is a part of the experiment proposed for Points 1 and 2.

10. Authors should explain which they name USP7 a negative regulator of p53, since it is supposed to de-ubiquitinate p53?!

Response: The effects of USP7 on WTp53 have indeed been difficult to elucidate (by Prof. Vogelstein PMID: 15118411, and PMID: 15058298, and seemingly opposite by Prof. Gu Wei, PMID: 15053880, and PMID: 11923872). However, consistent with Prof. Vogelstein group, the inhibition of USP7 (either by inhibitor or genetically via CRISPR in our studies), has resulted in elevated p53 level.

11. Figure 2E: the effect of MG132 on p53 seems to be very minimal on this Western blot; it would need quantification to be convincing...Quality of the blot is also not great.

The fact that in control cells the levels of p53 R273H are not affected by MG132 treatment fits with Suppl Figure 2E, indicating that the proteasome has no effect on p53 R273H.

Revision Plan

Response: We indeed noticed that while the proteasome pathway is largely implicated in the WTp53 screen, it has much reduced effects on R273H. Interestingly, the treatment of MG 132 also has limited effects using PANC-1 line (with endogenous R273H). We will repeat this experiment and provide quantifications and modify the text accordingly.

12. Suppl figure 3b, 3c, 3d:

Somehow, I have the feeling that the results from the western blots and the FACS do not match fully, although not all the time-points are shown in the various experiments.

For example, the FACS analysis (3b) suggests that in control-transduced cells after 16hr p53 is still increased. However, that is not clear at all in the Western blot (3c)

Is Suppl Figure 3d the quantification of 3c experiment? If so, in the blot also the 24 hrs should be shown.

The blot shown in Suppl Figure 3c suggests that CCDC6 expression increased upon irradiation. Do the authors agree with that? Would that explain why depletion of CCDC6 has more effect upon irradiation?

Suppl Figure S3E: if I am right, this is essentially the same type of experiment as shown in figure 2e, but analysis of p53-expression by Western blot. In that blot no real effect of MG132 on p53 levels could be seen. But here, in the FACS analysis, MG132 clearly increases the p53-Clover fusion levels; for me again that Western blot and FACS data do not necessarily match.

Response: We apologise for the confusion. In the revised submission, we will improve the figure legends for better readability. Furthermore, in anticipation to the multiple cell lines involved in the revision, we will also clarify the cell lines in the figure.

With regards to the difference between the flow cytometry and WB data, we have generally observed the flow cytometry bimodal shifting to be more sensitive than the WB, e.g. a 50% shift in population (FACS) is reflected by a 15% reduction in WB (which may be partially explained as WB is a measurement across the cell population and FACS determines the p53-GFP levels of every cell and thus the shift of cells between peaks). Similarly, we noticed flow-cytometry based quantification by antibody staining the endogenous p53 yielded similar sensitivity (PMID: 29653964). As such, we will ensure the validation of hits is performed in two modes. For WB experiment, we will do so in two cell lines carrying the endogenous mutants as suggested by Reviewers #1 and 2.

13. Figure 3B: In the CCDC6 IP a very small amount of p53 can be found. I don't know how much input lysate compared to amount of lysate for IP is used, but the percentage of p53 found interacting with CCDC6 seems so marginal that is difficult to explain the effect of KO of CCDC6 in PANC1 cells.

And, the authors called it a 'reciprocal IP' (Suppl Figure 4a) after transfection of V5-tagged CCDC6 into PANC1 cells, but it actually is the same type of IP. Did the authors try to IP p53 and blot for CCDC6? That would be a reciprocal IP.

Response: We apologise for the confusion. In the revised submission, we will specify the portion of the lysates used for pre-IP (5% lysate) and IP (1 mg). As for the IP, we will also include the true reciprocal IP (IP p53, and blot for CCDC6).

14. Figure 3H: how can authors explain that basal levels of USP28 in control and CCDC6-KO cells transfected with control plasmid are more or less the same and not reduced in the CCDC6-KO cells?

We will provide a better blot and quantification for this observation. In the current Fig 3H, the CCDC6-KO lane is slightly overladed as seen by the H3 loading control.

15. Figure 3I: Essentially the whole blot here is of low quality; especially the FBXO42 blot; is deletion of USP28 increasing FBXO42 protein levels, or is it just the quality of the blot?

All in all it seems that FBXO42 is very low expressed in the used cell lines.

Revision Plan

Response: We apologise for the confusion. In the revised submission, we will repeat and try to include higher quality WB, with more optimised condition for using the FBXO42 antibody. FBXO42 messenger level is readily detected using qRT.

16. Figure 4B: I find it a bit surprising that USP7 is also found in the synthetic viability screen, since it has been shown that USP7 has many more essential targets and KO of p53 only partially rescues the development of USP7-KO mouse embryos.

Response: We thank the reviewer for this critical observation. While the double p53-USP7 knockout line is viable, we acknowledge that it is amongst the top scored hits due to the large differential viabilities between WT and p53-null lines. In the revised submission, we will further clarify the screen analysis and the associated interpretation.

17. Figure 5: the authors nowhere show the efficacy of the guides targeting c16orf72. A Western blot showing the expression and the reduction upon expressing the guide-RNAs is essential.

Response: We thank the Reviewer for this suggestion. The efficacy of each guide has been verified using ICE (at the genomic level), and in the revised submission, we will include this critical information as part of the Figure S2F.

18. Figure 5E: First, here probably parental RPE1 cells have been used, but that is not stated. Second, the authors state 'only a slight increase in p53 levels upon siHUWE1'; I would say none compared to scrambled.

I know HUWE1 is a very huge protein, but the blot of HUWE1 is not convincing. I seem to be able to conclude that siMDM2 and siUSP7 reduces HUWE1 levels?

Response: We apologise for the confusion. In the revised submission, we will be specific of the cell line information on the figure, to improve the readability.

We agree with the reviewers that assessment of large protein by WB is often difficult but given that this band almost completely disappears upon HUWE1 knock-down, strongly argues that we are indeed assessing the endogenous HUWE1. We also agree that it is an interesting observation that the levels of HUWE1 seem to be slightly reduced upon knock-down of MDM2 and USP7. We will repeat this experiments and provide quantitative data for HUWE1 and p53. Of note, in the screen, HUWE1 also scored as a negative regulator of wt-p53 and did not quite reach statistical significance for the p53 mutants.

Regarding the relationship between C16orf72 and HUWE1, a newly published work (PMID: 35776542) seems to suggest that siHUWE1 has resulted in an increased C16orf72 level (termed HAPSTR1 in the paper), while siC16orf72 seemed to have no effect on HUWE1 level, although the stability of such a large protein by WB is often difficult to conclude.

19. Figure 5F, in relation to figure 5D. Here the author overexpress both c16orf72 and USP7, and find an interaction. The implication of that is not clear. If they want to make point of this interaction, they should have looked at endogenous proteins.

Response: We acknowledge the many concerns associated with coIP with ectopically, and especially overexpressed proteins in large quantity. In the revised submission, we will attempt to perform endogenous-based IP experiment (antibody permitting).

20. It is worrying that USP7 apparently was not one of the hits in the Mass-spec experiment of which results are shown in Figure 5D. Also in that experiment c16orf72 was overexpressed, and USP7 is very highly expressed in essentially all cell lines, so do the authors have an explanation?

Response: We indeed acknowledge this discrepancy. In the revised submission, we will attempt the coIP/IP using endogenous proteins (antibody permitting, or at least using endogenous target for one of the

Revision Plan

two partners). We also acknowledge that the limitation associated with the APMS for the detection of interactors.

21. Suppl. figure 5D is missing

Response: We apologise for the confusion. The Figure S5D was inconveniently placed at the top of the figure panel due to space limitation. In the revised submission, we will address this as a part of the overall readability improvement.

Reviewer #1, Significance.

The topic of the **paper is of high interest** given the relevance of p53 and its gain-of-function mutants in oncology, and the **screens are well executed and clearly presented**. In terms of novelty, FBXO42 has been linked to p53-degradation before, and c16orf72 was recently shown to be able to destabilize p53. However, the link between CCDC6 and p53 is novel and of interest, since they are both substrates of USP7 and are both regulators of the cell cycle.

We think the manuscript has potential to add something to the field, but would benefit greatly from a better understanding of the molecular underpinnings of their newly described mechanisms, as well as the conditions in which the mechanism is active.

Therefore, it might be advisable to shorten the manuscript, and go more in-depth in finding the mechanisms of regulation.

Response: We sincerely thank the reviewer for all the constructive critiques. We will incorporate them in to our revision.

Revision Plan

Reviewer #2.

Reviewer #2 summary:

The paper describes several genome-wide CRISPR screens designed to identify regulators of p53 stability. The authors use a system in which p53 levels are marked by mClover expression, using RFP expression to normalise for gene expression changes.

Reviewer #2, major points.

1. The bimodal distribution of p53 expression levels in some reporter cell lines (G245S, R248Q, R248W and R273H) hampers the implementation of a robust readout and makes correct interpretation of the results challenging. While it is possible that the bimodal distribution indicates dynamic changes in p53 levels within one population, it also seems possible that a subclone of these cells have acquired additional alterations affecting p53 stability, and that the authors are screening a mixed population of two intrinsically different cell populations. This would make it difficult to interpret the results of the screen in these cell lines and may be a challenge when trying to identify something that has not already been highlighted on depmap.

Response: We thank the reviewer for this critical observation. We strongly believe that this bimodal distribution is actually an inherent property of the p53 mutants in these cells for the following reasons: (1) The observation of the similar bimodal appearance in cell lines harbouring corresponding endogenous mutant p53s (PMID: 29653964) suggest that these two populations are of biological significance. (2) We have established 5-10 clonal lines each from the G245S, R248Q, R248W and R273H p53 reporter line and all of them exhibit a bimodal distribution, making it very unlikely that these populations are all through stochastic outgrowth of sub-populations with spontaneous mutations/alterations. (3) The bimodal distribution is stable over several months to years in culture. If it were a spontaneous mutations giving rise to a clone with higher mutant p53 levels, we would likely expect that over time this clone takes over the population. (4) We observed that such a pool of bimodal cells could be “synchronised” (e.g. by Nutlin, or MDM2 knockout) to one population, and later return to and repopulate the other (e.g. Nutlin washoff, Figure 1B). (5) When we sort out a single cells from the upper or the lower peak, expand them, we obtain again populations of cells with the same bimodal distribution, indicating that this is a dynamic process. Thus, we believe that these two populations were rather intrinsic, such that a cell in the population may assume both states.

We also acknowledge the difficulties of screening using a bimodal population; however, we took advantage of these “bimodal” mutants and using FACS assessed the state of a single cell in relation to a genetic perturbation. Each guide has an equal chance of entering a cell that belongs to one of the two populations. If a gene knock-out really affects p53 levels, the cells with the respective guides enrich in one and deplete in the other population and the analysis comparing the guide abundances from these two peaks ensures the experiment are being perfectly internally controlled.

While many of the top scored hits from the resulting screens are known regulators, it is critical that we validate our hits in an independent system, such as the cell lines harbouring endogenous p53 mutations, echoed by both Reviewers #1 and 2.

2. The coverage of the sgRNA library (200x) is rather low for a negative selection screen, where a coverage of 500x would be more desirable. The FDR threshold is also rather lenient, a more stringent FDR threshold would seem more appropriate and shorten the list of potential hits.

Response: We thank the reviewer for this constructive suggestion. A higher coverage, along with a more stringent FDR, will ensure an even stronger confidence for the remaining individual hits. The present reporter-based enrichment screen and the synthetical viability drop-out screen used four guides per gene, and with 200x coverage for each guide.

Revision Plan

In determining the coverage, we tried to reference recent successful screenings and apply earlier titration result for the 200x coverage (e.g. PMID: 26627737, PMID: 33465779, and reviewed in Nat Rev Methods Primers 2, 8 (2022). <https://doi.org/10.1038/s43586-021-00093-4>). While the threshold of FDR was often arbitrary, we fully agree that a more stringent FDR, which results in shortened hits list, may further boost the confidence of the hits, though also at the cost of losing potential hits due to collateral effects (e.g. guide efficiency).

We agree with this reviewer that a higher FDR, esp. at the hits that result in p53 stabilization, would make sense as any gene whose loss causes cellular or genotoxic stress, would likely lead at least in part to p53 stabilization. In the revised submission, we will adjust the FDR accordingly.

3. Although the study is focused on the regulation of p53 stability, there are no experiments to show that any of the manipulations alter the ubiquitination or degradation (half-life) of p53. The rescue of expression by proteasome inhibition is very modest (Figure 2E), suggesting the loss of expression may not be a reflection of degradation. A role for endogenous FBXO42 and C16orf72 in regulating the ubiquitination and half-life of endogenous p53 should be confirmed

Response: We thank the reviewer for this suggestion. In the revised submission, we will monitor the ubiquitination status and also degradation (cycloheximide-chase) experiments for R273H cells, with or without the genetic alteration of CCDC6/FBXO42/C16orf72.

4. Many p53 mutants are used for the initial screens, but very little validation is carried out to show that the apparent differences in factors regulating their stability persists in cells naturally expressing these mutants. For example, FBXO42 is identified as a protein required to maintain the stability of R273H, 248W and R248Q, but not R175H, G245S and R337H. While the authors show an association of CCDC6 and p53 in PANC1 cells (expressing 273H), it would be important to show a panel of R273H, 248W and R248Q expressing tumor cells and the response of p53 to FBXO42 and CCDC6 depletion, compared to similar experiments in a panel of R175H, G245S and R337H expressing tumor cells. Again, it would be important to show that any changes in protein levels are due to changes in protein stability.

Response: We thank the reviewer for this suggestion. In the revised submission, we will include validations in more cell lines carrying endogenous mutant p53s, with a focus on the R273H mutant. We will also try to involve a line with an endogenous p53 mutation that does not respond to FBXO42/CCDC6 alteration.

5. The potential hits should also be tested in wild type p53 expressing cells to confirm the specificity to mutant p53s.

Response: In the revised submission, we will include WB for WT lines (e.g. RPE1) upon genetic alteration of CCDC6 and FBXO42. This was already performed for C16orf72 (Figure 6D).

6. (6A) The role of C16orf72 in restraining p53 activity has been reported previously, as has the interaction with HUWE1 (including a new publication PMID: 35776542). The authors suggest an interaction between C16orf72 and USP7, although this should be shown with endogenous proteins. The relative importance of USP7 and HUWE1 binding is not explored. (6B) The effect of C16orf72 overexpression in promoting mammary tumors is impressive, although maybe the more interesting question is whether inhibition of C16orf72 expression can limit tumor development in this system.

Response to 6A: we are excited about the independent observations by other group(s) confirming similar results! As a part of our improvement for mechanistic work-up, in the revised submission, we will attempt to address, whether C16orf72' regulation of p53 is dependent on USP7 and/or HUWE1, or other known E3s, such as MDM2.

(1) Whether the interaction of C16orf72 and HUWE1 or USP7 is required for the C16orf72 regulation of p53. Specifically, for example, we will perform epistasis experiments to test USP7' or HUWE1' ability to

Revision Plan

rescue the p53 levels in reporters upon $\Delta C16orf72$. Due to the toxicity/lethality in WTP53 lines induced by the loss of C16orf72, we intend to test using R273H-reporter, or RPE1-line with $\Delta CDKN1A$ (p21) that is a synthetic viable rescue for $\Delta C16orf72$.

(2) In the revised submission, we will attempt to perform endogenous-based C16orf72-USP7 IP experiment (antibody permitting).

6B. The effect of C16orf72 overexpression in promoting mammary tumors is impressive, although maybe the more interesting question is whether inhibition of C16orf72 expression can limit tumor development in this system.

Response: We are also equally excited about the *in vivo* result supporting the idea that C16orf72 overexpression in tumour-prone mice (Pik3ca^{H1047R}) mice harbouring WTP53 may accelerate tumour formations. In the revised submission, we will further support that this effect is specific to WTP53/C16orf72, by including data of the control cohort with p53-null background (LSL-Pi3kH1047R; p53Flox/Flox).

In regard to the effects of C16orf72-depletion in controlling tumour growth - we agree that this would be a very exciting avenue. Conditional C16orf72 mice are being made at the moment and these mice will allow us to comprehensively address this question. However, it will take several more months to generate and validate this line, and then another 2 breeding rounds to generate homozygous C16orf72^{fl/fl}; Pik3ca^{H1047R} mice. In addition, the long time required to form tumours in the control mice with WTP53 (~250 days), it becomes not feasible for us to test whether the inhibition of C16orf72 could limit the tumour development, given the revision timeline. As such we respectfully believe that this would be beyond the scope of this manuscript.

Reviewer #2, Minor comments.

7. Figure 1b: The nutlin concentration stated in the methods section is wrong. Should be 10 μ M instead of 10 nM (correct in figure legend).

Figure 6b: y-axis label is missing.

Figure 1e/f Legend: Should be FDR <0.5 not >0.5.

Response: We apologise for typos. The current submission has incorporated the corrections.

8. Figure 1c: Include results for a mutant that is not regulated by MDM2, such as R175H. Otherwise, as a standalone experiment, this figure doesn't add much.

Response: We thank the reviewer for this suggestion. In the revised submission, we will include R175H/R337H.

9. Figure 1h: While an UpSet plot is an elegant way to present unique and overlapping hits between different screens, Venn diagrams might be more 'accessible' to many readers and easier to understand.

Response: We thank the reviewer for this feedback. The choice of UpSet blot was largely motivated by the different categories involved, which made the area representation and the intersection of the conventional Venn diagram no longer feasible.

In the revised submission, we will improve our figure legend for the UpSet blot, to improve the readability.

10. Might be worth stating that mClover is an eGFP variant and can therefore be targeted by eGFP sgRNAs so that it is easier to understand the following:

o Page 5, paragraph 1: "We used the TKOv3 sgRNA library, which contains [...] 142 control sgRNAs targeting EGFP, LacZ and luciferase"

Revision Plan

o Page 5, paragraph 2: "As expected, sgRNAs targeting p53 and mClover were the most depleted sgRNAs, [...]"

Response: We thank the reviewer for this suggestion. We believe this will also improve the readability and have incorporated this into our current submission.

Reviewer #2, Significance.

Reviewer #2 (Significance (Required)):

This is an **interesting concept and the results could provide a useful resource** for groups interested in the regulation of p53. The authors chose to focus on candidate genes that could have been identified by looking for the top 30 p53 co-dependent genes on depmap (C16orf72 is #24 in this list and FBXO42 is #28, most of the other genes ranking above are already known as p53 regulators). While this validates the screen, it would have been interesting if the authors had identified and validated new regulators of p53 that were not apparent from previously published work.

Response: We thank the reviewer for all the thorough and constructive comments! In relation to the DepMap dataset, we are excited that many of the top hits from our screens are indeed top Wtp53-correlators/anti-correlators (e.g. MDM2, USP28)!

While the DepMap dataset used cell fitness/viability to construct the genetic relation score, this assay may not effectively rule out the many regulators that could otherwise elicit their regulation of p53 via regulating the general cell response to cell cycle, stress, etc. In our screen systems (i.e. protein stability and synthetic viability screens), we attempted to focus on the regulators of p53-stability (post-translational), and further coupled it with the synthetic viability screens to concentrate on hits that have a more direct role in p53 regulation (e.g. MDM2, C16orf72).

One other difficulty to fully couple our screens to the DepMap dataset is due to the limited cell lines harbouring endogenous mutant p53s, e.g. R337H. This may also contribute to the uniqueness of the identified R337H-reporter specific hits (where cell lines harbouring R337H have not yet been included in the DepMap dataset), e.g. several Aminoacyl tRNA synthetases (SARS, YARS, etc) were identified as R337H unique regulators and subsequently verified using different guides in the reporter line, but could not be obtained via DepMap.

We largely see this paper as a resource for the p53 field and would like to publish it as soon as possible. In fact, when we started working on C16orf72 or CCDC6/FBXO42, these hits were not known for their ability to regulate p53. We will work up several other hits, but this would be beyond the scope of this paper and the first author's Ph.D. thesis that needs to be completed under a timeline.

Revision Plan

Reviewer #3.

Reviewer #3 summary:

The manuscript by Lu and coworkers performed genome wide CRISPR screens to search for genes that when knocked out, lead to p53 accumulation or degradation. Wt p53 and a panel of p53 hotspot mutants were chosen as reporter for the screen. The approach reassuringly identified many previously described regulators of p53 degradation, and also found a large set of new hits that many appear to be indirectly affecting p53 level.

A key step of this approach is the follow up functional and mechanistic study of the hits. To this end, the authors chose FBXO42 as a top hit that blocks mutant p53 degradation, and C16orf72 as a top hit that promotes wt/mutant p53 degradation.

Overall the functional data for FBXO42 is disappointing. FBXO42 knockout has quite modest effect on mutant p53 level (~50% reduction). The knockout also showed some effect on p53 mRNA level (~25% reduction), making the determination of mechanism difficult. It does not appear to be a promising targeting for reducing mutant p53 level and gain of function activity in tumor cells.

We thank the reviewer for this constructive comment! We will address this in the revision, as proposed in Point #3.

The C16orf72 finding unfortunately lost some novelty because it was independently identified as a p53 regulator in a recent study using CRISPR screening (PMID: 33660365). However, the repeated identification is reassuring and the **current work provides more convincing functional data**, showing C16orf72 knockout increase wt p53 level, inhibits cell proliferation specifically in p53^{+/+} cells, and overexpression of C16orf72 reduce wt p53 level and accelerates progression of a breast tumor mouse model. Their results suggest C16orf72 is a biologically relevant regulator of p53 in cancer development. In order to provide a reasonable amount of new information and set it further apart from the published study, some biochemical analysis looking into the mechanism of C16orf72 will be helpful.

Reviewer #3 Major and Minor comments:

Specific comments:

1. There appears to be a mix up in the figure legend for Fig.1A describing line 1 and 2.

Response: We sincerely apologise for the mix up in the figure legend! In the current submission, this has been fixed.

2. Fig.2. Data for some p53 mutants mentioned in the text cannot be found in the main figure 2D and supplemental figure S3A.

Response: We apologise for having not included the R175H and R337H mutants in Supplemental Figure S3A. In the revised version, we will include these two mutants.

3. Fig.2 E-F. The effects of FBXO42 and CCDC6 KO on endogenous mutant p53 level is small (~50% decrease). Given that mutant p53 accumulates at high levels, whether a 50% decrease has meaningful effect on its gain of function activities is questionable. The knockouts also caused a ~25% decrease in p53 mRNA (FigS3F) which makes the mechanism quite difficult to investigate further.

Response: We agree with the reviewer that the current data makes it difficult to conclude the mechanism. Given the design of our reporter, we still believe that the regulations could largely be at the post-translational level. In our revised version, we plan to exam the ubiquitination status of p53 upon losses of CCDC6/FBXO42, and also monitor the p53 degradation via cycloheximide chase.

Revision Plan

To further address whether this reduced level of mutp53 has biological impacts, we plan to test it in the tumour cell context. Given the difference in migration capability observed between PANC-1 and PANC-1- Δ p53 line (e.g. PMID: 35439056), we plan to also evaluate the migration pattern of PANC-1, with the presence and absence of FBXO42/CCDC6 (controlled by similar FBXO42/CCDC6 loss in PANC-1- Δ p53 background). Furthermore, in tissue culture, although there is only marginal to no difference in cell growth rate between many mutant p53 lines (e.g. PANC-1) and their Δ p53 line, we plan to test whether a reduced serum or nutrient level could exacerbate the difference, and hence further be used to monitor the difference resulted from the loss of FBXO42/CCDC6.

4. Fig.3B. The IP experiment using p53 shRNA and control shRNA should be done by IP of p53 followed by CCDC6 western blot. If CCDC6 IP is used as in the figure, then a CCDC6 shRNA knockdown sample should be compared to control shRNA. The current data does not rule out the possibility that CCDC6 antibody can nonspecifically pull down some p53.

Response: We apologise for the confusion. In the revised version, we will include the proper reciprocal IP, with IP of endogenous p53 (R273H) followed by blotting of CCDC6.

5. Fig.3D. The in vitro pull down experiment needs specificity controls such as non affected R175H p53 core domain. The data presented would suggest that MBP-FBXO42c captured more than 1:1 molar ratio of R273H core domain, which is unusual for specific binding unless there is aggregation of p53.

Response: We thank the reviewer for this constructive comment! In the revised version, we will incorporate this, by repeating the *in vitro* pull-down assay including a non-p53 control protein.

6. To increase the impact of the current study, the authors could provide more mechanism insight on how C16orf72 regulates p53 level, which was also missing in the other published study. For example, addressing whether C16orf72 effect is dependent on MDM2. Does it cooperate with MDM2 to ubiquitinate p53. Does it promote p53 ubiquitination in the absence of MDM2, since it interacts with HUWE1. Does it act by recruiting usp7 to stabilize MDM2.

Response: we thank the reviewer for this very constructive and thorough comment! In our revised version, we will attempt these assays and incorporate them into the submission.

Together with our response to Reviewer #2, Point #6, in the revised submission, we will attempt to address if C16orf72 regulation of p53 is dependent on MDM2 or HUWE1.

(1) Whether the interaction of C16orf72 and HUWE1, or C16orf72 and USP7 is required for the C16orf72 regulation of p53. Specifically, for example, we will perform epistasis experiments to test HUWE1' or USP7's ability to rescue the p53 levels in reporters upon the loss of C16orf72 (Δ C16orf72). Due to the toxicity/lethality in WTp53 lines induced by the loss of C16orf72, we intend to test using the R273H-reporter, or RPE1-line with Δ CDKN1A (p21) that is a synthetic viable rescue for Δ C16orf72.

(2) Whether C16orf72 dependent upon or cooperate with MDM2 in regulating p53.

We will first probe whether C16orf72 overexpression increased the p53 ubiquitination, and then decide whether overexpression of C16orf72 has additive effects to MDM2 overexpression in regulating p53 levels.

We previously observed that overexpressing C16orf72 could not rescue the R273H level resulted from losing MDM2 (using flow-cytometry in R273H-reporter- Δ MDM2), and as such, we plan to test the C16orf72-MDM2 relation in the MDM2-proficient context.

7. The manuscript is in a form extremely unfriendly to review, text, figures and legends are all split up at multiple locations, the pdf figures are very sluggish to scroll.

Response: We sincerely apologise for the inconvenience. In the current submission, we have split the submission into three separate files, (1) main text, (2) main figures, and (3) supplemental figures, along

Revision Plan

with (4) supplemental tables as individual EXCELS. We will also reduce the resolution of a few images, so the overall higher resolution is retained, while still fitting into the file size limit.

Reviewer #3 (Significance (Required)):

The work is significant in identifying a functionally relevant regulator of p53 stability.

Response: we thank the reviewer again for the very constructive feedback!

3. Description of the revisions that have already been incorporated in the transferred manuscript

Please insert a point-by-point reply describing the revisions that were already carried out and included in the transferred manuscript. If no revisions have been carried out yet, please leave this section empty.

We have revised the manuscripts largely aiming at readability improvement, e.g. typos fix, improved figure legends for clarity, etc.

Specifically, the following points have been revised in the current version. It is also mentioned in the point-to-point response found in Section #2, to each comment.

Reviewer #1: Point 8.

Reviewer #2: Points #7 and 10.

Reviewer #3: Points #1 and 7.

4. Description of analyses that authors prefer not to carry out

Please include a point-by-point response explaining why some of the requested data or additional analyses might not be necessary or cannot be provided within the scope of a revision. This can be due to time or resource limitations or in case of disagreement about the necessity of such additional data given the scope of the study. Please leave empty if not applicable.

We have incorporated almost all suggestions from the reviewers, with the only exceptions of the following.

1. Reviewer #2, Point #1.

“.....it would be important to show a panel of R273H, 248W and R248Q expressing tumor cells and the response of p53 to FBXO42 and CCDC6 depletion, compared to similar experiments in a panel of R175H, G245S and R337H expressing tumor cells.....”

Response: We will attempt to test the effects of FBXO42/CCDC6 depletions in cell lines endogenously carrying these mutants. However, due to the potential availability of cell lines carrying specific mutant (e.g. R337H, G245S), we may focus first on the R273H, R248Q/W, R175H mutants, covering both the FBXO42/CCDC6-responsive and non-responsive groups.

2. Reviewer #2, Point #6.

“The effect of C16orf72 overexpression in promoting mammary tumors is impressive, although maybe the more interesting question is whether inhibition of C16orf72 expression can limit tumor development in this system.”

Revision Plan

Response: Conditional C16orf72 mice are being made at the moment and these mice will allow us to comprehensively address this question. However, it will take several more months to generate and validate this line, and then another 2 breeding rounds to generate homozygous C16orf72^{fl/fl}; Pik3ca^{H1047R} mice. In addition, the long time required to form tumours in the control mice with WTp53 (~250 days), it becomes not feasible for us to test whether the inhibition of C16orf72 could limit the tumour development, given the revision timeline.

However, to strengthen that the C16orf72's tumour-promoting effect is specific in the WTp53 context (LSL-Pi3kH1047R; WTp53), we will also include data of the control cohort with p53-null background (LSL-Pi3kH1047R; p53Flox/Flox), where the overexpression of C16orf72 has not accelerated the tumour formation.

16th Sep 2022

RE: MSB-2022-11350, Genome-wide CRISPR screens identify novel regulators of wild-type and mutant p53 stability

Dear Daniel,

Thank you for submitting your manuscript to Molecular Systems Biology. I have now read the manuscript and your revision plan described in the point-by-point response to the comments of the Review Commons reviewers. I would like to invite you to submit your revised manuscript to Molecular Systems Biology.

Overall, we agree with the reviewers that including validations in additional cell lines and including analyses enhancing the level of mechanistic insight provided by the study would be required. We think that your revision plan sounds promising for addressing the issues raised by the reviewers.

The eventual acceptance of the study will depend on how well the issues raised by the referees have been addressed. As you might already know, our editorial policy allows in principle a single round of major revision, and it therefore is essential to provide responses to the reviewers' comments that are as complete as possible.

To speed up the evaluation of your revised manuscript we would also ask you to address the following editorial points:

- Please include 5 keywords.

- We have replaced Supplementary Information by the Expanded View (EV format). In this case, all additional figures can be included in a PDF called Appendix. Appendix Figures should be labeled and called out as: "Appendix Figure S1, Appendix Figure S2, ... etc.". Each Appendix Figure legend should be provided below the corresponding Figure in the Appendix. Please include a Table of Contents in the beginning of the Appendix. For detailed instructions regarding expanded view please refer to our Author Guidelines: .

- Supplementary Tables 1-7 should be provided as Datasets EV1-EV7. Please include in each .xls file a separate tab with the description of the dataset.

- Please provide a "standfirst text" summarizing the study in one or two sentences (approximately 250 characters), three to four "bullet points" highlighting the main findings and a "synopsis image" (550px width and max 400px height, jpeg format) to highlight the paper on our homepage.

- All Materials and Methods need to be described in the main text. We would encourage you to use 'Structured Methods', our new Materials and Methods format. According to this format, the Material and Methods section should include a Reagents and Tools Table (listing key reagents, experimental models, software and relevant equipment and including their sources and relevant identifiers) followed by a Methods and Protocols section in which we encourage the authors to describe their methods using a step-by-step protocol format with bullet points, to facilitate the adoption of the methodologies across labs. More information on how to adhere to this format as well as downloadable templates (.doc or .xls) for the Reagents and Tools Table can be found in our author guidelines: . An example of a Method paper with Structured Methods can be found here:

- Please include a Data availability section describing how the data and code have been made available. This section needs to be formatted according to the example below:

The datasets and computer code produced in this study are available in the following databases:

- Chip-Seq data: Gene Expression Omnibus GSE46748 (<https://www.ncbi.nlm.nih.gov/geo/query/acc.cgi?acc=GSE46748>)

- Modeling computer scripts: GitHub (<https://github.com/SysBioChalmers/GECKO/releases/tag/v1.0>)

- [data type]: [full name of the resource] [accession number/identifier] ([doi or URL or identifiers.org/DATABASE:ACCESSION])

- Please include a "Disclosure Statement & Competing Interests" in the main text.

- The References should be formatted according to the Molecular Systems Biology reference style.

- For data quantification: please specify the name of the statistical test used to generate error bars and P values, the number (n) of independent experiments (specify technical or biological replicates) underlying each data point and the test used to calculate p-values in each figure legend. The figure legends should contain a basic description of n, P and the test applied. Graphs must include a description of the bars and the error bars (s.d., s.e.m.).

- When you resubmit your manuscript, please download our CHECKLIST (<https://bit.ly/EMBOPressAuthorChecklist>) and include the completed form in your submission.

Please note that the Author Checklist will be published alongside the paper as part of the transparent process (<https://www.embopress.org/page/journal/17444292/authorguide#transparentprocess>).

When you submit your revision please attach a covering letter giving details of the way in which you have handled each of the points raised by the referees. A revised manuscript will be once again subject to review and you probably understand that we can give you no guarantee at this stage that the eventual outcome will be favorable.

Kind regards,

Maria

Maria Polychronidou, PhD
Senior Editor
Molecular Systems Biology

We realize that it is difficult to revise to a specific deadline. In the interest of protecting the conceptual advance provided by the work, we recommend a revision within 3 months (16th Oct 2022). Please discuss the revision progress ahead of this time with the editor if you require more time to complete the revisions. Use the link below to submit your revision:

IMPORTANT: When you send your revision, we will require the following items:

1. the manuscript text in LaTeX, RTF or MS Word format
2. a letter with a detailed description of the changes made in response to the referees. Please specify clearly the exact places in the text (pages and paragraphs) where each change has been made in response to each specific comment given
3. three to four 'bullet points' highlighting the main findings of your study
4. a short 'blurb' text summarizing in two sentences the study (max. 250 characters)
5. a 'thumbnail image' (550px width and max 400px height, Illustrator, PowerPoint or jpeg format), which can be used as 'visual title' for the synopsis section of your paper.
6. Please include an author contributions statement after the Acknowledgements section (see <https://www.embopress.org/page/journal/17444292/authorguide>)
7. Please complete the CHECKLIST available at (<https://bit.ly/EMBOPressAuthorChecklist>).

Please note that the Author Checklist will be published alongside the paper as part of the transparent process (<https://www.embopress.org/page/journal/17444292/authorguide#transparentprocess>).

See also figure legend guidelines: <https://www.embopress.org/page/journal/17444292/authorguide#figureformat>

9. Please note that corresponding authors are required to supply an ORCID ID for their name upon submission of a revised manuscript (EMBO Press signed a joint statement to encourage ORCID adoption).

(<https://www.embopress.org/page/journal/17444292/authorguide#editorialprocess>)

Currently, our records indicate that the ORCID for your account is 0000-0001-9977-2104.

Link Not Available

The system will prompt you to fill in your funding and payment information. This will allow Wiley to send you a quote for the article processing charge (APC) in case of acceptance. This quote takes into account any reduction or fee waivers that you may be eligible for. Authors do not need to pay any fees before their manuscript is accepted and transferred to the publisher.

*** PLEASE NOTE *** As part of the EMBO Press transparent editorial process initiative (see our Editorial at <https://dx.doi.org/10.1038/msb.2010.72>), Molecular Systems Biology publishes online a Review Process File with each accepted manuscripts. This file will be published in conjunction with your paper and will include the anonymous referee reports, your point-by-point response and all pertinent correspondence relating to the manuscript. If you do NOT want this File to be published, please inform the editorial office at msb@embo.org within 14 days upon receipt of the present letter.

Rev_Com_number: RC-2022-01563

New_manu_number: MSB-2022-11350

Corr_author: Schramek

Title: Genome-wide CRISPR screens identify novel regulators of wild-type and mutant p53 stability

Full Revision

Manuscript number: RC-2022-01563R

Corresponding author(s): Daniel, SCHRAMEK

[Please use this template only if the submitted manuscript should be considered by the affiliate journal as a full revision in response to the points raised by the reviewers.]

1. General Statements [optional]

Our team sincerely thank the reviewers for their constructive reviews of our manuscript, entitled “Genome-wide CRISPR screens identify novel regulators of wild-type and mutant p53 stability”. We were excited that overall the reviewers were unanimously positive about the manuscript: Reviewer #1: ‘*generating a wealth of data*’, ‘*paper is of high interest*’, ‘*screens are well executed and clearly presented*’; Reviewer #2: ‘*interesting concept and the results could provide a useful resource*’; and Reviewer #3: ‘*current work provides more convincing functional data*’, and ‘*The work is significant in identifying a functionally relevant regulator of p53 stability*’. We think that our study and screen designs, and its impact and significance in finding novel regulators of p53 will be interesting in the broad fields of functional genetics and oncology. Our team further agree with the reviewers that this manuscript would benefit from a more detailed molecular underpinning relevant to the novel mechanisms identified. Along with improved readability, we think we addressed all concerns and questions raised by the reviewers in our revised manuscript and the point-by-point review. The reviewer comments in their entirety can be found below, followed by our response in **green** font.

Considering that the manuscript was very well received, we believe it makes a strong candidate for publication in Molecular Systems Biology, and it will be of great interest to a wide readership in the fields of p53, functional genetics, and cancer biology.

We hope that you will concur with us that the revised manuscript with the incorporated peer reviewers’ suggestions has further strengthen our paper.

This section is mandatory. Please insert a point-by-point reply describing the revisions that were already carried out and included in the transferred manuscript.

While the main scope and strength of this work lies within comprehensive profiling the physiological regulators of 8 of the most common p53 hotspot mutants and as such should be viewed and reviewed as a resource paper, we strengthened the mechanistic insights as suggested by the reviewers. Two recurring comments from all three reviewers highlighted the need for (1) **a more mechanistic insight how the newly identified factors regulate p53**, and (2) **additional functional validation of our hits *in vitro* as well as *in vivo***. In addition, we provide new data describing 2 additional synthetic p53 viability screens in 2 additional human cancer cell lines.

Please find below our point-to-point responses to the reviewer comments.

Reviewer #1.

Reviewer #1 summary:

In this manuscript by Lu et al., the authors describe some CRISPR screens and protein-protein interaction screens to identify novel regulators of wild-type p53 and mutant p53 function and stability. Besides generating a wealth of data, they discover FBXO42-CCDC6 as positive regulators of the some p53 hotspot mutants, including R273H mutant p53, but not of all p53 mutants tested and also not of wild-type, indicating selectivity. Furthermore, they found C16orf72(TAPR1) as a negative regulator of p53 stability. Mechanistically, the authors claim a direct interaction between FBXO42 and CCDC6 and p53, but the importance of these interactions has not been shown. On the other hand the authors suggest that the FBXO42/CCDC6 regulate p53 via destabilization of USP28, but also the mechanism has not been worked out. For C16orf72, they show that it interacts with USP7, but no relevance of this interaction is shown either.

We sincerely thank the reviewer for the constructive and thorough review. We have incorporated most of the suggestions into our revised manuscript.

Reviewer #1, major points.

1. One very important point for me is that the authors do not show the levels of expression of p53 in the p53-mClover stable cell lines. It is known that overexpressed p53 is usually more stable than endogenous levels of wt-p53. Therefore, I think it is necessary that the authors show the levels of the p53-mClover fusion proteins in the stably transduced cell lines compared to endogenous p53 levels in the parental RPE1 cells and also compared to the endogenous levels of R273H mutant in the PANC-1 cells.

We fully agree that the levels of overexpressed p53s are often more than the endogenous ones, due in part to increased expression and stability. In designing the reporter, we first tried to avoid the stabilization of p53-GFP due to GFP aggregation by using the monomeric mClover-variant. Further, we titrated the WT and R273H clones (similar to our recent work in PMID: 35439056), to select clones with p53 levels closer to endogenous protein, and exhibiting high dynamic response to Nutlin-3a treatment.

In the revised submission, we have now included Western blots to directly compare the levels of p53-mClover (WT and R273H) to the endogenous p53s in RPE1 (WT) and PANC1 (R273H) cell lines, in the presence or absence of Nutlin-3a. Upon activation by the MDM2 inhibitor Nutlin-3a the levels of the wt p53 reporter was comparable to endogenous p53 levels in parental RPE1 cells. The R273H p53 reporter showed increased levels even in untreated cells, as expected for p53 hotspot mutations (please see new **Suppl. Fig. 1a**):

2. Also the functionality of the wild-type p53-mClover fusion is questionable, at least not shown. One would expect that the overexpression of a functional wt-p53 in p53-KO cells will affect the survival of the RPE1 cells. In Figure 5A the authors show that depletion of MDM2 or C16ORF72 is toxic for the RPE1 cells in a p53-dependent manner, indicating that elevated levels of p53 cannot be handled by these cells. So, experiment(s) showing that the wt-p53/mClover fusion is functional is needed.

We agree that it is important to benchmark the reporter.

We have now added new data showing that the WTp53-reporter can induce upregulation of the p53 target gene CDKN1A/p21, albeit to a reduced level compared to endogenous p53 in parental RPE1 cells. This indicates that the reporter is transcriptionally active and can form active tetramers, albeit to a lesser degree than endogenous untagged p53. In addition, we also showed that a mutant p53 reporter such as R273H p53-mClover lacks transactivation capabilities, as expected for p53 hotspot mutations (please see previous blot and new **Suppl. Fig. 1a**).

3. A second important point is that the 'verification' of the hits from the screens is only done in one cancer cell line, PANC-1, with mutant p53. I would have like to see at least one other cell line with another p53 mutant endogenously expressed that is also regulated by FBXO42/CCDC6.

We now include validation of the hits (FBXO42, CCDC6) in an additional human pancreatic cancer cell line, MiaPaCa2, which is homozygous for p53 R248W, corroborating our findings in an additional cell line (please see new **Suppl. Fig. 3h**):

4. For many of the p53-mutants, a bimodal expression is observed. In the FBXO42- and CCDC6-depleted cells, the equilibrium shifts towards more negative cells but the levels in the two populations itself don't change (while for example for USP28 depletion also the right peak shifts further up, Fig S4E). Is there any correlation with the cell cycle and p53 expression? And can the authors exclude that FBXO42 and CCDC6 are involved in cell cycle progression and hereby influence p53 indirectly (by combining PI staining with Clover-p53 for example).

We have indeed observed that the "bimodal" levels in the reporters of several mutants, which are also observed in other studies probing the endogenous p53 level (PMID: 29653964) and they also observed that the population equilibrium shifts, while the location of each peak (as a proxy of the level of p53s) are more stable. As such, this seems to be a 'endogenous' behavior, which we have seen in all clones of a given reporter.

Regarding the relation between p53-level and cell cycle stage, indeed, both the authors in the paper above and we have probed this possibility. Indeed, we observed that e.g. the R273H p53 levels are higher in S- and G2/M-phase RPE1 cells compared the G1-phase. Conversely, cells with low R273H p53 levels

show a higher percentage of G1 cells compared to cells with high R273H p53 levels, consistent with the data from Galit Lahav (PMID 30057196):

Left: G1, S, and G2-M.

Right: Pool, p53-low, and p53-high.

To address the question whether CCDC6 or FBXO42 regulate cell cycle and therefore maybe indirectly p53 levels, we performed cell cycle analysis in RPE1 as well as PANC1 cells, and we did not detect any alteration in cell cycle distribution between control and CCDC6 or FBXO42 knock-out cells (please see new **Suppl. Fig. 3g**):

5. The authors claim that the FBXO42-CCDC6 axis regulates stability specifically some p53-mutants, including R273H-mutant, in a manner involving USP28. But USP28 regulates all forms of p53, not just some mutants version. How can the authors reconcile this apparent contradiction?

We thank the reviewer for this critical and very insightful observation. From our screen (Supplemental Table 1A), we have indeed noticed a pronounced effect ($|Z \text{ score}| \geq 3$) of FBXO42 on R273H and R248Q stability, and a marginal effect on wild-type p53. Similarly, USP28 had pronounced effects on R273H and R248Q but also on WTP53.

In the discussion of the paper, we note that USP28 was shown to regulate p53 levels through distinct mechanisms: *'USP28 was originally implicated as a protective deubiquitinating enzyme counteracting the proteasomal degradation of p53, TP53BP1, CHCK2, and additional proteins⁶⁸⁻⁷¹. USP28 regulates wild-type p53 via TP53BP1-dependent and -independent mechanisms. Concordantly, our data shows that USP28 and TP53BP1 are strong positive regulators of wild-type p53. However, while USP28 was also a strong hit in the mutant R273H p53 screen, TP53BP1 was not, indicating that the effects we see upon loss of USP28 on R273H p53 are independent of TP53BP1.'*

Together, this suggests that the R273H-mutant is regulated by a FBXO42-CCDC6-USP28 axis while wild-type p53 is regulated mainly via a USP28-TP53BP1 axis. Further explorations will be needed to unravel the intricacies of these fine-tuned regulatory mechanisms.

6. On a similar note, the authors show that FBXO42 and CCDC6 interact with p53, but not USP28. Do FBXO42 and CCDC6 interact with each other and with USP28? And is the interaction with p53 specific for the R273H version? This part of the mechanism is very poorly defined and the Co-IPs are not very convincing or relevant for the proposed model.

We have indeed observed the interaction between FBXO42 and CCDC6 (via BioID and APMS, e.g. Fig. 3a); however, we failed to recover USP28 as an interactor of either FBXO42 or CCDC6. We showed that CCDC6 binds to wt p53 by BioID and to R273H p53 by BioID, proximity ligation assay and co-IP (please see Fig. 3a-c and Suppl. Fig 4e). We also showed that FBXO42 binds to wt p53 by BioID as well as AP-MS and to R273H p53 by BioID and *in vitro* using recombinant proteins (please see Fig. 3d). We could not detect an interaction between CCDC6 or FBXO42 with R175H p53 by AP-MS or BioID, which is in line with the lack of an effect upon genetic ablation of CCDC6 or FBXO42 on R175H p53 protein level. However, in our opinion, failure to detect an interaction is not a very conclusive proof of lack thereof and as such, we would prefer not to emphasize this too much in the manuscript.

In the manuscript, we clearly show that there is a genetic interaction between FBXO42-CCDC6-USP28-p53 as detected by unbiased anchored genetic screens, as well as through epistasis experiments analyzed by Western blotting and immunofluorescence as well as rescue experiments.

While the exact molecular underpinnings of the FBXO42-CCDC6-USP28-p53 axis remain to be elucidated in future work, we hope that this reviewer agrees that the presented evidence clearly shows a new regulatory facet with important insights of the so far unknown “upstream” regulation of USP28 and its regulation of p53.

Reviewer #1, minor points.

7. The mechanisms of p53 regulation may vary greatly in different cell lines. Can the authors discuss why they choose to do the screen with different mutants, rather than with different cell lines expressing these same mutant endogenously?

While it is certainly very interesting to assess how WT and mutant p53 is regulated in different cell lines, such an approach is confounded by the ‘genetic make-up’ of the respective tested cell lines. For example, TP53BP1 might be a regulator in one cell line but not in another for the simple reason that the later cell line harbors a TP53BP1 deletion or mutation or reduced expression levels. In addition, while working with endogenous p53 mutations certainly has many advantages, comparing different mutants in different cell lines is again very much confounded by the ‘genetic make-up’ of the respective tested cell lines. . These confounders are avoided when using an isogenic set of cell lines, enabling a more rigorous elaboration of differences between different mutants.

However, to start to address how the genetic background of different human cancer cell lines affects the regulation of p53, we now conducted another 4 genome-wide CRISPR screens in p53-proficient as well as isogenic p53-deficient A549 lung cancer as well as RKO colon cancer cell lines. These comprehensive synthetic lethal screens showed that the regulatory network of p53 within RPE and A549 cells is notably

similar, while RKO cells exhibit a drastically different regulatory inventory. This further highlights the need to identify regulators of p53 stability within isogenic cell lines rather than across cell lines (please see **new Suppl. Fig. 6a**; and **Suppl. Table 4**):

As such, our focus was to specifically assess the difference between p53 hotspot mutations. Are they all the same or are there differences and importantly, are there differences between mutants and WT p53? This can only be achieved when working in the same cellular background. In designing the screen, we have thus tried to optimize different p53 hotspot mutant reporters in an isogenic screening system. As such, we first depleted the endogenous WTp53 to minimize its interference and built the current isogenic series of non-transformed RPE1 (“normal”) p53 reporter lines.

However, as discussed above, we have now corroborated our results in an additional pancreatic cell line carrying endogenous mutants, further underscoring the importance of our findings and the validity of our screens, which we hope will offer a rich resource to the broader community.

8. Figure 1: Typo in the legends : Nultin ipv Nutlin

We apologise for the typos. This is addressed in the current submission, along with improved figure legends to improve readability.

9. Figure 1b,1c : Show basal and Nutlin-3 induced MDM2 levels and in the overexpression cell lines; if WT-p53 is functional, MDM2 levels should be higher in WT-transduced cells compared to control or mt-p53 expressing cells.

To show that the WT-p53 is functional, we have now included Western blotting probing the levels p21 as well as MDM2 upon nutlin (please see new Suppl. Fig. 1a, and WB on the right below – not shown in the paper), which show that p21 as well as MDM2 is induced in parental RPE1 cells as well as RPE1 cells expressing the p53 wt reporter, but not in RPE1 cells expressing the mutant p53 R273H reporter, as expected (please see new Suppl Fig. S1a):

10. Authors should explain which they name USP7 a negative regulator of p53, since it is supposed to de-ubiquitinate p53?!

Indeed, USP7 deubiquitinates both p53 and MDM2/MDM4. The effects of USP7 on WTp53 have been elucidated by Prof. Vogelstein (PMID: 15118411, and PMID: 15058298), and by Prof. Wei Gu, (PMID: 15053880, and PMID: 11923872), with seemingly opposite results. However, the consensus in the field is now that USP7 prefers MDM2/MDM4 over p53, at least in the absence of severe DNA damage, and therefore its depletion stabilizes MDM2/MDM4 and leads to degradation of p53 (for recent reviews see PMID: 32300595 and PMID: 36428632). Thus, USP7 is primarily an MDM2/MDM4 activator and hence a negative regulator of p53. Indeed, consistent with the current knowledge, the inhibition of USP7 (either by inhibitor or genetically via CRISPR as in our studies), has resulted in elevated p53 level.

11. Figure 2E: the effect of MG132 on p53 seems to be very minimal on this Western blot; it would need quantification to be convincing...Quality of the blot is also not great.

The fact that in control cells the levels of p53 R273H are not affected by MG132 treatment fits with Suppl Figure 2E, indicating that the proteasome has no effect on p53 R273H.

We agree with this reviewer and have indeed noticed that while the proteasome pathway is largely implicated in the WTp53 screen, it has much reduced effects on R273H, which we had actually shown using genetic ablation of proteasome subunits in the original Suppl. Fig. 2e. We thus rephrased this sentence to:

'Inhibition of the proteasome by MG132 modestly increased mutant p53 levels in FBXO42 and CCDC6 knock-out cells as assessed by WB analysis and flow cytometry (Fig. 2e and Suppl. Fig. 3e), which is consistent with our genetic results of ablating proteasome subunits (Suppl. Fig. 2e).'

12. Suppl figure 3b, 3c, 3d:

Somehow, I have the feeling that the results from the western blots and the FACS do not match fully, although not all the time-points are shown in the various experiments.

For example, the FACS analysis (3b) suggests that in control-transduced cells after 16hr p53 is still increased. However, that is not clear at all in the Western blot (3c)

Is Suppl Figure 3d the quantification of 3c experiment? If so, in the blot also the 24 hrs should be shown.

The blot shown in Suppl Figure 3c suggests that CCDC6 expression increased upon irradiation. Do the authors agree with that? Would that explain why depletion of CCDC6 has more effect upon irradiation?

Suppl Figure S3E: if I am right, this is essentially the same type of experiment as shown in figure 2e, but analysis of p53-expression by Western blot. In that blot no real effect of MG132 on p53 levels could be seen. But here, in the FACS analysis, MG132 clearly increases the p53-Clover fusion levels; for me again that Western blot and FACS data do not necessarily match.

With regards to the difference between the flow cytometry and WB data, we have generally observed the flow cytometry bimodal shifting to be more sensitive than the WB, which may be partially explained as WB is a measurement across the cell population and FACS determines the p53-GFP levels of every cell and thus the shift of cells between peaks. However, it is important to note that we always observe the same overall shift and direction of change when assessing the cells by either WB or flow cytometry. In addition, as pointed out by this reviewer, we have quantified the effects across many blots as shown in Suppl Figure 3c and the quantification in 3d, which is esp. important with these rather large WB experiments with several knock-out lines and time points. This also makes it hard to find the best representative WB, and as such, we hope that the quantification suffices.

13. Figure 3B: In the CCDC6 IP a very small amount of p53 can be found. I don't know how much input lysate compared to amount of lysate for IP is used, but the percentage of p53 found interacting with CCDC6 seems so marginal that is difficult to explain the effect of KO of CCDC6 in PANC1 cells.
And, the authors called it a 'reciprocal IP' (Suppl Figure 4a) after transfection of V5-tagged CCDC6 into PANC1 cells, but it actually is the same type of IP. Did the authors try to IP p53 and blot for CCDC6? That would be a reciprocal IP.
We apologize for the confusion. In the revised submission, we specified the portion of the lysates used for pre-IP (5% lysate) and IP (1 mg). As for the IP, we corrected the sentence regarding our 'reciprocal' mistake:

*'In addition, endogenous CCDC6 as well as V5-tagged CCDC6 co-immunoprecipitated (co-IP) with endogenous p53 R273H in PANC-1 cells (**Fig. 3b** and **Suppl. Fig. 4e (=old Suppl Fig 4a** ).'*

Despite our efforts to perform a reciprocal IP, we have not been successful. However, we do want to point out that we used several reciprocal approaches to show this interaction, including proximity ligation assays (please see Fig. 3c) as well as BioID and AP-MS. Together with our genetic data, we hope that this reviewer agrees that this interaction is real.

14. Figure 3H: how can authors explain that basal levels of USP28 in control and CCDC6-KO cells transfected with control plasmid are more or less the same and not reduced in the CCDC6-KO cells?
We agree with this notion and want to clarify that the control and CCDC6-KO cells are clonally derived, (which is not otherwise technically possible) and as such it is a little hard to compare across those clones. However, the most important take home message of this plot is that within those clonal cell lines, overexpression of CCDC6 beautifully rescues USP28 (and p53) levels in CCDC6 KO cells, while overexpression of FBXO42 does not, as expected. Similarly, overexpression of FBXO42 increases USP28 (and p53) levels in FBXO42 knockout cells. Even more so, in WT cells, overexpression of FBXO42 and CCDC8 increase USP28 (and p53) levels, showing the exact opposite of their knock-down. Of note, we have already clarified in the figure legends that these are clonal cell lines. Overall, we strongly believe and hope this reviewer agrees that these are really strong data confirming our findings.

15. Figure 3I: Essentially the whole blot here is of low quality; especially the FBXO42 blot; is deletion of USP28 increasing FBXO42 protein levels, or is it just the quality of the blot?
All in all it seems that FBXO42 is very low expressed in the used cell lines.
We apologize for the low quality blot. In the revised submission, we included a higher quality WB, with more optimized conditions for using the FBXO42 antibody (Fig. 3I):

16. Figure 4B: I find it a bit surprising that USP7 is also found in the synthetic viability screen, since it has been shown that USP7 has many more essential targets and KO of p53 only partially rescues the development of USP7-KO mouse embryo's.

We thank the reviewer for this critical observation. While the double p53-USP7 knockout line is viable, we acknowledge that it is amongst the top scored hits due to the large differential viabilities between WT and p53-null lines, in line with the mentioned data showing partial rescue of USP7-KO mouse embryo by p53 loss. This is not to say that USP7 does not have other essential targets. However, as this is not the focus of the current manuscript and given that the manuscript is already very broad in scope and long, we would prefer not to pick this up in the discussion. However, if this reviewer believes that this is important, we are happy to.

17. Figure 5: the authors nowhere show the efficacy of the guides targeting c16orf72. A Western blot showing the expression and the reduction upon expressing the guide-RNAs is essential.

We thank the Reviewer for this suggestion. The efficacy of each guide has been verified using ICE (at the genomic level), and in the revised submission, we have included this critical information as part of the Figure S2F:

18. Figure 5E: First, here probably parental RPE1 cells have been used, but that is not stated. Second, the authors state 'only a slight increase in p53 levels upon siHUWE1'; I would say none compared to scrambled. I know HUWE1 is a very huge protein, but the blot of HUWE1 is not convincing. I seem to be able to conclude that siMDM2 and siUSP7 reduces HUWE1 levels?

Response: We apologize for this oversight, yes, parental RPE1 cells were used in this experiment. We have now stated this in the figure legend.

We agree with the reviewer that assessment of large proteins by WB is often difficult but given that this band almost completely disappears upon HUWE1 knock-down, strongly argues that we are indeed assessing the endogenous HUWE1. We also agree that it is an interesting observation that the levels of HUWE1 seem to be slightly reduced upon knock-down of MDM2 and USP7 – while this is an interesting observation, the effects are modest and do not directly affect the main conclusion of the experiment and as such, we did not mention it in the text. HUWE1, a known E3 ubiquitin ligase for p53, scored as a hit in our marker-based p53 stability screens and showed the strongest co-essentiality in DepMap (Fig. 2C and 5e). As such, we are confident that HUWE1 is indeed regulating p53 levels.

However, HUWE1 is a common essential gene as shown in the DepMap data but also as shown in our synthetic viability screen in p53-deficient as well as p53-proficient RPE1, RKO and A549 cells (please see **new Suppl. Fig. 6a and Suppl. Table 4**), which strongly indicates that HUWE1 is not only a negative regulator of wt-p53 but also regulates several other substrates essential for cell survival. This makes even transient knock-down experiments very difficult, as cells are very unhealthy and on the path to cell death as soon as HUWE1 is knocked-down.

Regarding the relationship between C16orf72 and HUWE1, a newly published paper (PMID:37167062) showed that C16orf72 (termed HAPSTR1 in the paper, the new official name, which we have now adopted in the revised manuscript) binds to HUWE1 in HCT116 cells, in line with our data. This new paper also showed that C16orf72/HAPSTR1 is shuttling HUWE1 into the nucleus where HUWE1 modulates several signaling pathways, including p53 and nuclear factor κ B (NF- κ B)-mediated signaling.

To gain further insights on how C16orf72 is regulating mutant p53 levels in RPE1 cells and to test how HUWE1 is involved in this regulation, we first performed rescue experiments using wildtype C16orf72 as well as an NLS-truncation mutant C16orf72. These experiments clearly showed that the nuclear localization of C16orf72 is important for the regulation of p53. We further showed that the ability of C16orf72 to regulate p53 is completely dependent on HUWE1 (please see **new Fig 5f and g**).

Additional cycloheximide-chase experiments together with Nutlin treatments showed that C16orf72 regulates p53 in an MDM2 independent manner (please see **new Suppl. Fig. 6c and d**), further supporting the importance of C16orf72-HUWE1 axis in regulating p53 stability:

19. Figure 5F, in relation to figure 5D. Here the author overexpress both c16orf72 and USP7, and find an interaction. The implication of that is not clear. If they want to make point of this interaction, they should have looked at endogenous proteins.

We acknowledge the many concerns associated with coIP with ectopically, and especially overexpressed proteins in large quantity. Given our new data showing that the effect of C16orf72 on p53 stabilization is dependent on HUWE1, we believe that the USP7 connection is maybe of less relevance. However, we feel that it is still worth reporting given that the Co-IP results are really strong and might be important for further investigations especially given that USP7 is the neighboring gene of C16orf72 and as such is co-amplified in many cancers to a very significant degree (please see Fig. 6 and Supplementary Fig. 7)

20. It is worrying that USP7 apparently was not one of the hits in the Mass-spec experiment of which results are shown in Figure 5D. Also in that experiment c16orf72 was overexpressed, and USP7 is very highly expressed in essentially all cell lines, so do the authors have an explanation?

We indeed acknowledge this discrepancy and the limitation associated with the AP-MS for the detection of interactors. USP7 was actually identified in the C16orf72 AP-MS experiments in both, 293 and U2OS cells, but it was also found in the GFP negative control and as such, did not end up as a specific enriched hit. USP7 is simply a sticky protein, which explains the AP-MS results. However, the CoIP results are very clear and we would like to refer to point 19 and keep that data in the manuscript. However, if this reviewer and the editor feel we should omit that data, we are happy to do so.

21. Suppl. figure 5D is missing

We apologize for the confusion. The Figure S5D (old) was inconveniently placed at the top of the figure panel due to space limitation. In the revised submission, panel S5D (now to S7C) is now clearly visible for overall readability improvement.

Full Revision

Reviewer #1, Significance.

The topic of the **paper is of high interest** given the relevance of p53 and its gain-of-function mutants in oncology, and the **screens are well executed and clearly presented**. In terms of novelty, FBXO42 has been linked to p53-degradation before, and c16orf72 was recently shown to be able to destabilize p53. However, the link between CCDC6 and p53 is novel and of interest, since they are both substrates of USP7 and are both regulators of the cell cycle.

We think the manuscript has potential to add something to the field, but would benefit greatly from a better understanding of the molecular underpinnings of their newly described mechanisms, as well as the conditions in which the mechanism is active.

Therefore, it might be advisable to shorten the manuscript, and go more in-depth in finding the mechanisms of regulation.

Response: We sincerely thank the reviewer for all the constructive critiques. We will incorporate them into our revision.

Reviewer #2.

Reviewer #2 summary:

The paper describes several genome-wide CRISPR screens designed to identify regulators of p53 stability. The authors use a system in which p53 levels are marked by mClover expression, using RFP expression to normalise for gene expression changes.

Reviewer #2, major points.

1. The bimodal distribution of p53 expression levels in some reporter cell lines (G245S, R248Q, R248W and R273H) hampers the implementation of a robust readout and makes correct interpretation of the results challenging. While it is possible that the bimodal distribution indicates dynamic changes in p53 levels within one population, it also seems possible that a subclone of these cells have acquired additional alterations affecting p53 stability, and that the authors are screening a mixed population of two intrinsically different cell populations. This would make it difficult to interpret the results of the screen in these cell lines and may be a challenge when trying to identify something that has not already been highlighted on depmap.

We thank the reviewer for this critical observation. We strongly believe that this bimodal distribution is actually an inherent property of the p53 mutants in these cells for the following reasons: (1) The observation of the similar bimodal appearance in cell lines harboring corresponding endogenous mutant p53s (PMID: 29653964) suggest that these two populations are of biological significance. (2) We have established 5-10 clonal lines each from the G245S, R248Q, R248W and R273H p53 reporter line and all of them exhibit a bimodal distribution, making it very unlikely that these populations are all through stochastic outgrowth of sub-populations with spontaneous mutations/alterations (we are happy to include this data into the supplementary information if this reviewer or the editor thinks it is beneficial). (3) The bimodal distribution is stable over several months to years in culture. If it were a spontaneous mutations giving rise to a clone with higher mutant p53 levels, we would likely expect that over time this clone takes over the population. (4) We observed that such a pool of bimodal cells could be “synchronized” (e.g. by Nutlin, or MDM2 knockout) to one population, and later return to and repopulate the other (e.g. Nutlin washoff, Figure 1B). (5) When we sort out single cells from the upper or the lower peak and expand them, we obtain again populations of cells with the same bimodal distribution, indicating that this is a dynamic process. Thus, we believe that these two populations are intrinsic, such that a cell in the population may assume both states.

We also acknowledge the difficulties of screening using a bimodal population; however, we took advantage of these “bimodal” mutants and using FACS assessed the state of a single cell in relation to a genetic perturbation. Each guide has an equal chance of entering a cell that belongs to one of the two populations. If a gene knock-out really affects p53 levels, the cells with the respective guides should be enriched in one and depleted in the other population. The analysis comparing the guide abundances from these two peaks ensures the experiment are being perfectly internally controlled.

While many of the top scored hits from the resulting screens are known regulators, it is critical to point out that we could validate our hits in an independent system, such as the cell lines harboring endogenous p53 mutations such as the pancreatic cancer cell lines PANC1 and MiaPaca2. Please see **new Fig 3i** and **new Suppl. Fig. 3h**:

2. The coverage of the sgRNA library (200x) is rather low for a negative selection screen, where a coverage of 500x would be more desirable. The FDR threshold is also rather lenient, a more stringent FDR threshold would seem more appropriate and shorten the list of potential hits.

We thank the reviewer for this constructive suggestion. We agree a higher coverage, along with a more stringent FDR, will ensure an even stronger confidence for the remaining individual hits. The present reporter-based enrichment screen and the synthetic viability drop-out screen used four guides per gene, and with 200x coverage for each guide.

In determining the coverage, we tried to reference recent successful screenings and apply earlier titration result for the 200x coverage (e.g. PMID: 26627737, PMID: 33465779, and reviewed in Nat Rev Methods Primers 2, 8 (2022). <https://doi.org/10.1038/s43586-021-00093-4>). While the threshold of FDR was often arbitrary, we fully agree that a more stringent FDR, which results in a shortened hits list, may further boost the confidence of the hits, though at the cost of losing potential hits due to collateral effects (e.g. guide efficiency).

We agree with this reviewer that a higher FDR, esp. at the hits that result in p53 stabilization, would make sense as any gene whose loss causes cellular or genotoxic stress, would likely lead at least in part to p53 stabilization. We have thus now also performed another set of synthetic viability screens in p53 deficient and p53 proficient RKO as well as A549 cells, which help to further prioritize hits, but that also show a striking context/cell type-specificity of hits. We have now reported all these 4 new screens in addition to all the other 18 genome-wide screens that were already reported in the originally submitted manuscript where we report also all the FDR. As we see this work as a resource to the p53 community, we think that every reader can use the FDR cut-off they feel confident with and design follow-up experiments accordingly.

3. Although the study is focused on the regulation of p53 stability, there are no experiments to show that any of the manipulations alter the ubiquitination or degradation (half-life) of p53. The rescue of expression by proteasome inhibition is very modest (Figure 2E), suggesting the loss of expression may not be a reflection of degradation. A role for endogenous FBXO42 and C16orf72 in regulating the ubiquitination and half-life of endogenous p53 should be confirmed

We agree with this reviewer. While the proteasome pathway is largely implicated in our WTp53 screen, it has much reduced effects on R273H, as we have actually shown using genetic ablation of proteasome subunits in the original Suppl. Fig. 2e. We also agree that the effect of proteasome inhibition is modest. We thus rephrased the manuscript:

'Inhibition of the proteasome by MG132 modestly increased mutant p53 levels in FBXO42 and CCDC6 knock-out cells as assessed by WB analysis and flow cytometry (Fig. 2e and Suppl. Fig. 3e), which is consistent with our genetic results of ablating proteasome subunits (Suppl. Fig. 2e).'

We have then monitored the degradation of CCDC6 using cycloheximide-chase experiments in PANC1 cells expressing the R273H p53 mutant, with or without the knock-down of CCDC6 in the presence of the proteasome inhibitor MG132. Interestingly, mutant R273 p53 levels decreased over time in control and even more so in CCDC6 knock-down PANC1 cells, indicating that there is a substantial proteasome-independent degradation of mutant p53.

We thus tested inhibition of the lysosome using chloroquine, which stabilized mutant p53, indicating that p53 is degraded via the lysosome particularly in CCDC6 and FBXO42 knock-out cells (please see **new Suppl. Fig. 3f**):

F

This data is in line with previous studies showing that mutant but not wildtype p53 is degraded via chaperone-mediated autophagy in a lysosome-dependent fashion. We have now added a new section to the discussion noting that: ‘*Vakifahmetoglu-Norberg et al. presented evidence that mutant p53 is resistant to proteasomal degradation due to an inability of mutant p53 to be ubiquitinated, which favors lysosomal degradation of mutant p53³⁹, and it is conceivable that FBXO42/CCDC6/USP28 might be involved in this phenomenon. While the exact molecular mechanisms are currently unclear, we provide strong evidence that FBXO42/CCDC6 is required for USP28-mediated regulation of p53 R273H stability, suggesting that interfering with this regulatory circuit could present an avenue to prevent or reduce mutant p53 accumulation in tumours.*’

4. Many p53 mutants are used for the initial screens, but very little validation is carried out to show that the apparent differences in factors regulating their stability persists in cells naturally expressing these mutants. For example, FBXO42 is identified as a protein required to maintain the stability of R273H, 248W and R248Q, but not R175H, G245S and R337H. While the authors show an association of CCDC6 and p53 in PANC1 cells (expressing 273H), it would be important to show a panel of R273H, 248W and R248Q expressing tumor cells and the response of p53 to FBXO42 and CCDC6 depletion, compared to similar experiments in a panel of R175H, G245S and R337H expressing tumor cells. Again, it would be important to show that any changes in protein levels are due to changes in protein stability.

We thank the reviewer for this suggestion. In the revised submission, we have included validations in more cell lines carrying endogenous mutant p53s. We used the pancreatic cancer cell lines PANC1, which harbors endogenous homozygous p53 R273H mutations, and MiaPaca2 which harbors endogenous homozygous p53 R248W mutations and genetically ablated FBXO42 and CCDC6 using 2-3 different sgRNAs, which corroborated our RPE1 data. Please see **new Fig 3i** and **new Suppl. Fig. 3h**:

5. The potential hits should also be tested in wild type p53 expressing cells to confirm the specificity to mutant p53s.

We tested genetic ablation of FBXO42 as well as CCDC6 using two independent sgRNAs in a panel of isogenic wt and p53 mutant reporters. Upon genetic ablation of FBXO42 or CCDC6, we observed significantly reduced R273H, R248Q and R248W p53-mClover levels but failed to see significant effects on wild-type, R175H, G245S or R337H p53-mClover levels (please see **Fig. 2d** and **Suppl. Fig. 3a**). To make any claims whether a hit is specific to certain mutants or wildtype p53, one has to use an isogenic background, as the genetic background of a given cell dramatically affects the p53 regulatory network, as demonstrated in our new synthetic viability screens in RPE, RKO and A549 cells.

In addition, testing the effect of FBXO42/CCDC6 ablation on wildtype p53 in cells with endogenous p53, is technically challenging, as baseline wildtype p53 levels are very low/undetectable, and as such showing that FBXO42/CCDC6 ablation further lowers p53 levels is not possible. In addition, the regulatory network of FBXO42/CCDC6 is parallel to the MDM2 feedback regulation of p53 as nutlin treatment can completely override the effects of FBXO42/CCDC6 loss:

As such, artificially increasing p53 levels using e.g. nutlin or doxorubicin in wildtype p53 expressing cells would not allow us to measure the effect of CCDC6/FBXO42 loss on p53-stability.

With regard to our other hit, C16orf72, we have shown its effect on wtp53 and mutant p53 by loss-of-function and gain-of-function experiments in vitro as well as *in vivo*:

Please see **Fig. 5b and e** for endogenous p53 wt in RPE cells as well as Fig. 5c for wt and p53 R273H RPE1 reporters and Suppl. Fig S6A for additional p53 reporters.

Please see **Fig 6c** for the effect of C16orf72 overexpression on the p53 R273H reporter and d for the effect on wildtype p53 on human mammary epithelial MCF10A cells, as well as Fig. 6f for the effect of C16orf72 overexpression on endogenous p53 in mouse mammary epithelium:

6. (6A) The role of C16orf72 in restraining p53 activity has been reported previously, as has the interaction with HUWE1 (including a new publication PMID: 35776542). The authors suggest an interaction between C16orf72 and USP7, although this should be shown with endogenous proteins. The relative importance of USP7 and HUWE1 binding is not explored. (6B) The effect of C16orf72 overexpression in promoting mammary tumors is impressive, although maybe the more interesting question is whether inhibition of C16orf72 expression can limit tumor development in this system.

Response to 6A: we are excited about the independent observations by other group(s) confirming similar results! As a part of our improvement for mechanistic work-up, in the revised submission we now added new data showing that C16orf72' regulation of p53 is actually dependent on HUWE1 and HUWE1's nuclear localization. This is indeed new data and strong genetic evidence for the C16orf72-HUWE1-p53 regulatory axis (please see **new Fig 5f and g**):

Given this new data showing that the effect of C16orf72 on p53 stabilization is dependent on HUWE1, we believe that the USP7 connection is maybe of less relevance. However, we feel that it is still worth reporting given that the Co-IP results are really strong and might be important for further investigations, especially given that USP7 is the neighboring gene of C16orf72 and as such co-amplified in many cancers to a very significant degree (please see Fig. 6 and Supplementary Fig. 7) However, if this reviewer and the editor feel we should omit that data, we are happy to do so.

6B. The effect of C16orf72 overexpression in promoting mammary tumors is impressive, although maybe the more interesting question is whether inhibition of C16orf72 expression can limit tumor development in this system.

We are also equally excited about the *in vivo* result supporting the idea that C16orf72 overexpression in tumour-prone (Pik3ca^{H1047R}) mice harboring WTp53 may accelerate tumor formations. In the revised manuscript, we provide further support that this effect is specific to WTp53/C16orf72, by overexpressing C16orf72 as well as the controls Usp7 and Mdm2 in an additional control cohort of p53-null Pik3ca mice

(= LSL-Pi3kH1047R; p53^{Flox/Flox}). Importantly, these new mouse cancer cohorts showed that the effect of C16orf72 (or Mdm2 or Usp7) overexpression strictly depends on the presence of endogenous wildtype p53 (please see **new Fig panel 6e**):

Regarding the effects of C16orf72-depletion in controlling tumor growth - we agree that this would be a very exciting avenue. Conditional C16orf72 mice are being made at the moment and these mice will allow us to comprehensively address this question. However, it will take several more months to generate and validate this line, and then another 2 breeding rounds to generate homozygous C16orf72^{fl/fl}; Pik3ca^{H1047R} mice. In addition, in view of the long time required to form tumors in the control mice with WTp53 (~250 days), it becomes unfeasible for us to test whether the inhibition of C16orf72 could limit the tumor development, given the revision timeline. As such we respectfully believe that this would be beyond the scope of this manuscript. However, we are looking forward to sharing the results of these experiments in future publications.

Reviewer #2, Minor comments.

7. Figure 1b: The nutlin concentration stated in the methods section is wrong. Should be 10 μ M instead of 10 nM (correct in figure legend).

Figure 6b: y-axis label is missing.

Figure 1e/f Legend: Should be FDR <0.5 not >0.5.

We apologize for these typos. The current submission has incorporated the corrections.

8. Figure 1c: Include results for a mutant that is not regulated by MDM2, such as R175H. Otherwise, as a standalone experiment, this figure doesn't add much.

The experiment in 1c is meant to validate (1) the Cas9 knock-out efficacy and (2) the efficacy of the reporter upon ablation as well as overexpression of MDM2, and as such, we feel it is important to show this data. The effect of MDM2 on various p53 mutations and wildtype p53 is already shown in Fig. 1b using nutlin.

9. Figure 1h: While an UpSet plot is an elegant way to present unique and overlapping hits between different screens, Venn diagrams might be more 'accessible' to many readers and easier to understand.

We thank the reviewer for this feedback. The choice of UpSet plot was largely motivated by the different categories involved, which made the area representation and the intersection of the conventional Venn diagram no longer feasible. A Venn diagram of 6 or in this case of 12 intersecting events is really hard to visualize.

10. Might be worth stating that mClover is an eGFP variant and can therefore be targeted by eGFP sgRNAs so that it is easier to understand the following:

o Page 5, paragraph 1: "We used the TKOv3 sgRNA library, which contains [...] 142 control sgRNAs targeting EGFP, LacZ and luciferase"

o Page 5, paragraph 2: "As expected, sgRNAs targeting p53 and mClover were the most depleted sgRNAs, [...]"

We thank the reviewer for this suggestion and have added the following sentence to the manuscript: *'mClover is a monomeric variant of GFP, so it is targeted by the GFP guides'*

Reviewer #2, Significance.

This is an **interesting concept and the results could provide a useful resource** for groups interested in the regulation of p53. The authors chose to focus on candidate genes that could have been identified by looking for the top 30 p53 co-dependent genes on depmap (C16orf72 is #24 in this list and FBXO42 is #28, most of the other genes ranking above are already known as p53 regulators). While this validates the screen, it would have been interesting if the authors had identified and validated new regulators of p53 that were not apparent from previously published work.

We thank the reviewer for all the thorough and constructive comments! In relation to the DepMap dataset, we are excited that many of the top hits from our screens are indeed top WTP53-correlators/anti-correlators (e.g. MDM2, USP28)!

While the DepMap dataset used cell fitness/viability to construct the genetic relation score, this assay may not effectively rule out the many regulators that could otherwise elicit their regulation of p53 via regulating the general cellular response to cell cycle changes, stress, etc. In our screen systems (i.e. protein stability and synthetic viability screens), we attempted to focus on the regulators of p53 stability (post-translational), and further coupled it with the synthetic viability screens to concentrate on hits that have a more direct role in p53 regulation (e.g. MDM2, C16orf72).

One other difficulty to fully couple our screens to the DepMap dataset is due to the limited cell lines harboring endogenous mutant p53s, e.g. R337H. This may also contribute to the uniqueness of the identified R337H-reporter specific hits (where cell lines harboring R337H have not yet been included in the DepMap dataset), e.g. several Aminoacyl tRNA synthetases (SARS, YARS, etc) were identified as R337H unique regulators and subsequently verified using different guides in the reporter line, but could not be obtained via DepMap.

We largely see this paper as a resource for the p53 field and would like to publish it as soon as possible. In fact, when we started working on C16orf72 or CCDC6/FBXO42, these hits were not known for their ability to regulate p53. We will work up several other hits in subsequent publications.

Reviewer #3.

Reviewer #3 summary:

The manuscript by Lu and coworkers performed genome wide CRISPR screens to search for genes that when knocked out, lead to p53 accumulation or degradation. Wt p53 and a panel of p53 hotspot mutants were chosen as reporter for the screen. The approach reassuringly identified many previously described regulators of p53 degradation, and also found a large set of new hits that many appear to be indirectly affecting p53 level.

A key step of this approach is the follow up functional and mechanistic study of the hits. To this end, the authors chose FBXO42 as a top hit that blocks mutant p53 degradation, and C16orf72 as a top hit that promotes wt/mutant p53 degradation.

Overall the functional data for FBXO42 is disappointing. FBXO42 knockout has quite modest effect on mutant p53 level (~50% reduction). The knockout also showed some effect on p53 mRNA level (~25% reduction), making the determination of mechanism difficult. It does not appear to be a promising targeting for reducing mutant p53 level and gain of function activity in tumor cells.

We thank the reviewer for this constructive comment! We address this in the revision, as proposed in Point #3.

The C16orf72 finding unfortunately lost some novelty because it was independently identified as a p53 regulator in a recent study using CRISPR screening (PMID: 33660365). However, the repeated identification is reassuring and the **current work provides more convincing functional data**, showing C16orf72 knockout increase wt p53 level, inhibits cell proliferation specifically in p53^{+/+} cells, and overexpression of C16orf72 reduce wt p53 level and accelerates progression of a breast tumor mouse model. Their results suggest C16orf72 is a biologically relevant regulator of p53 in cancer development. In order to provide a reasonable amount of new information and set it further apart from the published study, some biochemical analysis looking into the mechanism of C16orf72 will be helpful.

Reviewer #3 Major and Minor comments:

Specific comments:

1. There appears to be a mix up in the figure legend for Fig.1A describing line 1 and 2.

We sincerely apologize for the mix up in the figure legend! In the revised manuscript, this has been fixed.

2. Fig.2. Data for some p53 mutants mentioned in the text cannot be found in the main figure 2D and supplemental figure S3A.

We apologize for this oversight – this has been corrected in the text.

3. Fig.2 E-F. The effects of FBXO42 and CCDC6 KO on endogenous mutant p53 level is small (~50% decrease). Given that mutant p53 accumulates at high levels, whether a 50% decrease has meaningful effect on its gain of function activities is questionable. The knockouts also caused a ~25% decrease in p53 mRNA (FigS3F) which makes the mechanism quite difficult to investigate further.

When assessing the functional consequences of reduced mutant p53 R273H levels upon knockout of FBXO42 or CCDC6, one must keep in mind that accumulation of mutant p53 is required for many gain-of-function properties, reducing its level may lead to suppression of tumor growth and attenuation of invasion and metastasis formation^{3, 4, 42}. We thus injected PANC-1 cells that were depleted of either FBXO42 or CCDC6 as well as control (sgAAVSI) PANC-1 cells into the tail vein of non-obese diabetic/severe combined immunodeficiency-gamma (NSG) mice, and evaluated the lung metastatic colonization. Genetic ablation of FBXO42 and CCDC6 resulted in a significant reduction of metastatic colonization relative to

the control, similar to the depletion of TP53 (R273H) (please see **new Fig. 2g and Suppl Fig. 4a**). Of note, loss of FBXO42 showed the most dramatic reduction, which was likely due to the strong knock-out efficacy compared to CCDC6 or TP53 (please see **new Fig. 3i**). Importantly, to test whether these effects were specifically mediated by mutant p53 R273H, we repeated this experiment using PANC-1 cells devoid of endogenous p53 R273H. Genetic ablation of FBXO42 or CCDC6 no longer affected the metastatic colonization of PANC-1- Δ p53 cells (please see **new Fig. 2g and Suppl Fig. 4a**). Collectively, these results suggested that the loss of FBXO42 and CCDC6 destabilize mutant p53 and attenuate mutant p53-driven metastatic colonization of the mouse lungs:

As such, we believe that these results show that the reduction is biologically important.

With regard to the transcriptional levels, the reductions shown in mRNA levels were not significant and given the design of our reporter, which controls for changes in transcription using the bicistronic RFP, we believe that the regulations is largely post-translational.

4. Fig.3B. The IP experiment using p53 shRNA and control shRNA should be done by IP of p53 followed by CCDC6 western blot. If CCDC6 IP is used as in the figure, then a CCDC6 shRNA knockdown sample should be compared to control shRNA. The current data does not rule out the possibility that CCDC6 antibody can nonspecifically pull down some p53.

We repeated the CCDC6 IP using a V5-tagged CCDC6 and a V5-antibody, which also immunoprecipitated p53 (please see Suppl Fig. 4e). We hope that this reviewer agrees that this further supports that CCDC6 interacts with p53 and that it was not the CCDC6 antibody nonspecifically pulling down p53. In addition, we used orthogonal approaches such as BioID showing proximity between CCDC6 and wt p53 in 293 cells as well as proximity ligation assay (PLA) showing an interaction of endogenous CCDC6 with endogenous p53 R273H in PANC-1 cells (please see Fig. 3a and c), further corroborating the CCDC6-p53 interaction.

5. Fig.3D. The *in vitro* pull down experiment needs specificity controls such as non affected R175H p53 core domain. The data presented would suggest that MBP-FBXO42c captured more than 1:1 molar ratio of R273H core domain, which is unusual for specific binding unless there is aggregation of p53.

We thank the reviewer for this constructive comment! In the revised version, we have incorporated several specificity controls.

Specifically, we replicated the *in vitro* pull-down experiments multiple times to verify the reproducibility of the result and added several specificity controls. Firstly, we provide Size Exclusion Chromatography data, which clearly shows that p53CD-R273H is obtained as a monomer, ruling out oligomerization or aggregation of the sample used for pull-down experiments (please see new **Suppl Fig.**

4f). The pull-down experiments had excellent reproducibility and the p53CD-R273HFBXO42c interaction could be replicated at least 4 independent times. One representative gel is shown in **revised new Fig. 3D and Suppl. Fig. 4g**). We included one prey specificity control to show that MBP-FBXO42c does not bind to unrelated GST protein, and a bait specificity control to show that p53CD-R273H does not bind to an unrelated bait (MBP-p107) (please see **new Fig. 3d and Suppl. Fig. 4g**). Additionally, we performed a more stringent specificity control to show that only p53FL but not p53CD-R273H was able to bind to the MBP-Mdm2 Bait. This matched the location of the known Mdm2-p53 binding site, which is located at the N-terminal transactivation domain (**Suppl. Fig. 4h**). Collectively, these data demonstrate a direct and moderate affinity interaction between the p53CD-R273H monomer and the FBXO42c Kelch domain.

We also include detailed methods for the recombinant protein expression and purification as well as for size exclusion chromatography experiments.

In the revised text we wrote:

'Using in vitro binding assays of recombinant proteins, we detected a direct interaction between FBXO42 Kelch domains 1-3 and the core DNA-binding domain of p53 R273H (Fig. 3d, Suppl. Fig. 4c and f and g). This interaction was specific as the p53 R273H core domain did not bind an unrelated protein of similar size such as p107 (Fig. 3d and Suppl. Fig. 4g). In addition, while wildtype p53 interacted with MDM2, the p53 R273H core domain did not, as expected given that MDM2 interacts with the N-terminal activation domain of p53 (Suppl. Fig. 4gh).'

6. To increase the impact of the current study, the authors could provide more mechanism insight on how C16orf72 regulates p53 level, which was also missing in the other published study. For example, addressing whether C16orf72 effect is dependent on MDM2. Does it cooperate with MDM2 to ubiquitinate p53. Does it promote p53 ubiquitination in the absence of MDM2, since it interacts with HUWE1. Does it act by recruiting usp7 to stabilize MDM2.

We thank the reviewer for this very constructive and thorough comment! In our revised version, we added additional mechanistic insights on how C16orf72 regulates p53:

Additional cycloheximide-chase experiments together with Nutlin treatments showed that C16orf72 regulates p53 in an MDM2 independent manner (please see **new Suppl. Fig. 6c**), further supporting the importance of C16orf72-HUWE1 axis in regulating p53 stability:

Regarding the C16orf72 effect and MDM2: Genetic ablation of MDM2 results in very high R273H mutant p53 levels. This can be rescued by re-expressing MDM2, but not by overexpressing C16orf72, indicating that MDM2 is functioning independently of C16orf72. Conversely, genetic ablation of C16orf72 also leads to high R273H levels, which can be reverted by re-expressing C16orf72, but not by re-expressing MDM2 (please see **new Suppl. Fig. 6d**):

Regarding the relationship between C16orf72 and HUWE1, a newly published paper (PMID:37167062) showed that C16orf72 (termed HAPSTR1 in the paper, the new official name, which we have now adopted also for our manuscript) binds to HUWE1 in HCT116 cells, in line with our data. This new paper also showed that C16orf72/HAPSTR1 is shuttling HUWE1 into the nucleus where HUWE1 modulates several signaling pathways, including p53 and nuclear factor κ B (NF- κ B)-mediated signaling.

To gain further insights of how C16orf72 is regulating mutant p53 levels in RPE1 cells and to test how HUWE1 is involved in this regulation, we first performed rescue experiments using wildtype C16orf72 as well as an NLS-truncation mutant C16orf72. These experiments clearly showed that the nuclear localization of C16orf72 is important for the regulation of p53. We further showed that the ability of C16orf72 to regulate p53 is completely dependent on HUWE1 (please see **new Fig 5f and g**).

To further highlight the function of C16orf72 on wildtype p53 *in vivo*, we performed additional experiments to support the notion that C16orf72 overexpression in tumor-prone mice (Pik3ca^{H1047R}) mice accelerates tumor formation by inhibiting endogenous wildtype p53 signaling. In the revised manuscript, we now show that the effect of C16orf72 is specific to p53, by overexpressing C16orf72 as well as the controls Usp7 and

Mdm2 in an additional control cohort of p53-null *Pik3ca* mice (= LSL-*Pi3kH1047R*; *p53*^{Flox/Flox}). Importantly, these new mouse cancer cohorts showed that the effect of *C16orf72* (or *Mdm2* or *Usp7*) overexpression strictly depends on the presence of endogenous wildtype p53 (please see **new Fig panel 6e**):

7. The manuscript is in a form extremely unfriendly to review, text, figures and legends are all split up at multiple locations, the pdf figures are very sluggish to scroll.

We sincerely apologize for the inconvenience. At the first submission, the manuscript and the pdf figures are assembled automatically. In the current submission, we have split the submission into three separate files, (1) main text, (2) main figures, and (3) supplemental figures, along with (4) supplemental tables as individual EXCELS. We also changed the resolution of a few images, so the overall higher resolution is retained, while still fitting into the file size limit. We hope that this will improve the reviewing, but the final format is again dictated by the online submission portal.

Reviewer #3 (Significance (Required)):

The work is significant in identifying a functionally relevant regulator of p53 stability.

Response: we thank the reviewer again for the very constructive feedback!

4th Jan 2024

Manuscript Number: MSB-2022-11350R

Title: Genome-wide CRISPR screens identify novel regulators of wild-type and mutant p53 stability

Dear Daniel,

Thank you once again for submitting your manuscript, revised after being reviewed at Review Commons. We have now heard back from the three reviewers who agreed to evaluate your revision. These are the same reviewers who reviewed the initial version of the study at Review Commons. As you will see below, the reviewers think that the study has improved as a result of the performed revisions and they support publication. As such, I am pleased to inform you that we can soon accept the manuscript for publication, pending some minor revisions listed below, all related to editorial issues.

- Our data editors have indicated the following missing information in the figure legends:
 - Please indicate the statistical test used for data analysis in the legends of figures 2f; 4a, c.
 - Please include information related to n in the legends of figures 1e-f.
 - Although 'n' is provided, please describe the nature of entity for 'n' in the legend of figure 2g.
 - For figure 4e, the scale bar for the heatmap is not numbered. This needs to be corrected.
- Please provide 5 keywords.
- The funding information provided in the manuscript text need to match the information entered in the online submission system. Currently, MSFSS 431649 is missing from the submission system.
- Please remove the 'Authors Contributions' from the manuscript. The 'Author Contributions' section is replaced by the CRediT contributor roles taxonomy to specify the contributions of each author in the journal submission system. Please use the free text box in the 'author information' section of the online submission system to provide more detailed descriptions if needed (e.g., 'X provided intracellular Ca⁺⁺ measurements in fig Y').
- There is a callout to Fig. 4G but panel 4G does not exist, please correct this.
- Please include page numbers in the Appendix Table of Contents.
- Our data integrity analyst has noted some similarities between the immunoblots for shown in Figure 3H for USP28 and p53. They have also noted a vertical line indicating potential splicing in the immunoblot for p21 in Figure 5B. We would ask you to provide source data for these blots. In case there has been an erroneous duplication in Figure 3H we would ask you to replace the blot(s) with the correct one(s).
- The Supporting Information section, Appendix Figure legends and Dataset EV legends should be removed from the manuscript text.
- The Materials and Methods section should be moved after the Discussion.

Please resubmit your revised manuscript online ****within one month**** and ideally as soon as possible. If we do not receive the revised manuscript within this time period, the file might be closed and any subsequent resubmission would be treated as a new manuscript. Please use the Manuscript Number (above) in all correspondence.

Click on the link below to submit your revised paper.

Link Not Available

All my best wishes for the New Year,

Maria

Maria Polychronidou, PhD

If you do choose to resubmit, please click on the link below to submit the revision online before 3rd Feb 2024.

Link Not Available

*** PLEASE NOTE *** As part of the EMBO Publications transparent editorial process initiative (see our Editorial at <https://www.nature.com/msb/journal/v6/n1/full/msb201072.html>), Molecular Systems Biology will publish online a Review Process File to accompany accepted manuscripts. When preparing your letter of response, please be aware that in the event of acceptance, your cover letter/point-by-point document will be included as part of this File, which will be available to the scientific community. More information about this initiative is available in our Instructions to Authors. If you have any questions about this initiative, please contact the editorial office (msb@embo.org).

Reviewer #1:

In this revised version of the manuscript the authors have adequately addressed the majority of my concerns. Even though I still would have liked to see the manuscript shortened and deliver more mechanistic insight, it does give the community more insight into the regulation of p53 stability. Therefore this manuscript can be seriously considered for publication.

Reviewer #2:

While the authors have not fully addressed all of the comments from the reviewers, I feel the paper is now sufficiently improved to allow recommendation for publication in Molecular Systems Biology.

Reviewer #3:

The manuscript by Lu and coworkers performed genome wide CRISPR screens to search for genes that when knocked out, lead to p53 accumulation or degradation. Wt p53 and a panel of p53 hotspot mutants were chosen as reporter for the screen. The approach identified many previously described regulators of p53 degradation, and also found a large set of new hits that many appear to be indirectly affecting p53 level.

The authors chose to further characterize FBXO42 as a top hit that blocks mutant p53 degradation, and C16orf72 as a top hit that promotes wt/mutant p53 degradation. The functional studies showed FBXO42 knockout has modest effect on mutant p53 level (~50% reduction) but significantly reduced the metastatic potential of Panc-1 cells in a mouse lung seeding assay. Their results of ectopic expression in a mouse model suggest C16orf72 is a biologically relevant inhibitor of p53 in promoting cancer development.

The revised paper addressed several critiques from previous review main on the lack of mechanistic data on the activities of the key hits, and provided new data to address the mechanisms of FBXO42 and C16orf72. The authors also performed additional screens to further investigate the functional connections of the hits.

Overall, the manuscript has been significantly improved and addressed many key questions raised in the previous review. The results and information provided will be instructive to investigators interested in the strategy or further explore the other candidate hits identified from this work.

Rev_Com_number: RC-2022-01563
New_manu_number: MSB-2022-11350R
Corr_author: Schramek
Title: Genome-wide CRISPR screens identify novel regulators of wild-type and mutant p53 stability

All editorial and formatting issues were resolved by the authors.

12th Mar 2024

Manuscript number: MSB-2022-11350RR

Title: Genome-wide CRISPR screens identify novel regulators of wild-type and mutant p53 stability

Dear Daniel,

Thank you again for sending us your revised manuscript. We are now satisfied with the modifications made and I am pleased to inform you that your paper has been accepted for publication.

Kind regards,

Maria

Maria Polychronidou, PhD
Senior Editor
Molecular Systems Biology

Rev_Com_number: RC-2022-01563
New_manu_number: MSB-2022-11350RR
Corr_author: Schramek
Title: Genome-wide CRISPR screens identify novel regulators of wild-type and mutant p53 stability